Hyainailourine and teratodontine cranial material from the late Eocene of Egypt and the application of parsimony and Bayesian methods to the phylogeny and biogeography of Hyaenodonta (Placentalia, Mammalia)

Borths Matthew R. 1 borths.1@gmail.com
http://orcid.org/0000-0003-1292-6356 Holroyd Patricia A. 2
Seiffert Erik R. 3
1 Department of Biomedical Sciences, Ohio University , Athens, Ohio , United States
2 Museum of Paleontology, University of California, Berkeley , Berkeley, California , United States
3 Department of Cell and Neurobiology, University of Southern California , Los Angeles, California , United States
Sues Hans-Dieter
Electronic publication date: 2016 Nov 10
Publication date: 2016
Volume: 4
Electronic Location ID: e2639
Received 2016 Jun 5; Accepted 2016 Sep 30
Copyright: © 2016 Borths et al.
Copyright year: 2016
Copyright holder: Borths et al.
License: This is an open access article distributed under the terms of the Creative Commons Attribution License, which permits unrestricted use, distribution, reproduction and adaptation in any medium and for any purpose provided that it is properly attributed. For attribution, the original author(s), title, publication source (PeerJ) and either DOI or URL of the article must be cited.
License URL: https://creativecommons.org/licenses/by/4.0/

Keywords: Paleogene, Carnivory, Bayesian systematics, Creodonta, Africa, Mammal evolution

Funding: U.S. National Science Foundation BCS-0416164, BCS-0819186, and BCS-1231288 NSF Doctoral Dissertation Improvement Grant DEB-1311354 Recent fieldwork in Egypt, and digital curation of Fayum fossils, has been supported by the U.S. National Science Foundation (BCS-0416164, BCS-0819186, and BCS-1231288), The Leakey Foundation, and Gordon and Ann Getty. M.R.B. was supported by an NSF Doctoral Dissertation Improvement Grant (DEB-1311354), a Turkana Basin Institute Graduate Fellowship, and a Stony Brook University Graduate Council Fellowship. The funders had no role in study design, data collection and analysis, decision to publish, or preparation of the manuscript.

==============================
Hyaenodonta is a diverse, extinct group of carnivorous mammals that included weasel- to rhinoceros-sized species. The oldest-known hyaenodont fossils are from the middle Paleocene of North Africa and the antiquity of the group in Afro-Arabia led to the hypothesis that it originated there and dispersed to Asia, Europe, and North America. Here we describe two new hyaenodont species based on the oldest hyaenodont cranial specimens known from Afro-Arabia. The material was collected from the latest Eocene Locality 41 (L-41, ∼34 Ma) in the Fayum Depression, Egypt. Akhnatenavus nefertiticyon sp. nov. has specialized, hypercarnivorous molars and an elongate cranial vault. In A. nefertiticyon the tallest, piercing cusp on M1–M2 is the paracone. Brychotherium ephalmos gen. et sp. nov. has more generalized molars that retain the metacone and complex talonids. In B. ephalmos the tallest, piercing cusp on M1–M2 is the metacone. We incorporate this new material into a series of phylogenetic analyses using a character-taxon matrix that includes novel dental, cranial, and postcranial characters, and samples extensively from the global record of the group. The phylogenetic analysis includes the first application of Bayesian methods to hyaenodont relationships. B. ephalmos is consistently placed within Teratodontinae, an Afro-Arabian clade with several generalist and hypercarnivorous forms, and Akhnatenavus is consistently recovered in Hyainailourinae as part of an Afro-Arabian radiation. The phylogenetic results suggest that hypercarnivory evolved independently three times within Hyaenodonta: in Teratodontinae, in Hyainailourinae, and in Hyaenodontinae. Teratodontines are consistently placed in a close relationship with Hyainailouridae (Hyainailourinae + Apterodontinae) to the exclusion of “proviverrines,” hyaenodontines, and several North American clades, and we propose that the superfamily Hyainailouroidea be used to describe this relationship. Using the topologies recovered from each phylogenetic method, we reconstructed the biogeographic history of Hyaenodonta using parsimony optimization (PO), likelihood optimization (LO), and Bayesian Binary Markov chain Monte Carlo (MCMC) to examine support for the Afro-Arabian origin of Hyaenodonta. Across all analyses, we found that Hyaenodonta most likely originated in Europe, rather than Afro-Arabia. The clade is estimated by tip-dating analysis to have undergone a rapid radiation in the Late Cretaceous and Paleocene; a radiation currently not documented by fossil evidence. During the Paleocene, lineages are reconstructed as dispersing to Asia, Afro-Arabia, and North America. The place of origin of Hyainailouroidea is likely Afro-Arabia according to the Bayesian topologies but it is ambiguous using parsimony. All topologies support the constituent clades–Hyainailourinae, Apterodontinae, and Teratodontinae–as Afro-Arabian and tip-dating estimates that each clade is established in Afro-Arabia by the middle Eocene.

Introduction

Hyaenodonta is an extinct clade of carnivorous mammals whose members were broadly distributed across Europe, North America, Asia, and Afro-Arabia during the Paleogene (Rose, 2006). In Eurasia and Afro-Arabia, some hyaenodont lineages persisted into the Miocene (Lewis & Morlo, 2010). Hyaenodonts ranged in body mass from small weasel-sized species like North American Thinocyon (Gunnell, 1998) and European Eoproviverra (Godinot, 1981) to gigantic, rhinoceros-sized species like North American Hemipsalodon (Mellett, 1969) and Afro-Arabian Megistotherium (Savage, 1973). In Europe, Asia, and North America, hyaenodonts shared carnivorous niche space with species from Carnivoramorpha, Mesonychia, and Oxyaenidae (Morlo, Gunnell & Nagel, 2010), but in Afro-Arabia, a continent that was largely isolated from all others from the Albian (Early Cretaceous, ∼100 Ma, Gaina et al., 2013) to the Miocene (∼16 Ma, Partridge, 2010), terrestrial carnivore niches were occupied almost exclusively by Hyaenodonta (Lewis & Morlo, 2010).

Historically, the first hyaenodonts recovered from Afro-Arabia were found in the early Oligocene beds of the Fayum Depression, Egypt, and were placed in genera known from Europe (Apterodon, Pterodon, and Hyaenodon) and North America (Sinopa) thereby implicitly linking hyaenodonts from the northern continents to the Fayum fauna (Andrews, 1904; Andrews, 1906). The dominant phylogenetic hypothesis at the time (Matthew, 1901; Matthew, 1906; Matthew, 1915) placed Pterodon and Hyaenodon (genera with specialized hypercarnivorous dentitions (Van Valkenburgh, 2007)) in the subfamily Hyaenodontinae, and Sinopa in the more generalized Proviverrinae. In this taxonomic arrangement, proviverrines were distinguished from other hyaenodonts by their retention of prominent metaconids on the lower molars and separated paracones and metacones on the upper molars; Proviverrinae was therefore seen as the generalized “stock” that gave rise to the more specialized hyaenodontines, which were derived in having lost lower molar metaconids, and having fused the paracone and metacone on the upper molars (Polly, 1996). Schlosser (1911) built on this phylogenetic framework in his analysis of the Fayum hyaenodonts, arguing for a North American origin of hyaenodonts from a Sinopa-like ancestor, some of which then dispersed to Europe and gave rise to Pterodon and Apterodon, before members of Pterodon, Apterodon, and Sinopa dispersed to Afro-Arabia from Europe during the late Eocene. This biogeographic hypothesis framed Afro-Arabia as something of a cul-de-sac for hyaenodont lineages that evolved during the early and middle Eocene on northern continents.

This scenario had to be reevaluated when Crochet (1988) described Koholia atlasense, a hyaenodont from the late early Eocene of Algeria that was argued to have no obvious links to North American, European, or Asian taxa. By providing evidence for the great antiquity of Hyaenodonta in Afro-Arabia, the presence of Koholia complicated the biogeographic history of the clade. The presence of Hyaenodonta in Afro-Arabia was subsequently pushed even deeper into time, first by the fragmentary remains of Tinerhodon described by Gheerbrant (1995), then with more complete material by Gheerbrant et al. (2006) with the description of the early Eocene Boualitomus marocanensis. Solé et al. (2009) pushed the antiquity of hyaenodonts in Afro-Arabia into the middle Paleocene (Kocsis et al., 2014) with the description of Lahimia selloumi. Boualitomus and Lahimia, both small-bodied species from Morocco, were hypothesized to be closely related to Koholia (Solé et al., 2009). Lahimia is the oldest-known hyaenodont from any continent, and multiple authors have recently advocated for the Afro-Arabian origin of Hyaenodonta in large part based on the great age of Lahimia (Solé, 2013; Morlo et al., 2014; Solé et al., 2014), though Gingerich & Deutsch (1989) suggested Afro-Arabia may be the center of origin for hyaenodonts before the discovery of Lahimia, based on the diversity of the group in the Fayum.

In addition to the discovery of ancient Afro-Arabian hyaenodonts, the reframing of the biogeographic history of Hyaenodonta has also been spurred by new phylogenetic hypotheses generated by parsimony-based cladistic analyses. Barry (1988) was the first to apply cladistic methodology to hyaenodont systematics. He employed 40 dental characters in his study of the relationships within Proviverrinae, particularly among the Afro-Arabian proviverrines Masrasector, Anasinopa, Metasinopa, and Dissopsalis, which were found to be paraphyletic with respect to the proviverrines Proviverra, Cynohyaenodon, Prodissopsalis, Paracynohyaenodon, and Allopterodon. The results of his analysis implied multiple dispersal events between Europe and Afro-Arabia, and between Asia and Afro-Arabia.

Polly (1996) conducted the first cladistic study that included proviverrines as well as more specialized hyaenodonts like Pterodon and Hyaenodon in the same character-taxon matrix. His study was also the first to incorporate cranial and postcranial characters. Polly found Proviverrinae to be paraphyletic, and to include at least two lineages that independently evolved specialized carnivory—Hyaenodontinae, which includes Hyaenodon, and Hyainailourinae, which includes Pterodon. Importantly, the cranial characters (particularly the construction of the nuchal crest) and postcranial characters (particularly the morphology of the astragalar-calcaneal joints) that he employed provided non-dental support for the hypothesis that hypercarnivory had evolved independently multiple times within Hyaenodonta.

More recent phylogenetic studies have focused on specific lineages within Hyaenodonta—Limnocyoninae (Morlo & Gunnell, 2003), Afro-Arabian and Asian proviverrines (Egi et al., 2005), early North American and European proviverrines (Zack, 2011; Zack & Rose, 2015), possible relatives of Apterodon (Grohé et al., 2012), European and North American proviverrines (Solé, 2013; Solé, Falconnet & Yves, 2014), and Hyainailourinae (Solé et al., 2015)—but each of these studies was limited in its biogeographic scope, and restricted its character sample primarily to dental morphology (Solé (2013) incorporated one cranial character and 14 postcranial characters into his study). This restriction is understandable because much of the hyaenodont record is composed of isolated dentaries, rostral fragments, and isolated teeth, and the inclusion of cranial and postcranial characters leaves a great deal of missing data, though simulation studies have shown that missing data is less problematic than might be expected (Wiens, 2003; Kearney & Clark, 2003; Wiens & Moen, 2008; Prevosti & Chemisquy, 2010).

Rana et al. (2015) was the first study that used most of the cranial and postcranial characters described by Polly (1996) as part of an expanded cladistic analysis, an effort further expanded by Zack & Rose (2015) in their study of North American hyaenodonts. The study of Rana et al. (2015) was the first since that of Polly (1996) to include both Hyaenodon and Pterodon in the same analysis. Two clades recovered by Rana et al. (2015) are particularly relevant to the present study—Teratodontinae and Hyainailourinae. Teratodontinae is a subfamily that was first proposed by Savage (1965) to accommodate Teratodon, a strange early Miocene hyaenodont from eastern Africa with massive premolars. Solé et al. (2014) found that many Afro-Arabian taxa formally considered to be proviverrines by Barry (1988) and Egi et al. (2005)—including Masrasector, Anasinopa, and Dissopsalis—formed a clade with Teratodon. In the topology recovered by Solé et al. (2014), Teratodontinae was the sister clade to European Proviverrinae and North American Arfia, implying that dispersal had occurred between Afro-Arabia and Europe. Rana et al. (2015) also recovered a monophyletic Teratodontinae, but found that it was more closely related to Apterodon and Hyainailourinae, two predominantly Afro-Arabian clades. In Rana et al.’s (2015) study, Hyainailourinae (similar to Polly’s Pterodontinae (= Hyainailourinae; Lewis & Morlo, 2010)) groups Pterodon species with Miocene Megistotherium/Hyainailouros and Eocene-Oligocene Akhnatenavus in an unresolved polytomy. Hyainailourinae is a cosmopolitan clade that was closely examined by Solé et al. (2015) that includes North American Hemipsalodon, several European forms (Paroxyaena, Kerberos, and Pterodon dasyuroides), Afro-Arabian “Pterodon” africanus and Akhnatenavus, and possibly Asian Orienspterodon (Egi, Tsubamoto & Takai, 2007). Rana et al. (2015) recovered Hyainailourinae as part of a polytomy with European Oxyaenoides, Afro-Arabian Koholia, and Afro-Arabian Metapterodon. Solé et al. (2015) proposed additional cranial features that distinguish Hyainailourinae from Hyaenodontinae, but these features were not incorporated into the Solé et al. (2015) phylogenetic analysis.

Here we describe several hyaenodont fossils from the latest Eocene of Egypt that bear on the content, interrelationships, and biogeography of Teratodontinae and Hyainailourinae, as well as Hyaenodonta generally. A new teratodontine genus and species is represented by two rostra and well-preserved mandibular remains, while a new hyainailourine species is represented by a largely complete, but crushed, cranium, as well as fragmentary dentaries. Both are known from sufficient dental material to facilitate estimations of body mass based on regression equations used by Van Valkenburgh (1990) and Morlo (1999). To place these species into phylogenetic context, we employed a character taxon matrix that includes 134 morphological characters and 78 taxa that builds upon previous analyses of hyaenodont systematic efforts.

All previous cladistic analyses of hyaenodont relationships (discussed above) have used parsimony analysis to reconstruct the evolution of the group. Therefore, we use parsimony for one part of our phylogenetic analysis to permit direct comparisons between topologies reconstructed here and topologies described in other studies that use the same analytical assumptions. We also apply Bayesian methods to the character-taxon matrix, a novel analytical approach for hyaenodontan systematics. Bayesian methods provide powerful tools for understanding evolutionary relationships by simultaneously estimating branch lengths, phylogenetic uncertainty, and evolutionary rates while inferring phylogenetic relationships (Holder & Lewis, 2003; Wiley & Lieberman, 2011; O’Reilly et al., 2016). As part of the Bayesian analysis, we employ a recently developed expansion of Bayesian phylogenetic inference that has been called “tip-dating” (Pyron, 2011; Ronquist et al., 2012a; Beck & Lee, 2014). In standard Bayesian phylogenetic inference, a posterior distribution of unique topologies with different branch lengths is generated using Markov chain Monte Carlo (MCMC) sampling, taking into account the data (character-taxon matrix), a model of evolution (for morphology typically the Mk model (Lewis, 2001)), and a parameter for evolutionary rate (Huelsenbeck et al., 2002; Archibald, Mort & Crawford, 2003). Clades are sampled by the MCMC process in proportion to their posterior probabilities (PP). Tip-dating is a logical extension of standard Bayesian inference that more realistically constrains rates of evolution across the tree by taking into account the actual ages of fossil taxa (Beck & Lee, 2014; Arcila et al., 2015); therefore, tip-dating provides additional information that contributes to the comparative likelihood of the branch-length-scaled topologies, and it can also be used to estimate divergence times among living and extinct taxa. This method has recently been applied to phylogenetic analysis of several clades (Schrago, Mello & Soares, 2013; Wood et al., 2013; Beck & Lee, 2014; Lee et al., 2014; Dembo et al., 2015; Arcila et al., 2015; Close et al., 2015; Sallam & Seiffert, 2016; Gorscak & O’Connor, 2016) and is applied here for the first time to hyaenodont systematics. Further details on the application of tip-dating methodology to Hyaenodonta are presented in the methods section.

A Note on the taxonomic terms “Creodonta” and Hyaenodonta

Hyaenodonta is discussed in this study as an elevation of the clade Hyaenodontidae, which has traditionally been nested within Creodonta along with another extinct family, Oxyaenidae (e.g., Matthew, 1915; Gunnell, 1998; Rose, 2006). Creodonta is then traditionally considered to be the sister clade to Carnivoramorpha in the larger clade Ferae (McKenna & Bell, 1997; Wesley-Hunt & Flynn, 2005; Spaulding & Flynn, 2012; Halliday, Upchurch & Goswami, 2015). Cope (1875) originally defined Creodonta and modified its definition through time, eventually determining that the Creodonta was part of Insectivora and that Insectivora also included Miacidae, Mesonychidae, Chrysochloridae, Centetidae, Talpidae, Mythomyidae, Oxyaenidae, and Hyaenodontidae (Cope, 1884). With additional fossil, osteological, and eventually genetic information, each of these families was moved to other orders and clades (Miacidae to Carnivoramorpha (Spaulding & Flynn, 2012); Mesonychidae as a sister group of artiodactyls and perissodactyls (Spaulding, O’Leary & Gatesy, 2009); “Centetidae” and “Mythomyidae” (tenrecs) and Chrysochloridae to Afrosoricida, and Talpidae to Eulipotyphla or Lipotyphla (Stanhope et al., 1998)) except Oxyaenidae and Hyaenodontidae, which have been retained as members of Creodonta by Gunnell & Gingerich (1991) and Gunnell (1998).

Multiple authors, first Van Valen (1966) then later Polly (1996), raised the possibility that Oxyaenidae and Hyaenodontidae are not sister taxa and that Creodonta is not a clade. This suggestion has been adopted in many recent studies (Grohé et al., 2012; Morlo et al., 2014; Solé et al., 2014), but there has been little discussion of what the sister taxon of Hyaenodontidae is if not Oxyaenidae as a whole or if these groups of mammalian carnivores are each indeed clades. Spaulding, O’Leary & Gatesy (2009), O’Leary et al. (2013) and Halliday, Upchurch & Goswami (2015) each applied cladistic methodology to an examination of large-scale relationships within Placentalia which included representatives of Ferae (Carnivora + Pholidota and possibly Creodonta). Ferae and Creodonta were monophyletic in Spaulding, O’Leary & Gatesy (2009), but their study was focused on the relationships within Cetartiodactlyla, rather than Ferae, and only included four species from Creodonta (and did not include Pholidota). O’Leary et al. (2013) also resolved a monophyletic Ferae, but only included one representative from Creodonta, the hyaenodont Sinopa rapax, thus this large-scale examination of Placentalia did not test for the monophyly of Creodonta. Halliday, Upchurch & Goswami (2015) focused on Paleocene mammal groups and found a monophyletic Ferae and a monophyletic Creodonta when all topological constraints were applied to the analysis, though their analysis was limited to four North American creodonts: the hyaenodonts Prolimnocyon and Pyrocyon and the oxyaenids Dipsalidictis and Tytthaena.

Recent phylogenetic studies that have examined relationships among non-oxyaenid creodonts have employed the order Hyaenodontida, and named clades (families and subfamilies) that reflect their inclusion in that order (Grohé et al., 2012; Solé, 2013; Solé et al., 2014; Solé et al., 2015). These studies cite Van Valen (1967) as the source of Hyaenodontida, but Van Valen (1967) actually used the suborder Hyaenodonta, a taxon that he employed to encompass Oxyaenidae, Hyaenodontidae, and Palaeoryctoidea. The name Hyaenodonta in the sense used in the present study was first used by Solé et al. (2015) to encompasses placental mammals with a carnassial complex between P4 and M1, M1 and M2, and M2 and M3 that were previously placed in Hyaenodontidae. Future analyses that sample broadly from Oxyaenidae, Hyaenodonta, and other placental orders are required to rigorously test the monophyly or polyphyly of Creodonta and the phylogenetic definition of Hyaenodonta; such an analysis is, however, beyond the scope of the current study.

Materials and Methods

Geological context

The material described here was collected from Locality 41 (L-41) in the Fayum Depression, Egypt (Fig. 1). The Fayum area preserves a near-continuous terrestrial record from the early late Eocene through the early Oligocene (Bown & Kraus, 1988). Quarry L-41 is at the top of the lower variegated sequence in the Jebel Qatrani Formation, and is interpreted to have been deposited during a period of reversed magnetic polarity (Kappelman, Simons & Swisher, 1992) that has been correlated with the Eocene-Oligocene spanning Chron C13r (Seiffert, 2006). The latest Priabonian (latest Eocene, ∼34 Ma) age of L-41 is supported by the identification of a major erosional unconformity just above L-41 that Seiffert (2006) hypothesized was caused by the major drawdown in global sea level that occurred during the earliest Oligocene (e.g., Miller et al., 2008). The age is also supported by biostratigraphic correlation with well-dated mammal sites in Oman (Seiffert, 2006), and extinctions of multiple strepsirrhine primate lineages upsection from L-41 that might have been due to earliest Oligocene cooling (Seiffert, 2007).

Figure 1 Map of the Fayum Depression, Egypt.

Stars indicate quarries. Red star indicates L-41 (latest Priabonian, ∼34 Ma) in the Jebel Qatrani Formation (Fm.). Well-defined formational contacts (Qasr el-Sagha Fm./Jebel Qatrani Fm. and Jebel Qatrani Fm./Khashab Fm.) are indicated by solid lines. The older and more ambiguous formational boundary (Birket Qarun Fm./Qasr el-Sagha Fm.) is indicated by a dashed line.

Many of the productive quarries in the Fayum are composed of a poorly consolidated fine- to medium-grained sandstone and gravel that are quarried through aeolian weathering, sweeping, and dry sieving (Bown & Kraus, 1988). In contrast, L-41 is a well-consolidated deposit that is dominated by green to yellow clay and postdepositional salt that is quarried in sheets, with fossils exposed by carefully prying apart silt and claystone bedding planes (Simons, Cornero & Bown, 1998). Vertebrate fossils are abundant at L-41 and the fine-grained matrix is capable of preserving small fossils that are delicately prepared from the clay matrix. The larger mammals known from the Fayum fauna, such as anthracotheres and hyraxes (Rasmussen & Simons, 1991) are preserved at L-41, but the quarry is particularly important for preserving the smaller components of the mammalian fauna, such as bats (Gunnell, Simons & Seiffert, 2008), rodents (Sallam, Seiffert & Simons, 2011; Sallam, Seiffert & Simons, 2012; Sallam & Seiffert, 2016), tenrecoids (Seiffert & Simons, 2000; Seiffert et al., 2007) and small primates (Simons, 1990; Simons, 1997; Simons & Rasmussen, 1996; Simons et al., 2001). Complete crania, jaws, and isolated skeletal elements are preserved in abundance, though most are crushed through post-depositional taphonomic processes (Simons, Cornero & Bown, 1998). The quarry was likely formed in the distal portion of a large freshwater lake, as suggested by the abundant preservation of freshwater fish fossils. The vertebrate remains are hypothesized to have floated into the lake during periodic flooding events and been buried with little disturbance from flowing water or predation (Simons, Cornero & Bown, 1998).

Taxonomy

The electronic version of this article in Portable Document Format (PDF) will represent a published work according to the International Commission on Zoological Nomenclature (ICZN), and hence the new names contained in the electronic version are effectively published under that Code from the electronic edition alone. This published work and the nomenclatural acts it contains have been registered in ZooBank, the online registration system for the ICZN. The ZooBank Life Sciences Identifiers (LSIDs) can be resolved and the associated information viewed through any standard web browser by appending the LSID to the prefix http://zoobank.org/. The LSID for this publication is urn:lsid:zoobank.org:pub:4EB91175-33FF-4A6C-B5B2-2F9933C0DED9. The online version of this work is archived and available from the following digital repositories: PeerJ, PubMed Central and CLOCKSS. The physical specimens described here with a CGM specimen code are deposited at the Egyptian Geological Museum, Cairo, Egypt and specimens described here with a DPC specimen code are deposited at the Duke Lemur Center, Division of Fossil Primates, Duke University, Durham, NC.

Morphological measurements and nomenclature

Dental measurements of the specimens were collected from digital photographs using ImageJ (Schneider, Rasband & Eliceiri, 2012) or with digital calipers, following the methods of Holroyd (1999). Dental nomenclature and measurements used in this description are illustrated in Fig. 2.

Figure 2 Dental nomenclature used in this study.

Upper left M2 and lower left M3 of Proviverra typica (A–E) and Pterodon dasyuroides (F–J) showing dental terminology and measurements used in this study. (A) Proviverra typica M2 in occlusal and (B) buccal views and M3 in (C) occlusal, (D) lingual, and (E) buccal views. (F) Pterodon dasyuroides M2 in occlusal and (G) buccal views and M3 in (H) occlusal, (I) lingual, and (J) buccal views. Measurements are indicated in italics. Abbreviations: ak, anterior keel; bc, buccal cingulum; bcd; buccal cingulid; cn, carnassial notch; co, cristid obliqua; ecf, ectoflexus; ed, entoconid; ecd, entocristid; hd, hypoconid; hld, hypoconulid; lc, lingual cingulum; lmdl, lower molar mesiodistal length; meh, metacone height beyond metastyle; me, metacone; mec, metaconule; med, metaconid; ms, metastyle; mtl, metastyle mesiodistal length; pa, paracone; pac, paraconule; pad, paraconid; pah, paracone height beyond metastyle; pmc, premetacrisid; pom, postmetacrista; pop, postparacrista; popad; postparacristid; popr, postprotocrista; ppc postparacristid; pr, protocone; prd, protoconid; prm, premetacrista; prp, preparacrista; prpr, preprotocrista; prprd; preprotocristid; ps, parastyle; tab, talon basin; tadb, talonid basin; tall, talonid mesiodistal length; talw, talonid buccolingual width; trb, trigon basin; trdb, trigonid basin tril, trigonid mesiodistal length; triw, trigonid buccolingual width; umdl, upper molar mesiodistal length; umw, upper molar buccolingual width.

Body mass was calculated using two sets of regression equations. The first is that of Morlo (1999), which predicts body mass based on an average of the mesiodistal lengths of M1–M3. The second regression equation was proposed by Van Valkenburgh (1990), which is based on the mesiodistal lengths of M1 in carnivorans. Although carnivorans show clear functional parallels with hyaenodonts due to their similar diets (Morlo, 1999; Van Valkenburgh, 2007; Friscia & Van Valkenburgh, 2010), their capacity for dental shearing was achieved in a different way, with carnivorans only having one pair of functional carnassials, and hyaenodontans having three (Rose, 2006). Many hyaenodont specimens have heavily worn M1s and shearing facets on M2 and M3, which suggests that the distal molars were used in adult hyaenodonts in the same way that the M1 carnassial is used by carnivorans; therefore, any results derived from Van Valkenburgh’s (1990) equation must be viewed with caution. The Van Valkenburgh (1990) equation was used in three ways, using 1) average mesiodistal molar length, 2) length of M2, and 3) length of M3.

Specimen scanning

All specimens presented in this analysis were micro-CT scanned on a Nikon XTH 225 ST scanner housed in the Duke MicroCT lab in the Shared Materials Instrumentation Facility in the Pratt School of Engineering at Duke University. All specimens described here are available for viewing and download on MorphoSource, an NSF-supported repository for 3D scan data (http://www.morphosource.org/) in Project P200. The voxel size, voltage, and amperage used for each scan are also accessioned in MorphoSource with PLY files. Three-dimensional surface models were constructed using Avizo 8.0 and were visualized using volume rendering or isosurface rendering for two-dimensional illustration.

Phylogenetic analysis

Multiple phylogenetic analyses were conducted to place the two new L-41 hyaenodonts into a broad phylogenetic framework. These analyses were also intended to test existing hypotheses about relationships among multiple recently proposed hyaenodont clades (e.g., Polly, 1996; Egi et al., 2005; Solé, 2013; Solé, Falconnet & Yves, 2014; Solé et al., 2014; Solé et al., 2015; Rana et al., 2015) with new character data. Of particular interest for this study are the structure of, and relationships within, Teratodontinae, Hyainailourinae, and Hyaenodontinae, the latter clade having only been incorporated into two other cladistic analyses (Polly, 1996; Rana et al., 2015) with an in-group expanded beyond Hyaenodon (note that Bastl, Nagel & Peigné (2014) limited their analysis to evaluation of the genus Hyaenodon). The character taxon matrix used in this study includes 134 discrete dental, cranial, and postcranial characters and 78 operational taxonomic units (OTUs—four outgroup taxa and 74 hyaenodonts). The ingroup was selected to include taxa with associated dental, cranial, and postcranial material; taxa that had been utilized in previous studies that included Afro-Arabian OTUs; and taxa from significant temporal or biogeographic contexts. An attempt was made to include relatively few specimens that were not directly observed either as fossils or casts.

For this study three basal eutherian taxa—Early Cretaceous Eomaia scansoria from China (Ji et al., 2002), Late Cretaceous Maelestes gobiensis from Mongolia (Wible et al., 2007; Wible et al., 2009), and Late Cretaceous Altacreodus magnus from North America (Lillegraven, 1969; Kielan-Jaworowska, Cifelli & Luo, 2004; Fox, 2015)—were included as outgroups for each analysis. This follows the outgroup selection used in previous phylogenetic analyses of Hyaenodonta (Polly, 1996; Zack, 2011; Zack & Rose, 2015; Solé et al., 2014; Rana et al., 2015; Zack & Rose, 2015). Species level OTUs were used for all taxa except Teratodon and Lesmesodon which were a composite of specimens referred to these genera.

Some of the characters used in this analysis were sampled from previous studies including those of Polly (1996), Egi et al. (2005), Zack (2011) and Solé et al. (2014). Some of these characters were modified by concatenating similar characters and anatomical terminology was modified to make the character descriptions consistent. Characters were expanded with additional character states and 65 new characters are described, some initially proposed as “features” in Solé et al. (2015). Inapplicable characters were reductively coded (Strong & Lipscomb, 1999). Seventeen multistate characters were treated as ordered following the recommendations of Slowinski (1993) in designating these characters and all characters were equally weighted. All characters are listed in Table S1 with relevant citations, and ordered characters are noted. All OTUs were rescored for each character in the analysis. Codings for each taxon are provided in Data S1 and references, including age, Formation, and locality, are listed in Table S2. Character descriptions, nexus files, and photographs of the specimens described in this study are also available on Morphobank (Project 2336) (http://www.morphobank.org/).

Parsimony analysis

Maximum parsimony analysis was performed in Tree Analysis using New Technology software package (TNT) version 1.1 (Goloboff, Farris & Nixon, 2008). The traditional search heuristic search algorithm was used across 10,000 replicates with random addition sequence and tree bisection and reconnection (TBR) branch swapping, holding 10 trees per TBR replicate. Consistency index (CI) and retention index (RI) values were calculated using STATS.RUN in TNT. Support for each node in the maximum parsimony analysis was calculated by running 10,000 bootstrap pseudoreplicates (Felsenstein, 1985) and Bremer support was calculated for each node (Bremer, 1994) in TNT. Parsimony character optimization across all most parsimonious trees (MPTs) was conducted in PAUP 4.0 (Swofford, 2003).

Standard Bayesian inference

Bayesian phylogenetic inference analysis was performed in MrBayes 3.2.3 (Ronquist et al., 2012b). The Mk model (Lewis, 2001) for morphological data was selected and the data type was set to “standard” with coding set to “variable” (Clarke & Middleton, 2008). The analysis was run for 50 × 106 generations. Two runs were performed simultaneously with four Markov chains, three of which were heated (temp = 0.02). A total of 2,000 generations were sampled (every 5,000th generation of the 10 × 106 generations, to avoid autocorrelation), the first 500 (25%) of which were discarded as burn-in. After the analysis was run, convergence was examined using the effective sample sizes and average standard deviation of split frequencies for the final generation. The resulting PP for the standard Bayesian analysis are listed to the right of the relevant node in the “allcompat” (majority rule plus compatible groups) tree. Parsimony character optimization for the “allcompat” tree was conducted in PAUP 4.0 (Swofford, 2003).

Bayesian “tip-dating”

Bayesian “tip-dating” takes into account the relationships between morphological character evolution and the temporal succession of fossil taxa to simultaneously infer rates of morphological evolution and phylogenetic relationships. The reconstructed evolutionary rates are taken into account in estimates of phylogenetic relationships, and divergence dates between all included taxa are estimated (Ronquist et al., 2012a; Lee et al., 2014). Note that this method does not operate directly upon a stratigraphic character as is utilized in stratocladistic methods. Instead, the inferred branch length from the Bayesian phylogenetic inference is divided by the tip-age to generate an implied rate along the branch and estimate divergence dates for branches. Beck & Lee (2014) showed that when a temporal constraint is imposed on an in-group, deeply nested but ancient taxa can be recovered at nodes where rapid evolutionary change is taking place and evolutionary rates are consistent across sister nodes. Tip-dating is an interesting alternative to parsimony analysis or standard Bayesian inference for inferring phylogenetic relationships among members of “explosive” adaptive radiations, such as those which are thought to have taken place in the placental mammalian lineage near the K-Pg extinction event (O’Leary et al., 2013). Rapid radiations might be expected to pose problems for parsimony analysis in particular, because rapid evolutionary change near the base of a radiation might be overwritten by subsequent evolution along long branches (Felsenstein, 1978) as lineages invade open niche space. Tip-dating seems especially appropriate for Hyaenodonta, given the variable phylogenetic positions occupied by the oldest Afro-Arabian hyaenodontans Lahimia, Boualitomus, and Tinerhodon in the analysis by Rana et al. (2015). These taxa were recovered deeply nested within Hyainailourinae in some MPTs and in very basal positions in other MPTs, a result consistent with early phylogenetic experiments performed with this data set using only dental characters.

The tip-dating analysis presented here was run in MrBayes 3.2.3 (Ronquist et al., 2012b) following methods employed by Beck & Lee (2014). The Mk model was used to model morphological character change and the independent gamma rates (IGR) relaxed clock model (Lepage et al., 2007; Ronquist et al., 2012a), which assumes no autocorrelation of rates in the phylogeny, was used to infer divergence ages from terminal taxa and reconstruct rates of morphological evolution. Tip-dating requires specific dates for each terminal taxon. Each OTU, with citations justifying its assigned date range, are listed in Table S2, from the oldest taxon in the analysis (Eomaia, 129.7–122.1 Ma) to the youngest (Dissopsalis, 15–9 Ma). The root of the tree was set with a prior of 120–130 Ma (Wible et al., 2009; O’Leary et al., 2013). Beck & Lee (2014) demonstrated that, in the case of placental mammalian supraordinal phylogeny, node ages tend to be reconstructed as particularly ancient and in extreme conflict with the fossil record if an in-group constraint is not applied. The prior for the divergence date of Hyaenodonta was set conservatively to be between 62 and 75 Ma, bracketing before and after the K/Pg boundary. This prior is also consistent with the estimated divergence date for Ferae, 63.8 Ma, a divergence proposed by O’Leary et al. (2013) and with age estimates for crown eutherians described by Halliday, Upchurch & Goswami (2016). The analysis was run for 50 × 106 generations. The priors that produced the strongest convergence across all parameters was clockratepr = normal (0.01, 0.007), and igrvarpr = exp(3). Two runs were performed simultaneously with four Markov chains, three of which were heated (temp = 0.02). A total of 10,000 generations were sampled, the first 25% of which were discarded as burn-in. The “allcompat” tree that results from the analysis includes evolutionary rate estimates for each branch. Beck & Lee (2014: 3) noted that rate estimates tend to have “strongly positively skewed distributions” and they advocated for the use of the median evolutionary rate rather than the mean evolutionary rate in discussions of branch evolution. Relative rates for each node are calculated by comparing the % change/Ma for a given node to the % change/Ma across the entire tree. Absolute % change/Ma was calculated by multiplying the relative median rate results for each branch by the median clock rate value for the entire tree, which is contained in .pstat output file, then multiplying by 100 to express the resulting value as a percentage.

Biogeographic methods

Three separate biogeographic methods were applied to the phylogenetic topologies recovered through parsimony analysis (strict consensus trees), standard Bayesian analysis (allcompat tree), and tip-dating Bayesian analysis (allcompat tree). The three biogeographic methods were ancestral state reconstruction using parsimony optimization (PO) (Brooks, 1990), likelihood optimization (LO) (Maddison & Maddison, 2015), and Bayesian Binary MCMC (BBM) (Yu et al., 2015). Four continental areas were designated (Afro-Arabia, Asia, Europe, and North America) for each analysis and each OTU was assigned to the continent where it was found (Table S2). While India was technically part of a separate continental mass from Asia for some portion of the Cretaceous and Cenozoic (Chatterjee, Goswami & Scotese, 2013) for the purposes of this study India will be considered part of the Asian biogeographic category.

PO of a continental biogeographic character not used in the phylogenetic analysis was implemented in Mesquite (Maddison & Maddison, 2015) using Mesquite’s Parsimony Ancestral States reconstruction. Ambiguous reconstructions are interpreted as equally parsimonious continental reconstructions for the origin of a clade. LO of the continental biogeographic character was also implemented in Mesquite using the Likelihood Ancestral States reconstruction with the model Mk1 (equally probable state change). Likelihood analysis incorporated branch length information from the standard Bayesian and tip-dating allcompat trees. Branch lengths for the maximum parsimony tree were all equal.

BBM, a statistical method for inferring ancestral states such as biogeographic distributions using Bayesian inference, was performed in RASP version 3.1 (Yu et al., 2015). The number of areas from which a lineage could originate was limited to one to model dispersal rather than vicariance events. Dispersal is a more likely explanation for the distribution of Hyaenodonta during the Late Cretaceous and early Paleogene than vicariance given global paleogeography during this interval. Vicariance would imply an origin for Hyaenodonta that proceeds the break-up of Pangea, which the fossil record does not currently support. The results of the analysis as expressed as the probability of a given clade originating from one of four continental areas. The MCMC analysis was performed over 10 × 106 generations with 10 Markov chains, sampling every 100 generations, with the temperature set to 0.1. The first 100 trees were discarded as part of the burn-in period, and the Jukes-Cantor model was used, with equal among-site rate variation.

Systematic Paleontology

HYAENODONTA Van Valen, 1967 sensu Solé et al., 2015

TERATODONTINAE Savage, 1965

BRYCHOTHERIUM Borths, Holroyd, and Seiffert, gen. nov.

urn:lsid:zoobank.org:act:A39C1414-CF72-4FDC-A087-9912FCEDB0C8

Type species: Brychotherium ephalmos, sp. nov.

Etymology: Meaning “greedily eating beast” in Greek from brycho (βρύχω) meaning to eat greedily or noisily and thēríon (θηρίον) meaning beast. The name was first used by Holroyd (1994) in her doctoral dissertation, and was subsequently used as a nomen nudum by Egi et al. (2005) and Solé et al. (2014).

Generic diagnosis: As for type species.

A note on the genus: Brychotherium was originally coined and recognized as a distinct genus in a dissertation (Holroyd, 1994), and was therefore not yet validly published under ICZN rules. Subsequent studies (e.g., Egi et al., 2005; Solé et al., 2014) have used the genus or lumped it into “African Sinopa spp.” (e.g., Rana et al., 2015), based solely on the lower dentition. We formally name the taxon here, with a diagnosis that includes the more complete sample now available including rostra and upper dentition. Notably, this formal diagnosis does not include all specimens initially assigned to the genus in Holroyd (1994), as an expansion of the L-41 sample in the last 22 years has further refined the understanding of the similarly-sized hyaenodont fauna at the locality. Further, Brychotherium does not formally include “Sinopa” ethiopica (Andrews, 1906). The position of “S.” ethiopica awaits further comparative work to place it in a larger phylogenetic context.

BRYCHOTHERIUM EPHALMOS Borths, Holroyd, and Seiffert, sp. nov.

(Figs. 3–9; Tables 1 and 2)

urn:lsid:zoobank.org:act:BCAACF37-E200-4172-A875-C4D5F6FFCEFB

Figure 3 Brychotherium ephalmos DPC 11990 rostrum.

Brychotherium ephalmos, gen. et sp. nov., DPC 11990, rostrum with left and right P4–M3 and alveoli for right and left I2–P3. (A) ventral view; (B) dorsal view; (C) left lateral view; (D) right lateral view. Specimen crushed mediolaterally with left maxilla shifted anteriorly relative to right maxilla.

Figure 4 Brychotherium ephalmos DPC 11990 sketch and digital rendering based on high-resolution micro-CT scans.

Brychotherium ephalmos gen. et sp. nov., DPC 11990; rostrum with P4–M3 with left and right P4–M3: and alveolus for right and left I2–P3. Sketch on the left (subscript 1) and digital rendering on the right (subscript 2): (A) ventral view; (B) dorsal view; (C) left lateral view; (D) right lateral view. Solid lines indicate definite sutures, dotted lines indicate interpreted sutures that have been obscured by crushing. Digital surface model is available on MorphoSource.

Figure 5 Brychotherium ephalmos DPC 17627 rostrum.

Brychotherium ephalmos gen. et sp. nov., DPC 17627, rostrum with left canine, dP4–M3 (M3 erupting) and alveolus for dP3 and right P4–M2; specimen photographs on the left (subscript 1) and digital rendering on the right (subscript 2): (A) occlusal view of left dentition, buccal aspect of right dentition visible; (B) buccal view of left dentition, protocones of right P4–M2 and M2 paracone and metacone visible; (C) lingual view of left dentition, buccal aspect of right dentition visible. Postmortem distortion involuted right side of rostrum. Occlusal portions of right dentition protrude through left maxilla. Digital surface model is available on MorphoSource.

Figure 6 Brychotherium ephalmos DPC 17627 dentary.

Brychotherium ephalmos gen. et sp. nov., DPC 17627, right dentary with P4–M3; specimen photographs on the left (subscript 1) and digital model on the right (subscript 2); (A) occlusal view; (B) lingual view; (C) buccal view. Digital surface model is available on MorphoSource.

Figure 7 Brychotherium ephalmos CGM 83750 dentary.

Brychotherium ephalmos gen. et sp. nov., CGM 83750, right dentary with C–M3; specimen photos on the left (subscript 1) and digital model images on the right (subscript 2); (A) occlusal view; (B) lingual view; (C) buccal view. Digital surface model is available on MorphoSource.

Figure 8 Brychotherium ephalmos DPC 11569A right dentary.

Brychotherium ephalmos gen. et sp. nov., DPC 11569A, right dentary with C, P2–M3; specimen photos on the left (subscript 1) and digital model images on the right (subscript 2); (A) occlusal view; (B) lingual view; (C) buccal view. Digital surface model is available on MorphoSource.

Figure 9 Brychotherium ephalmos DPC 11569B left dentary.

Brychotherium ephalmos gen. et sp. nov., DPC 11569B, left dentary with C, P2–M3; specimen photos on the left (subscript 1) and digital model images on the right (subscript 2); (A) occlusal view; (B) lingual view; (C) buccal view. Digital surface model is available on MorphoSource.

Table 1 Specimen measurements for the upper dentition of Brychotherium ephalmos.

Brychotherium ephalmos	Locus	Length	Width	Metastyle length	Paracone height	Metacone height	Paracone base length	Metacone base length	
DPC 11990	P4	7.12	6.53	2.4	—	—	3.72	—	
(left side)	M1	8.44	7.42	3.87	2.46	3.01	2.21	2.48	
M2	9.66	9.65	4.35	3.3	4.67	2.3	2.94	
M3	3.37	11	—	1.93	∼0.83	2.03	0.9	
DPC 17627	dP4	7.72	5.95	3.36	—	—	1.67	2.75	
(left side)	M1	8.73	7.05	4.13	2.47	3.21	1.81	2.89	
M2	9.91	9.82	4.74	3.54	4.55	2.22	3.47	
Note:

Length, maximum mesiodistal length; Width, maximum buccolingual width; Metastyle length, maximum mesiodistal length from base of the paracone (premolars) or metacone (molars); Paracone height, paracone height from alveolar margin to the apex; Metacone height, metacone height from alveolar margin to the apex; Paracone length, paracone mesiodistal length at base of cusp; Metacone length, metacone mesiodistal length at base of cusp.

Table 2 Specimen measurements of the lower dentition of Brychotherium ephalmos.

Brychotherium ephalmos	Element	Max. length	Max. trigonid length	Max. talonid length	Max. trigonid width	Max. talonid width	Talonid height	Paraconid height	Protoconid height	
DPC 17627	P4	7.15	5.48	1.49	3.65	3.18	3.54	2.6	6.08	
M1	7.56	4.85	2.52	4.18	3.2	2.98	∼4.17	5.8	
M2	9.55	5.4	3.86	5.1	3.47	3.08	6.23	8.85	
M3	10.19	7.22	2.8	5.12	2.31	2.55	7.12	9.48	
CGM 83750	C	5.75	—	—	—	—	—	—	—	
P1	5.33	3.63	1.45	2.1	2.21	1.52	—	∼2.62	
P2	6.52	5.06	1.47	3.08	2.85	1.48	2.67	∼3.8	
P3	6.7	5.45	1.23	3.31	3.13	2.1	1.8	∼3.7	
P4	7.01	5.21	1.86	3.6	3.57	2.7	2.8	∼4.75	
M1	6.25	4.24	2.06	3.89	3.49	2.67	∼2.16	∼3.6	
M2	8.17	5.29	2.57	4.61	3.97	2.95	∼3.98	∼5.95	
M3	9.35	6.49	2.97	5.43	3.55	2.69	5.23	7.29	
DPC 11569A	C	5.35	—	—	—	—	—	—	—	
P2	5.18	4.33	0.81	2.51	1.69	0.61	—	3.4	
P3	6.37	5.2	1.02	2.5	1.72	1.07	1.09	∼3.10	
P4	6.94	5.55	1.41	3.1	2.19	2.37	1.62	5.21	
M1	6.06	3.77	2.35	3.02	2.34	2.24	∼3.83	∼4.33	
M2	∼7.68	5.07	∼2.57	4.13	∼2.27	—	4.46	6.52	
M3	9.68	6.81	2.81	4.92	2.46	1.93	5.91	8.24	
DPC 11569B	C	4.38	—	—	—	—	—	—	—	
P2	∼4.6	—	1.17	2.31	1.84	0.78	—	—	
P3	6.68	5.05	1.68	2.31	2.1	1.31	1.13	3.6	
P4	—	—	∼1.66	—	2.46	∼1.76	—	—	
M1	6.16	4.07	2.09	2.73	2.28	2.39	∼3.20	∼4.95	
M2	7.52	4.95	2.62	3.95	2.42	2.88	4.54	6.32	
M3	9.07	6.52	2.54	4.69	2.19	1.77	5.87	7.74	
Tooth length		C	P1	P2	P3	P4	M1	M2	M3	
N		3	1	2	3	3	4	3	4	
Mean (Std. Dev.)		5.16 (0.70)	5.33	5.85 (0.95)	6.58 (0.19)	7.03 (0.11)	6.51 (0.71)	8.41 (1.04)	9.57 (0.48)	
Notes:

Max. length, maximum mesiodistal length; Max. trigonid length, maximum mesiodistal length of trigonid; Max. talonid length, maximum mesiodistal length of talonid; Max. trigonid width, maximum buccolingual width of trigonid; Max. talonid width, maximum buccolingual talonid width; Talonid height, tallest point on talonid to alveolar margin; Paraconid height, apex of paraconid to alveolar margin; Protoconid height, maximum height from cristid obliqua to cusp apex. Std. Dev., standard deviation.

Measurements preceeded by “∼” indicate measurement taken from a heavily worn cusp.

Summary statistics only include specimens with minimal wear.

Etymology: Meaning “pickled in salty brine” in Greek from ephalmos (έϕαλμος) in reference to the high post-depositional salt content in the sediments of L-41.

Holotype: CGM 83750, right dentary with canine–M3.

Referred specimens: DPC 17627, right dentary with P4–M3 and rostrum with canine, dP4–M2, and erupting P4. Specimens were associated and probably represent a single individual; DPC 11990, rostrum with left and right P4–M3; DPC 11569A, right dentary with canine, P2–M3; DPC 11569B, left dentary with P2, P3, M1–M3.

Type locality: Locality 41 (L-41), Jebel Qatrani Formation, Fayum Depression, Egypt.

Age and distribution: Late Eocene, latest Priabonian, ∼34 Ma (Seiffert, 2006). Only known from Locality 41, approximately 14.5 km west of Qasr el-Sagha Temple, and 2 km north of the contact between the Qasr el-Sagha Formation and the Jebel Qatrani Formation.

Diagnosis: Differs from early Oligocene Masrasector species by being larger; having relatively narrow talonid basins on the lower molars that taper distally toward the hypoconulids, rather than being buccolingually wide and box-like; having tall lower molar trigonids that are more than twice the height of the talonid, rather than being less than half the height of the trigonid; having relatively small lower molar metaconids, rather than having metaconids that are nearly subequal in height to paraconid; and having preprotocristids and postprotocristids that more closely parallel the long axis of the horizontal ramus, rather than angling steeply lingually. Differs from middle-late Miocene Dissopsalis by being smaller; having a pronounced metaconid on M3 rather than a metaconid that is very reduced or absent; having a larger and more complex M3 talonid rather than a very reduced M3 talonid with poorly developed cusps; having preprotocristid and postparacristid oriented somewhat lingually relative to the horizontal ramus rather than being nearly parallel to the long axis of the horizontal ramus; having taller paracones on M1 and M2 that are only slightly shorter than metacones, rather than having reduced paracones that are distinctly shorter than metacones; and having wide upper molar protocones that are more lingually placed relative to paracone, rather than having narrow protocones that are shifted distally relative to the paracones. Differs from early Miocene Anasinopa by being smaller; having taller, but mesiodistally short, lower molar trigonids rather than relatively low and long trigonids; having a more buccolingually compressed P4 rather than a buccolingually broad P4; having relatively elongate upper molar metastyles that form a deep and distinct ectoflexus on M2, rather than short metastyles that form a relatively shallow M2 ectoflexus; having more buccolingually compressed paracone and metacone cusps (with elliptical cross-sections) on the upper molars, rather than having paracone and metacone cusps that are more connate with rounded cross-sections; and having upper molar paracones that are relatively large, when compared to the size of the metacone, rather than having paracones that are relatively low and mesiodistally much shorter than metacones. Differs from early or middle Eocene Furodon by having a relatively short P4 with a more distinct paraconid; having a P4 protoconid whose long axis in buccal view is perpendicular to the alveolar margin, rather than being distally inclined; having relatively low entocristids, rather than tall entocristids that close the lower molar talonids lingually, especially on M3; and having upper molar metacones that are mesiodistally longer and taller than the paracones. Differs from early or middle Eocene Glibzegdouia by having an M1 metaconid that is shorter than the M1 paraconid, rather than an M1 metaconid that is taller than the M1 paraconid; having an M1–M2 trigonid that is more than twice the height of the talonid, rather than having an M1–M2 trigonid that is low compared to talonid; having M1–M2 talonids that open lingually, rather than closed by a notched entocristid; having indistinct M1–M2 entoconids rather than clear entoconid cusps; having M1–M2 trigonids that are buccolingually wider than the talonids, rather than having M1–M2 trigonids that are of the same buccolingual width as the talonids; and having a shallower M1 ectoflexus and an elongate metastyle that is of approximately the mesiodistal length of paracone/metacone base, rather than a deep M1 ectoflexus with a metastyle that is shorter than the paracone/metacone base. Differs from early Miocene Teratodon by having a P4 with multiple cusps, rather than a bulbous P4; having M1–M3 trigonids that are more than twice the height of the talonid; having buccolingually narrow upper premolars that are not buccolingually wider than they are mesiodistally long; and having a mesiodistally elongate M2 metastyle that parallels the buccal margin, rather than a metastyle that is shorter than the paracone/metacone base that angles lingually from buccal margin.

Description

Rostrum

DPC 11990 (Figs. 3 and 4) is a crushed rostrum referred to Brychotherium ephalmos. The specimen preserves most of the anterior part of the cranium, from the premaxilla back to the palatines, along with the left and right P4–M3. Like many specimens from L-41, the specimen is crushed and many of the cranial bones are fragmentary, making sutures difficult to interpret. Most of the distortion occurred through mediolateral crushing combined with minor anterior-posterior shear. The left side of the rostrum is better preserved than the right. The rostral remains of DPC 17627 also preserve dP4–M3; portions of the lateral and palatal aspects of the left maxilla are relatively undistorted.

The premaxilla preserves the alveoli of I2–I3 and it frames the partially preserved nasal aperture. Though the region is distorted, it is clear that the anterior and posterior borders of the premaxilla incline dorsally and posteriorly and, as such, the nasals were somewhat retracted, leaving the dorsal face of the palatal process of the premaxilla visible in dorsal view. The premaxilla-maxilla suture traces the anterior margin of the canine alveolus. Neither canine is preserved in DPC 11990 but the collapsed alveoli are present and indicate that the root of the canine was wide and arched posteriorly over both roots of P1 and the anterior root of P2. From the nasal aperture, the nasals become broader posteriorly. The nasal does not contact the lacrimal; instead, there is an intervening maxilla-frontal suture. The facial process of the maxilla is broad, and perforated by the infraorbital foramen dorsal to the anterior root of P3. The maxilla does not contribute to the anterior margin of the orbit; instead the dorsoventrally tall lacrimal has a broad facial process that extends anteriorly at least as far as the distal root of P4. A prominent lacrimal tubercle is present on the anterior margin of the orbit, and a wide lacrimal canal is completely contained within the orbit. The anterior margin of the orbit is positioned above the distal root of M1. The inferior margin of the orbit is formed by the jugal, which has a broad contact with the lacrimal, excluding the maxilla from the orbital margin. The jugal process of the maxilla is preserved along with fragments of the jugal. The zygomatic arch was robust and dorsoventrally deep. The dorsal portion of the orbital margin is formed by the frontal, which contacts the lacrimal dorsal to M1. No postorbital process protrudes from the frontal. The linea temporalis on the frontal has a low relief and trends medially from a lateral position near the superior orbital margin toward the origin of the sagittal crest.

In ventral view, the palatal processes of the maxilla preserve the large alveolus of the canine, two P1 alveoli, two P2 alveoli, and two P3 alveoli. The rostral portion of the palate is narrow, but the palate expands laterally near the distal root of P3. The maxilla contacts the palatine midway between the protocones of M1 and M2. A distinct palatine torus is present just distal to the M3 protocone. The internal choana originates posterior to M3. As the choana opens distally, it is framed by the left and right palatines, which trend laterally.

Upper dentition

The alveoli of I2 and I3 are preserved in the premaxilla, but it is difficult to discern whether an I1 alveolus is present. The diameter of the I3 alveolus is approximately twice the size of the I2 alveolus, while the diameter of the canine root and alveolus is approximately twice the diameter of the I3 alveolus. DPC 17627 preserves the crown of the canine, which has crenulated enamel and is buccolingually compressed. The collapsed alveolus of the canine is preserved in DPC 11990 and it arches posteriorly toward the nasal. P1 had two roots, and the mesial root was smaller than the distal root and set very close to the root of the canine. The crown of P1 would have been close to, or in contact with, the base of the canine. P2 and P3 also had two roots though the anterior alveolus of P3 is not well preserved. The premolars, from P1 to the anterior root of P3, were in the same anteroposterior plane and the margins of the maxilla holding these teeth were parallel. At the posterior root of P3 the palate flares laterally and broadens distally.

P4 is the only premolar whose crown is preserved. The parastyle is buccolingually compressed, forming a crista that connects with the preparacrista. A thin buccal cingulum surrounds the parastyle and runs along the base of the paracone to the base of the metastyle. The preprotocrista connects the base of the parastyle to the protocone, forming a distinct mesial shelf along the base of the paracone. The protocone is mesiodistally wide and connate, though much lower than the paracone, and the preprotocrista and postprotocrista form a broad equilateral triangle around the base of the paracone. The protocone is shifted slightly mesially relative to the paracone. The paracone is ellipsoid in cross-section and the postparacrista tapers to a sectorial blade that connects with the buccolingually compressed metastyle. The metastyle forms a distinct carnassial notch with the postparacrista, and the metastyle rises distally from the notch to approximately one-third the height of the paracone.

DPC 17627 represents a subadult individual and provides insight into an earlier ontogenetic stage than that of DPC 11990. The specimen preserves the right P4, which has fully erupted, and the left dP4, which still has its roots. The left M3 was erupting into occlusion. The parastyle of dP4 is wide and shelf-like, leaving space between the base of the paracone and the cusp of the parastyle. The parastyle connects the wide buccal cingulum to the preprotocrista. A large paraconule forms a distinct crest along the preprotocrista, which slopes to the protocone where the paraconule and protocone form a notch. The protocone is large and triangular and almost rises to the point of divergence between the paracone and metacone. No lingual cingulum is evident along the base of the protocone. The protocone is shifted slightly mesially, as it is on P4. The postprotocrista slopes to a small metaconule, which is very reduced compared to the paraconule. The postprotocrista terminates at the base of the metacone rather than coursing along the base of the metastyle. The paracone and metacone are heavily worn, though the cross-sections of both indicate that the cusps were buccolingually compressed, especially the postparacrista and the premetacrista, which together form a distinct notch where the cusps diverge. The postmetacrista forms a carnassial notch with the long metastyle. The metastyle is subequal in mesiodistal length to the mesiodistal length of the paracone/metacone base. The buccal face of the metastyle slopes steeply to the thin buccal cingulum, which traces the alveolar margin without forming an ectoflexus. The metastyle of dP4 contacts the parastyle of M1 at its mesial-most point. In DPC 11990, the metastyle of P4 also contacts the mesial-most point of M1.

M1 is generally similar to dP4. The parastyle forms a broad shelf between the apex of the parastyle and the base of the paracone. The parastyle is connected to the broad buccal cingulum, which forms a very slight ectoflexus near the base of the paracone. The buccal cingulum rises slightly along the base of the paracone then slopes distally along the base of the metacone and terminates at the base of the metastyle. The preprotocrista terminates at the base of the paraconule, forming a distinct notch between the paraconule and protocone, and a preparaconule crista courses from the apex of the paraconule to the parastyle, forming a broad mesial cingulum. The protocone rises to a prominent cusp that is equal in height to the divergence between the paracone and metacone; it has a more mesial position, relative to the paracone, than the protocone of P4. There is no lingual cingulum. The postprotocrista slopes steeply to the metaconule, which is not as mesiodistally broad as the paraconule, though still distinct. The metaconule does not have a postmetaconule crista connecting to the metastyle; instead, the postmetaconule crista abuts the lingual face of the metacone. The paracone is buccolingually compressed with a distinctly elliptical cross-section. The apex projects mesially and overhangs the parastylar region. The postparacrista is blade-like and, in buccal view, meets the premetacrista at a right angle. The metacone is more buccolingually compressed than the paracone near its apex, and is mesiodistally longer. The postmetacrista is blade-like and slopes to a deep carnassial notch at the junction with the metastyle. The mesiodistal length of the sectorial metastyle is subequal to the mesiodistal length of the paracone/metacone base. The lingual face of the metastyle is perpendicular to the palate, while the buccal face of the metastyle slopes more gently to the buccal cingulum.

M1 contacts M2 at the mesial-most point of the parastyle. M2 is similar in many ways to M1, with many of the distinctions between dP4 and M1 expressed even more extremely between M1 and M2. The parastyle of M2 is shelf-like with a broad region between the parastyle and paracone, but the parastyle is more buccolingually narrow. M2 has a deeper ectoflexus than M1 though its depth is variable, with the M2 ectoflexus on DPC 11990 deeper than the M2 ectoflexus on DPC 17627. The paraconule of M2 is very pronounced and forms most of the mesial border of a broad talon basin. The protocone projects as far lingually as the protocone of M1, leaving the protocone buccolingually more elongate than the protocone of M1. The metaconule of M2 is more reduced, compared to the size of the paraconule, than that of M1. On M2 the metaconule only forms a slight ridge. The postmetaconule crista runs along the base of the metacone and terminates at the base of the metastyle. The paracone and metacone are more buccolingually compressed and the paracone is lower than the metacone, projecting mesially from the metacone. The metacone is relatively wider and its long axis is aligned closer to perpendicular to the palate. As on M1, the sectorial postmetacrista forms a deep carnassial notch where it meets the metastyle. The metastyle rises distally from the notch before tapering to its distal-most point.

M3 is reduced primarily to a long parastyle and the paracone and protocone cusps. The parastyle contacts the distal-most point of M2. Mesially, the parastyle connects to the preparacrista, forming a steep mesial face. The protocone projects as far lingually as the protocone of M2 and it frames a deep trigon basin. The protocone rises close to the height of the paracone. The paracone is more connate than the paracone of M2, though the postparacrista is buccolingually compressed. The postparacrista terminates at the buccal cingulum, which connects the parastyle to the postprotocrista. The buccal cingulum rises slightly near the distal aspect of the paracone.

Dentary and lower dentition

The holotype of Brychotherium, CGM 83750, is a right dentary that preserves the lower dental row from the canine to M3. The cusps of CGM 83750 are worn, particularly the premolars and M1. Three other dentary specimens are referred to Brychotherium ephalmos: DPC 11569A (right dentary), DPC 11569B (left dentary), and DPC 17627 (left dentary). CGM 83750 is the only specimen with a complete coronoid process and tooth row distal to the canine. There is variation among the dentary specimens referred to Brychotherium ephalmos. This description will first refer to the morphology preserved by CGM 83750, then will address the morphological variation present in the referred specimens.

The horizontal mandibular symphysis is rugose and was unfused. The symphysis extends distally to the mesial root of P3. There are multiple mental foramina preserved along the buccal aspect of the horizontal ramus. The most rostral mental foramen is ventral to the mesial root of P1. The second mental foramen is ventral to the mesial root of P2 and the third mental foramen is the largest and is ventral to the space between the distal root of P3 and the mesial root of P4. The ventral margin of the corpus of the dentary gently curves to the partially preserved angular process then inflects at the midpoint of the coronoid process, forming a convex ventral margin ventral to the dental row. The ventral margin slightly tapers to the canine. The anterior margin of the coronoid process rises at an obtuse angle (∼125°) distal to the talonid of M3. A broad ridge originates on the buccal face of the dentary, ventral to the distal edge of the talonid of M3. The anterior fibers of the temporalis muscle would have inserted along this margin. The ridge rises to form the anterior margin of the coronoid process. The anterior edge of the masseteric fossa is deeply excavated but the ventral margin of the masseteric fossa is not as well-defined as the anterior portion.

The lower incisors are not preserved. The crown of the canine is worn. The buccal face of the canine is traced by multiple longitudinal ridges of enamel. The mesial root of P1 is very close to the distal edge of the root of the canine and it sweeps distally with the root canine. The distal root of P1 parallels the distally swept mesial root of P1. The crown of P1 is worn, but a portion of the mesiodistally short talonid is preserved. The crown of P1 is set at an oblique angle relative to the mesiodistal axis of P2.

Like P1, P2 has two roots. The roots of P2 are perpendicular to the alveolar margin. In buccal or lingual view, P2 is an asymmetrical triangle. There is a small, but pronounced paraconid on the mesial portion of the tooth. The paraconid is mesiodistally aligned with the protoconid. The shorter paraconid is linked to the protoconid by a short preprotocristid that rises steeply from the paraconid to the apex of the protoconid. The postprotocristid slopes to a mesiodistally short talonid.

There is no diastema between P2 and P3. Like P2, P3 is asymmetrical in buccal and lingual views with a mesiodistally short preprotocristid and mesiodistally long postprotocristid. The paraconid of P3 is small, but distinct and in a more lingual position than the protoconid. The protoconid of P3 is at least twice the height of the paraconid. The postprotocristid is buccolingually compressed and slopes to an indistinct talonid basin. A thin lingual cingulum connects the paraconid to the talonid.

The crown of P4 forms an equilateral triangle in lingual view, and, like each of the premolars, bears striated enamel. P4 is a stout tooth in occlusal view, its buccolingual width about half its mesiodistal length. The paraconid is a small but distinct cusp, with a postparacristid that forms a small notch with the longer preprotocristid. The paraconid is connected to a weak lingual cingulum that terminates at the base of the protoconid. A distal lingual cingulum begins just posterior to the apex of the protoconid. The cingulum forms the lingual margin of a very shallow talonid basin. The talonid of P4 has a small hypoconulid that connects to the hypoconid. The hypoconid is buccolingually compressed and rises to half the height of the protoconid. The hypoconid forms a distinct notch with the postprotocristid. The preprotocristid and postprotocristid are sectorial, and the apex of the protoconid curves slightly lingually and inclines very slightly distally.

A thin anterior keel on the buccal face of the M1 paraconid contacts the hypoconulid of P4. M1 is heavily worn on CGM 83750, but is well-preserved on DPC 17627. The protoconid is the tallest of the trigonid cusps, followed by the paraconid and the metaconid. The preprotocristid curves slightly mesially as it runs from the apex of the protoconid to the carnassial notch, where the preprotocristid and postparacristid meet at an angle of approximately 90°. The shearing surface created by the protoconid and paraconid is set at about an angle of 45° relative to the long axis of the dentary. The apex of the paraconid projects mesially. The metaconid is connate and connects with the base of the paraconid; its apex projects distally and is positioned slightly distal to the apex of the protoconid. A slight depression descends from the junction of the paraconid and metaconid, defining the base of each cusp. The distal face of the trigonid slopes at an obtuse angle (∼100°) to the talonid. The talonid basin is about one-third the total mesiodistal length of M1. The talonid basin is deep, closed buccally by the hypoconid, and closed lingually by the entocristid. The talonid cusps are crestiform. The entoconid is particularly indistinct, effectively submerged into the entocristid, which slopes distally from the base of the metaconid to meet the apex of the hypoconulid. The hypoconulid is a small cusp that is distinguished from the hypoconid by a weak intervening notch or inflection. The hypoconid is the most pronounced of the talonid cusps, and the cristid obliqua slopes ventrally and lingually toward the base of the protoconid from its apex.

M2 is mesiodistally longer, buccolingually broader, and taller than M1. The contact between M1 and M2 is small, with a gap formed between the M2 paraconid and the distal M1 talonid. M1 and M2 are similar in morphology, but differ in relative proportions. The metaconid is relatively low when compared with the paraconid, and the paraconid is relatively broader at its base, forming a stout cusp. The paraconid apex projects mesially and more lingually than the apex of the metaconid. The talonid basin of M2 makes up ∼40% of the mesiodistal length of the entire tooth, and the talonid basin is only about one-third the height of the protoconid. As on M1, the talonid cusps are crestiform and the entoconid is reduced to an undifferentiated entocristid. In buccal view, the angle formed between the alveolar margin and the distal edge of the protoconid is approximately 100°.

M3 is the tallest tooth in the dentary. It is subequal in mesiodistal length to M2 though more of its mesiodistal length is occupied by the trigonid. The talonid is ∼27% the mesiodistal length of the tooth, and ∼25% the height of the protoconid. The paraconid and protoconid on M3 are taller than the same cusps on M2 but the metaconids on M3 and M2 are almost the same height above the alveolar margin, making the M3 metaconid proportionally smaller compared to the rest of the trigonid. One distinctive feature of M3 is the morphology of the preprotocristid, which arcs mesially from the apex of the cusp to the deep carnassial notch. The apex of the protoconid projects distally like the metaconid, and both cusps arch somewhat distally toward the talonid basin. The talonid of M3 is relatively narrow when compared with the talonids of M1 and M2. The hypoconid is proportionally smaller, and the hypoconulid forms a more distinct distal point than it does on the more mesial molars.

Dental variation

Compared to CGM 83750, which was utilized for most of the description of the lower dentition, DPC 17627 is very similar, though the corpus of the dentary is more gracile than the dentary of CGM 83750. DPC 17627 preserves the two alveoli of P3 and the distal alveolus of P2. Like CGM 83750, there is no indication of a diastema between P2 and P3. This contrasts with DPC 11569A and DPC 11569B, two specimens that likely represent the right and left dentary of the same individual. Both specimens preserve a diastema between P2 and P3 that is half the mesiodistal length of P3. On DPC 11569A and DPC 11569B the paraconids on P2–P4 are very small compared to the paraconids on the same premolars on CGM 83750 and DPC 17627. Finally, the talonid basin of M3 is relatively smaller and narrower, with less clearly defined cusps than are found on the talonids of M3 on CGM 83750 and DPC 17627. We do not consider these differences significant enough to designate a new taxon based on the current sample. Future work in the L-41 collections will further explore morphological variation in the hyaenodont fauna found at this locality.

Body mass

The average mesiodistal length of the lower molars is 9.57 mm, which yields a body mass estimate of 5.24 kg using the equation of Morlo (1999), and 5.96 kg using the equation of Van Valkenburgh (1990). Using only M2 length yields an estimate of 6.10 kg, and only M3 length yields an estimate of 6.20 kg. Carnivorans with a comparable body mass include Vulpes vulpes (red fox) and Taxidea taxus (American badger).

Comparisons

Rostrum

Dissopsalis and Teratodon, both Miocene taxa (Barry, 1988; Savage, 1965), are the only demonstrable teratodontines (Solé et al., 2014; Rana et al., 2015) for which cranial morphology has been described. Dissopsalis carnifex is known from the middle to late Miocene of Asia; the holotype was reconstructed and described by Colbert (1933). The fragmentary specimen preserves much of the left and right maxillae as well as the frontal. Colbert (1933) reconstructed the zygomatic arches and much of the posterior skull. As in Brychotherium, the palatal margins from P1 to the anterior root of P3 of Dissopsalis are parallel. P3 is angled and its buccal margin follows the lateral flare of the maxilla. The infraorbital foramen is positioned dorsal to P3 in Dissopsalis, as it is in Brychotherium. Also like Brychotherium, the rostral profile of Dissopsalis, created by the gently sloping nasals and frontals, is low, and the sagittal crest emerges from the frontal caudal to subtle postorbital “peaks” rather than distinct processes. This differs from late Eocene European taxa like Hyaenodon and Cynohyaenodon, which have more pronounced postorbital processes and more deeply excavated lineae temporales (Lange-Badré, 1979).

Teratodon enigmae, from the early Miocene of East Africa (Savage, 1965), is also known from the rostrum. KNM-RU 14769 is a fragment of the left maxilla that contains the complete left canine and P1–P4. While the premolars are bulbous and very different from those of Brychotherium, the alveolus of the buccolingually compressed canine is similar in morphology to that preserved in DPC 11990. The P1 of Teratodon is also set close to, and slightly lingual of, the upper canine, as is suggested by the disposition of the canine and P1 alveoli of Brychotherium. The holotype of Teratodon preserves portions of the rostrum from the premaxilla back to the distal aspect of the palate; in this specimen, too, the anterior root of P1 is lingual to the canine alveolus.

Limited comparisons can also be made with Indohyaenodon from the early Eocene of India (Rana et al., 2015), Tritemnodon from the early Eocene of North America (Wortman, 1902), Paroxyaena from the late Eocene of Europe (Lavrov, 2007), Apterodon macrognathus from the early Oligocene of Egypt (Osborn, 1909; Szalay, 1967), and Pterodon dasyuroides from the late Eocene of Europe (Lange-Badré, 1979). Each of these taxa has a long, narrow rostrum with a broad nasal aperture. The nasals do not project prominently over the aperture and the nasals widen slightly as they approach the nasal-frontal suture. The frontal in each of these taxa does not exhibit a distinct postorbital process, but instead a postorbital peak (Pterodon) or subtle bump (Apterodon, Paroxyaena). The linea temporalis (= supraorbital boss in Rana et al. (2015)) is demarcated, but not deeply excavated (it is particularly subtle in Apterodon and Paroxyaena and more distinct in Indohyaenodon and Pterodon). Along the anterior orbital margin, the lacrimal has a particularly broad facial wing in Pterodon, Apterodon, and Paroxyaena, as it does in Brychotherium.

The fragmentary palatines of Brychotherium indicate that the internal choanae open distal to M3 and are delimited ventrally by a rugose palatine torus, comparable to the palatine construction of Paroxyaena and Tritemnodon. The internal choanae of Pterodon dasyuroides and Apterodon macrognathus open more caudally. In P. dasyuroides and A. macrognathus the palatines are fused along the midline posterior to the dentition. The palatines of Apterodon are fused for more of their length than those of P. dasyuroides, forming a palatine tube that extends to the basicranium. Colbert (1933) reconstructed the palatines of Dissopsalis as a long, fused palatine tube, likely based on comparisons to the skull of North American Hyaenodon, which also has a Apterodon-like tube (Mellett, 1977). The palatine morphology of Dissopsalis is, in fact, largely unknown.

Upper dentition

Like the teratodontines Dissopsalis and Anasinopa, Brychotherium has distinct paracones and metacones on the upper molars that are fused at their bases, but diverge well before their apices. In all three taxa, the paracone and metacone are buccolingually compressed, giving them an elliptical cross section, and the paracone is the smaller of the two cusps. The apex of the metacone projects perpendicular to the plane of the hard palate, while the smaller paracone projects mesially. This differs from hyainailourines (e.g., Pterodon dasyuroides, Kerberos, and “Pterodon” africanus), which fuse the paracone and metacone near the apices of the cusps. In hyainailourines, the paracone is the taller of the two cusps and would have been the leading piercing cusp during mastication. The paracone/metacone morphology of Teratodontinae is comparable to the arrangement of these cusps in Hyaenodon and Eurotherium which, like teratodontines, had metacones that were taller than paracones. However, in Hyaenodon and Eurotherium, the cusps are not as divergent; rather, in Hyaenodon, the paracone projects perpendicular to the plane of the hard palate, like the metacone, and is almost completely fused to the metacone, essentially forming part of the premetacrista (a state evident in unworn upper molars, such as the M1 of the subadult AMNH 75646). Apterodon also has distinct paracones and metacones as in teratodontines, but these cusps are more circular in cross-section and diverge closer to the buccal margin.

The molar paracones are more distinct in Brychotherium than in Dissopsalis, and the molar protocones of Brychotherium are more triangular and project lingually, rather than being elongate and shifted far mesially as in Dissopsalis. The trigon basins of Brychotherium more closely resemble those of Anasinopa, though the paraconules and metaconules of Brychotherium are relatively well-developed. The deep ectoflexus and large parastyle of the M1–2 of Brychotherium set the new genus apart from Anasinopa and Dissopsalis. Teratodon has small, tritubercular molars and a distinct ectoflexus on M2. The M1–2 metastyles of Brychotherium are mesiodistally longer than the parastyles whereas the para- and metastyles on the M2 of Teratodon are subequal in length. Indohyaenodon also has an elongate, arching M2 metastyle, like Brychotherium, but its protocone is more mesiodistally broad. The P4 paracone of Indohyaenodon is more buccolingually compressed than that of Brychotherium and its P4 protocone is less distinct. P4 in Kyawdawia, from the middle Eocene of Myanmar (Egi et al., 2005) is also more compressed buccolingually, and the metastyle is taller and more sectorial than that of Brychotherium. The M2 paracone and metacone of Kyawdawia are both buccolingually compressed and the mesiodistally elongate metacone is only slightly taller than the paracone. The M2 ectoflexus of Kyawdawia has the same degree of lingual curvature as that of Brychotherium and the same well-defined buccal cingulum. The M1 of middle Eocene Furodon shares many features with Brychotherium, including an elongate metastyle, buccolingually compressed and apically divergent paracones and metacones, a broad talon basin with a large paraconule, and a prominent and only slightly mesially oriented protocone. Furodon differs from Brychotherium by having a paracone that is slightly taller than the metacone. Glibzegdouia, from the same locality as Furodon, shares with Brychotherium a divergent paracone and metacone. Glibzegdouia differs from Brychotherium by having a metastyle is not as elongate as the metastyle of Brychotherium and by having a paracone that is closer to the height of the metacone. Koholia is from the late early Eocene and is the oldest Afro-Arabian hyaenodontan known from its upper dentition. The P4 protocone of Brychotherium is mesiodistally wider than the P4 protocone of Koholia, and the P4 metastyle of Brychotherium is mesiodistally short compared to the elongate metastyle of Koholia, which is the same mesiodistal length as the paracone. The M1 parastyle of Koholia is buccolingually wide, with space formed between the apex of the parastyle and the base of the paracone. The M1 parastyle in Brychotherium is buccolingually narrow and cingulum-like. Like in Brychotherium, the M1 paracone and metacone of Koholia have distinct apices, but the paracone and metacone are more fully fused in Koholia and the paracone is distinctly taller than the metacone. On the M1 of Brychotherium the metacone is slightly taller than the paracone.

Lower dentition

Solé et al. (2014) recovered Furodon as a hyainailourine with Pterodon and Akhnatenavus, with Furodon as the only member of this clade with a prominent metaconid. The phylogenetic analyses presented below do not support the Solé et al. (2014) hypothesis and instead resolve Furodon in a close relationship to metaconid-bearing teratodontines. Brychotherium and Furodon share many features, including their comparable size, relatively tall trigonids, low metaconids, and wide talonid basins with poorly defined talonid cusps. They primarily differ in the morphology of P4, the protoconid of which projects perpendicular to the alveolar margin in Brychotherium, and in occlusal view curves lingually. The lower molar paraconids also project more lingually in Brychotherium than they do in Furodon, the talonid basins are buccolingually more broad, and the molar hypoconulids are larger.

The oldest teratodontine in the analyses of Solé et al. (2014) and Rana et al. (2015) is Glibzegdouia, an early or middle Eocene taxon from Algeria. The M1 metaconid of Glibzegdouia is subequal in height to the paraconid, and the talonid basins of M1–2 are lined with distinct entoconids, hypoconulids, and hypoconids, unlike the poorly differentiated talonid cusps of Brychotherium. The talonid basin also occupies more than 50% of the total mesiodistal length of the molars of Glibzegdouia. The talonid is proportionally smaller and shorter in Brychotherium.

Masrasector aegypticum is a teratodontine from Quarry G (early Oligocene) in the Fayum succession (Simons & Gingerich, 1974). Like Brychotherium, M. aegypticum has broad talonid basins with indistinct talonid cusps. The smaller M. aegypticum is further differentiated from Brychotherium in having buccolingually broad premolars and a more tightly packed trigonid, with the paraconid apex perpendicular to the alveolar plane, rather than projecting lingually as it does in Brychotherium. The other possible teratodontine from the Fayum, Metasinopa, differs from Brychotherium in having a much deeper mandibular corpus, a more reduced talonid on M3, and much smaller M2-M3 metaconids, particularly on M3 where the metaconid barely connects to the paraconid. The reduction of the M3 metaconid is even more extreme in Dissopsalis and Anasinopa; the paraconid and protoconid of both taxa are divergent and form an obtuse carnassial notch between the preprotocristid and postparacristid and the metaconid is reduced to a very low or absent cusp (especially in Dissopsalis). Both Dissopsalis and Anasinopa have broad and lingually closed talonids on M1 and M2 and reduced talonids on M3, with the talonid on Dissopsalis reduced to a small distal projection from the M3 trigonid. These features contrast with the connate metaconid, well-developed talonid, and more acute carnassial notch found on the M3 of Brychotherium.

Lahimia, from the middle Paleocene of Morocco (Solé et al., 2009), and Boualitomus, from the early Eocene of Morocco (Gheerbrant et al., 2006), are both small hyaenodontans that Solé et al. (2014) and Rana et al. (2015) classified as part of Koholiinae. Like Brychotherium, they both retain lower molar metaconids that are slightly lower than the paraconids, and distinct talonids with indistinct talonid cusps. They differ from Brychotherium in their much smaller size, and by having molars that are mesiodistally subequal in length to each other. Brychotherium, like many hyaenodontans including Masrasector and Pterodon, but unlike Lahimia and Boualitomus, has molars that increase in length distally.

Indohyaenodon and Kyawdawia, part of Indohyaenodontinae in Solé et al. (2014) and Rana et al. (2015), have buccolingually broad talonids that are lingually closed by the entocristid, rather than lingually open as the talonid is in Brychotherium. Both taxa also have distinct buccal cingulids on the lower molars that originate from the anterior keel. In Brychotherium the anterior keel is small and does not connect to any cingulid. The metaconid is much lower than the paraconid on M3 in Indohyaenodon, Kyawdawia, and Brychotherium, but the metaconid is subequal to the height of the paraconid on M1 in the indohyaenodontines and is lower than the paraconid in Brychotherium. Indohyaenodon and Brychotherium also share relatively gracile dentaries that are only a little deeper dorsoventrally than the crown height of M3.

HYAINAILOURINAE Pilgrim, 1932

AKHNATENAVUS Holroyd, 1999

Type species: Akhnatenavus leptognathus Holroyd, 1999.

Emended generic diagnosis (modified from Holroyd, 1999): Differs from “Pterodon” africanus and “Pterodon” phiomensis by being smaller and by having more buccolingually compressed lower premolars; narrow, mesially shifted M1–2 protocones; and more buccolingually compressed and elongate M1–2 metastyles. Differs from Metapterodon by being smaller, retaining a talonid on M3 rather than having no talonid, having distinct M1–2 paracone and metacone cusps rather than being completely fused into a single cusp, having more lingually projecting M1–2 protocones rather than having small protocones that are close to the paracone bases, and having distinct M1–2 parastyles with spaces between the parastyle and the base of the paracone rather than the parastyles forming only a small projection. Differs from Pterodon dasyuroides by having smaller talonids on molars, a reduced M3 that does not project lingually as far as the M2 protocone, and narrower M1–2 protocones with preprotocrista and postprotocrista nearly parallel rather than protocones that are mesiodistally broad and triangular. Differs from Apterodon by having fused paracone and metacone cusps on M1–2, molar paraconids that are much shorter than the protoconids, mesiodistally short molar talonids, and M1–2 metastyles that are mesiodistally longer than the mesiodistal length of the paracone and metacone bases. Differs from Brychotherium by having M1–2 paracones that are taller than metacones and narrow, simple M1–2 protocones rather than triangular protocones with metaconule and paraconule cusps.

AKHNATENAVUS NEFERTITICYON Borths, Holroyd, and Seiffert, sp. nov.

(Figs. 10–16; Table 3 and 4)

urn:lsid:zoobank.org:act:19CBE178-447C-4182-9AED-C70280CD0673

Figure 10 Akhnatenavus nefertiticyon cranium CGM 83735.

Akhnatenavus nefertitcyon sp. nov., holotype CGM 83735, cranium with right canine, P3–M3 and alveolus of P2 and left P2, P4, M2; (A) right lateral view; (B) dorsal view; (C) left lateral view. Postmortem distortion mediolaterally crushed the specimen with the left dentition involuted. Specimen also preserves atlas (cervical vertebra 1) appressed to the basicranium and a proximal rib appressed to the right parietal.

Figure 11 Akhnatenavus nefertiticyon cranium CGM 83735 labeled.

Akhnatenavus nefertiticyon sp. nov., CGM 83735, cranium sketch (subscript 1) and digital model (subscript 2) with right canine, P3–M3 and left P2, P4, M2; (A) right lateral view; (B) dorsal view; (C) left lateral view. Dotted lines indicate uncertain sutures or boundaries. Unlabeled regions are fragmentary. Abbreviations: l., left; r., right; inf. orb. f., infraorbital foramen; mand. fossa, mandibular fossa; r. occ. condyle, right occipital condyle. Digital surface model is available on MorphoSource.

Figure 12 Akhnatenavus nefertiticyon CGM 83735 dentition detail.

Akhnatenavus nefertiticyon sp. nov., CGM 83735, photographs (subscript 1) and digital model images (subscript 2) of right P3–M3; (A) occlusal view. DPC 13518, left M1; (B) occlusal-lingual view.

Figure 13 Akhnatenavus nefertiticyon DPC 18242 cranium.

Akhnatenavus nefertiticyon sp. nov., DPC 18242, palate with left P2–M2 and P1 roots and right M1 and P1–P4 roots; (A) ventral view; (B) dorsal view; (C) left lateral view; (D) right lateral view. Postmortem distortion involuted the right maxilla and dorsoventrally compressed the cranium.

Figure 14 Akhnatenavus nefertiticyon DPC 18242 cranium labeled.

Akhnatenavus nefertiticyon sp. nov., DPC 18242, volume rendered images of palate with left P2–M2 and P1 roots and right M1 and P1–P4 roots; (A) ventral view; (B) dorsal view; (C) left lateral view; (D) right lateral view. Postmortem distortion involuted the right maxilla and dorsoventrally compressed the cranium. Digital surface model is available on MorphoSource.

Figure 15 Akhnatenavus nefertiticyon DPC 18242 and DPC 13518 dentition detail.

Akhnatenavus nefertiticyon sp. nov., detail from DPC 18242 of left P2–M2; photographs of dental specimen on the left (subscript 1) and digital model images on the right (subscript 2); (A) occlusal view; (B) buccal view; DPC 13518, isolated left M1 in (C) occlusal; (D) buccal, and (E) lingual view. Digital surface model is available on MorphoSource.

Figure 16 Akhnatenavus nefertiticyon DPC 7765 dentary.

Akhnatenavus nefertiticyon sp. nov., DPC 7765, fragmented right dentary with P2–M3; (A) occlusal view; (B) lingual view; (C) buccal view.

Table 3 Specimen measurements for the upper dentition of Akhnatenavus nefertiticyon.

Akhnatenavus nefertiticyon	Locus	Length	Width	Metastyle length	Paracone height	Metacone height	Para/Meta base length	
DPC 18242	P2	9.18	4.08	1.37	6.2	—	5.66	
(left side)	P3	10.09	5.22	2.18	6.3	—	6.58	
P4	10.94	8.54	2.96	6.25	—	6.09	
M1	14.08	11.08	6.6	5.57	4.63	7.21	
M2	15.3	12.8	6.9	7.27	6.07	7.44	
(right side)	M1	13.5	—	5.1	7.37	5.9	5.37	
CGM 83735	C	9.95	—	—	—	—	—	
(right side)	P3	10.98	4.44	2.34	—	—	6.03	
P4	11.98	9.15	3.65	—	—	5.71	
M1	13.28	9.78	5.33	—	—	5.72	
M2	15.14	13.5	7.28	—	—	7.04	
M3	2.49	—	—	—	—	—	
(left side)	P2	9.51	—	2.44	5.95	—	—	
P4	11.79	—	3.49	—	—	—	
M1	13.34	—	5.5	—	—	—	
M2	15.18	—	5.91	7.87	5.54	7.15	
DPC 13518	M1	14.73	9.5	6.37	5.41	4.41	6.96	
Tooth length		P2	P3	P4	M1	M2	M3	
N		2	2	3	5	3	1	
Mean (Std. Dev.)		9.35 (0.23)	10.54 (0.63)	11.57 (0.55)	13.79 (0.62)	15.21 (0.08)	2.49	
Notes:

Length, total mesiodistal length; Width, total buccolingual width; Metastyle length, mesiodistal length from base of the paracone (premolars) or metacone (molars); Paracone height, from alveolar margin to the apex; Metacone height, from alveolar margin to the apex; Para/Meta base length, mesiodistal length of base of both paracone and metacone; Std. Dev., standard deviation.

Summary statistics only include specimens with minimal wear.

Table 4 Specimen measurements of the lower dentition of Akhnatenavus nefertiticyon.

Akhnatenavus nefertiticyon	Locus	Max. length	Max. trigonid length	Max. talonid length	Max. trigonid width	Max. talonid width	Talonid height	Paraconid height	Protoconid height	
DPC 7765	P2	7.01	5.01	2.20	2.23	2.33	1.28	—	3.05	
P3	9.25	6.53	2.76	3.00	3.27	1.90	—	3.90	
P4	10.26	7.35	2.70	5.23	5.31	3.62	—	8.81	
M1	13.52	10.13	3.35	6.25	3.72	4.17	6.82	10.79	
M2	13.48	11.64	—	7.68	—	—	10.14	8.56	
M3	15.20	13.77	1.75	8.79	1.75	1.90	13.36	15.50	
DPC 11318	LC	5.66	—	—	—	—	—	—	—	
LP3	8.24	6.04	2.17	3.29	2.40	2.31	1.57	4.02	
LP4	8.54	6.65	1.93	4.22	2.73	2.13	—	∼4.76	
LM1	10.34	8.41	2.34	4.82	2.86	2.45	∼3.61	∼5.11	
LM2	14.43	11.04	3.41	6.03	2.87	3.76	7.76	10.74	
LM3	16.76	14.30	2.41	8.12	2.64	3.43	9.50	13.08	
DPC 11318	RP4	10.11	—	—	4.10	2.60	—	—	—	
RM1	10.31	—	—	—	—	—	—	3.4	
RM2	14.78	11.33	3.66	6.55	3.26	3.17	9.61	∼8.20	
RM3	15.62	13.67	1.38	7.95	2.80	—	11.53	13.53	
DPC 15250	C	5.44	—	—	—	—	—	—	—	
P2	7.98	5.05	2.82	2.44	2.23	1.94	—	4.15	
P3	10.14	7.36	2.90	3.53	2.94	2.25	—	5.88	
M1	10.79	—	—	—	—	—	—	—	
M2	15.46	11.95	3.63	7.45	4.00	3.63	9.30	12.78	
M3	∼15.11	15.38	∼0.78	8.35	—	—	12.85	17.37	
Tooth length		C	P2	P3	P4	M1	M2	M3		
N		2	2	3	3	4	4	3		
Mean (Std. Dev.)		5.55 (0.16)	7.50 (0.69)	9.21 (0.95)	9.64 (0.95)	11.24 (1.54)	14.54 (0.82)	15.86 (0.81)		
Notes:

Max. length, maximum mesiodistal length; Max. trigonid length, maximum mesiodistal length of trigonid; Max. talonid length, maximum mesiodistal length of talonid; Max. trigonid width, maximum buccolingual width of trigonid; Max. talonid width, maximum buccolingual talonid width; Talonid height, tallest point on talonid to alveolar margin; Paraconid height, apex of paraconid to alveolar margin; Protoconid height, maximum height from cristid obliqua to cusp apex. Std. Dev., standard deviation.

Measurements proceeded by “∼” indicate measurement taken from a heavily worn cusp.

Summary statistics only include specimens with minimal wear.

Etymology: Meaning “Nefertiti’s dog,” in reference to Nefertiti, the wife of Akhnaten, who is known from an exceptional cranial specimen.

Holotype: CGM 83735, cranium with canine, P2–M3.

Referred specimens: DPC 13518, M1; DPC 18242, palate with partial canine, alveoli for P1, and P2–M2; DPC 7765, dentary with P2–M3 (Described by Holroyd, 1999).

Type locality: Jebel Qatrani Locality 41 (L-41), Jebel Qatrani Formation, Fayum Depression, Egypt.

Age: late Eocene, latest Priabonian, ∼34 Ma (Seiffert, 2006)

Geographic distribution: Only known from Locality 41.

Diagnosis: Differs from Akhnatenavus leptognathus by being smaller and by having mesiodistally shorter diastemata between the premolars. Rostrum is inferred to be relatively shorter in A. nefertiticyon than is implied by the elongate dentary of A. leptognathus which has wider diastemata between adjacent premolars. P3 in A. nefertiticyon is distally inclined and relatively buccolingually wider than P3 in A. leptognathus. P4 in A. nefertiticyon is taller than P4 in A. leptognathus and P4 paraconid height is subequal to the mesiodistal length of the tooth rather than shorter than the mesiodistal length. M1–M3 have weak to absent ectocingulids and the talonid is reduced in A. nefertiticyon compared to the slight ectocingulids and better developed cusp-bearing talonids of A. leptognathus.

Description

Cranium

CGM 83735 is a complete cranium that was dorsoventrally crushed, with the left portion of the skull folded medially. The best preserved portions of the cranium are the right palate and dentition, the right squamosal, the nuchal crest, the posterior aspect of the sagittal crest, portions of the parietals, and the anterior portion of the frontal. A complete atlas is preserved on the right basicranial region and a rib fragment is preserved attached to the right parietal.

The atlas is preserved in dorsal view with deep facets for articulation with the occipital condyles visible in ventral view, along with the vertebral foramina and the articular facets for the axis. The left articular facet for the axis, visible in Figs. 10A and 11A, is broad and not as concave as the facets for the occipital condyles. The rugose dorsal tubercle is preserved along the dorsal arch, which is craniocaudally wide, approximating the dorsoventral diameter of the vertebral foramen. Fragments of the right transverse process suggest that the structure swept laterally before curving caudally. The proximal portion of a rib is preserved in caudal view with a deep costal groove along its body. It is mediolaterally broad, likely a first or second rib.

The nasals are relatively well-preserved, though the nasal aperture is not. Originating rostrally from a slight lateral expansion, the nasals narrow posteriorly and then expand laterally superior to the infraorbital foramen. The nasal-frontal suture reaches its posterior-most point at the midline of the cranium, and the suture trends at a 45-degree angle to the nasal-maxilla suture along the lateral border of the nasal. The facial portion of the maxilla is rostrally elongate. The maxilla is perforated by the infraorbital foramen, which is dorsal to the distal root of P3. The maxilla flares laterally posterior to the infraorbital foramen. The rugose maxillary portion of the maxilla-jugal suture indicates that the jugal formed the inferior margin of the orbit. The large lacrimal formed the anterior portion of the orbital rim. The orbital margin of the lacrimal has a dorsoventrally elongate lacrimal tubercle. The facial wing of the lacrimal is extensive, reaching from the anterior orbital margin, which is superior to the mesial root of M2, anteriorly to the distal root of P4. The frontal forms the superior margin of the orbit and preserves the deeply excised linea temporalis, which leads to the sagittal crest. The sagittal crest tracks the interparietal suture to the nuchal crest. The sutures of the bones that form the nuchal crest are not clear, though the occipital forms the lateral portions of the nuchal crest in Pterodon dasyuroides and Apterodon macrognathus, and the parietals form the medial portions of the crest near the sagittal crest. There are suggestions of a suture in these regions of the skull of Akhnatenavus, which are indicated in Fig. 11. The entire nuchal crest stands prominently above the cranial vault with the tallest portion of the sagittal crest bridging the space between the posterior aspect of the cranial portion of the parietal and the apex of the nuchal crest. The crest curves laterally, then recurves medially towards the foramen magnum, with the supraoccipital forming a clover-leaf-shape in caudal view. The exoccipital curves laterally from the ventral extension of the nuchal crest and supraoccipital. The occipital condyle is dorsoventrally elongate, though its relationship to the foramen magnum is difficult to interpret.

The left mandibular (or glenoid) fossa indicates that the mandibular condyle was mediolaterally wide and dorsoventrally short, with the width of the condyle about three times the height. A large postglenoid foramen is preserved posterior to the mandibular fossa. The right zygomatic process of the squamosal has an anteroposteriorly broad origin. The zygomatic process is dorsoventrally tall and robust near the mandibular condyle and it tapers as it trends rostrally. The fragmentary jugal would have formed a short contact with the inferior margin of the squamosal and continued the zygomatic arch to the inferior orbit and the contact of the jugal with the maxilla. The palatal portion of the maxilla is rugose and deeply embayed between the protocones. The maxilla-palatine suture is not easily traced, but the posterior margin of the palate is marked by a torus between the distal-most molars. The palatines extend posterior to the torus; though they are broken and the morphology of the internal choanae is obscured. The ventral face of the left palatine preserves a broken suture, evidence that the right and left palatine formed a suture posterior to the last upper molar, and diverged approximately mid-cranium.

Upper dentition

The upper right canine is preserved in CGM 83735. The root is ∼1.5 times the length of the crown and is widest just dorsal to the enamel-dentine junction. The enamel is longitudinally striated and the crown is round in cross-section rather than dorsoventrally depressed. The two roots of P1 are preserved in DPC 18242 and the tooth is crowded close to the mesial margin of the alveolus of the canine. P2 also has two roots and there is a small gap between P1 and P2. P2 is mesiodistally longer than P1. The buccolingual width of P2 is about half its mesiodistal length. The tooth has no parastyle or buccal cingulum. The paracone is recurved, and a convex curve is formed between the metastyle and the apex of the paracone. The metastyle is short, < 25% the mesiodistal length of the tooth, and buccolingually compressed. The mesiodistal length of P3 is subequal to the length of P2. A small gap separates the two premolars. A very small parastyle is present on P3. Like P2, the paracone sweeps distally and a buccolingually compressed metastyle protrudes from P3. The metastyle is offset from the mesiodistal axis of the paracone where the palate widens laterally, posterior to the infraorbital foramen. A slight lingual shelf is present, though it is not developed into a distinct protocone as is present on P4. P4 is mesiodistally longer than P3. The parastyle forms a prominent mesial cusp and the metastyle is an elongate, buccolingually compressed blade that is more than half the mesiodistal length of the base of the paracone. The buccolingually compressed paracone sweeps distally and the postparacrista forms a deep carnassial notch with the arching metastyle. The protocone cusp is buccolingually shorter than its mesiodistal width and is lined mesially by a cingulum-like preprotocrista and lined distally by the postprotocrista, which runs along the base of the paracone and metastyle.

The metastyle of P4 is braced buccally by the parastyle of M1 where the teeth are in contact. The parastyle of M1 is a broad cingulum-like shelf with a small and distinct apex, and there is no space between the apex of the parastyle and the base of the paracone. A thin buccal cingulum traces the base of the paracone and metastyle. The ectoflexus is very slightly distal to the metacone. The parastyle cusp is in line with the preparacrista, which is a buccolingually compressed, sectorial blade. The apex of the paracone is slightly taller than the metacone. The metacone is also buccolingually compressed but its base is mesiodistally elongate compared to the base of the paracone. The groove that distinguishes the fused paracone and metacone is visible in buccal and lingual view. The postmetacrista is thinner and mesiodistally longer than the preparacrista. The sectorial blade on the metacone forms a deep carnassial notch with the metastyle. The elongate metastyle is half the mesiodistal length of the tooth. From the carnassial notch the metastyle traces a convex, then concave, line to the distal-most point of the tooth. The protocone is broad with a distinct preprotocrista, a small paraconule, and no metaconule. The protocone is buccolingually as broad as it is mesiodistally wide and it has a strong mesial deviation relative to the paracone. The protocone is shifted so far mesially that it is lingual to the metastyle of P4. M1 contacts M2 lingual to the parastylar apex. M2 is mesiodistally longer than M1. Compared with M1, M2 has a larger parastyle that projects buccally. This gives M2 a deeper ectoflexus than M1. M2 is taller than M1 and the cleft between the paracone and metacone is more faint, though the apex of the paracone and metacone are both distinct, with the paracone slightly taller than the metacone. The protocone is buccolingually longer than its mesiodistal width, making the protocone on M2 appear slenderer than the protocone on M1. The protocones on both molars are subequal in width, though the protocone of M2 does not angle as far mesially as the protocone of M1. M3 contacts the lingual face of the metastyle of M2. The tooth is not buccolingually wide, only reaching the lingual-most point of the metacone of M2. The paracone is very small and the entire tooth is mesially angled. This does not appear to be postmortem distortion, but the natural orientation of the small terminal molar.

Lower dentition

The lower dentition is preserved in DPC 7765, though the specimen is fragmentary and interpretations of the masseteric fossa and coronoid process should be made carefully. A single alveolus for P1 is preserved. A diastema shorter than the mesiodistal length of P2 separates P1 from P2. P2 is a simple, short premolar mesiodistally longer than it is tall. The mesial margin of the protoconid of P2 is distally inclined as it slopes to the apex of the protoconid. The postprotocristid slightly undercuts the apex and gradually slopes to a short, simple talonid. A long diastema separates P2 from P3. P3 is a long, low tooth in buccal view with a mesiodistal length that is twice its height. The postprotocristid is buccolingually compressed into a slight ridge that slopes to a short talonid that bears a small cusp. P4 is subequal in mesiodistal length to P3, but the protoconid of P4 projects to a height subequal to the mesiodistal length of P4. A slight preprotocristid ridge forms at the mesial base of the protoconid. The postprotocristid of P4 is more compressed and bladelike than the preprotocristid. The postprotocristid steeply slopes from the acute apex of the protoconid to a distinct notch formed with the cristid obliqua. The hypoconid is prominent, though less than half the height of the protoconid. The hypoconid is buccolingually narrow and there is no evidence of other talonid cusps on P4. There is a slight lingual and buccal cingulum along the base of P4.

The lower molars are sectorial, carnassial teeth that are dominated by prominent paraconids and protoconids that frame deep carnassial notches. The paraconid is slightly shorter than the protoconid on each molar and the postparacristid and preprotocristid are compressed into shearing blades that are set at 30° angles relative to the mandibular ramus. There is no indication of a metaconid on any of the lower molars. Further, the lower molars do not dramatically increase in total mesiodistal length distally. The proportion of the tooth occupied by the trigonid does increase and the complexity of the talonid and the mesiodistal length of the talonid decreases. The talonid of M1 is 25% of the total length of the tooth. It has a prominent hypoconid, small hypoconulid, and is open lingually. The talonid of M2 is 23% of the total mesiodistal length of the tooth in DPC 15250 and DPC 11318 and it is also dominated by a buccolingually compressed hypoconid. The talonid of M3 is proportionally smaller than the talonid of M1 or M2. DPC 11318 preserves the small talonid, which is traced by a slight buccal cingulum and bears a very low hypoconid, essentially a tiny cusp, on the distal margin of the tooth.

Body mass

The mesiodistal length of the lower molars for CGM 83735 and DPC 18241 was measured between the postprotocrista of adjacent molars, which yielded an average molar length of 12.45 mm for DPC 18241 and 12.0 for CGM 83735. The mesiodistal length for the molars in the dentaries referred to Akhnatenavus nefertiticyon were measured directly from the specimens. The average molar length over all specimens is 13.18 mm, M2 average length is 14.1 mm, and M3 average mesiodistal length is 15.1 mm. Using Morlo (1999), the average body mass estimate is 21.6 kg and using Van Valkenburgh (1990) applied to M2 the average body mass is 18.9 kg, and applied to M3 the average body mass is 19.1 kg. The average estimated body mass across all methods is 19.2 kg, within the range of Gulo gulo (wolverine), Lynx lynx (Eurasian lynx), and Canis simensis (Ethiopian wolf) (body mass estimates from Finarelli & Flynn (2009)).

Comparisons

Akhnatenavus nefertiticyon shares many cranial and dental features with species placed in Hyainailourinae by Polly (1996), Solé et al. (2014) and Solé et al. (2015). Most notably, Pterodon dasyuroides, Hemipsalodon, and Megistotherium each also have distinctive, wedge-shaped nuchal crests that trend medially toward the foramen magnum, a feature also preserved in A. nefertiticyon, the oldest Afro-Arabian hyainailourine known from cranial material. Apterodon macrognathus also has this narrowed nuchal crest, which inclines caudally like the nuchal crest of Akhnatenavus nefertiticyon. This morphology contrasts with the broad nuchal crest of Hyaenodon and Eurotherium, which trends laterally toward the mastoid process (see Solé et al., 2015: Fig. 5). Another cranial feature shared by A. nefertiticyon and the other hyainailourines is the extensive facial wing of the lacrimal, which reaches from the anterior margin of the orbit to the distal root of P4. A rostrally extensive lacrimal is also shared with Brychotherium. More difficult to determine is the extent of palatine fusion in A. nefertiticyon, a feature that differs between Pterodon dasyuroides, which has palatines that diverge closer to the M3 than the mandibular fossa, and Megistotherium, which has palatines that are fused through the middle section of the cranium and only diverge close to basicranium. A. nefertiticyon does share the dorsoventrally deep zygomatic process of the squamosal with Megistotherium. The zygomatic arch is not completely preserved in Pterodon dasyuroides, but is preserved in Apterodon macrognathus and Kerberos langebadreae, both of which have dorsoventrally deep zygomatic arches that form robust attachment sites for the masseter muscle. Unlike Apterodon but like Kerberos, Akhnatenavus has distinct temporal lines leading to the origin of the sagittal crest. These deep lines, which indicate the anterior origin of the temporalis muscle, are comparable in depth to the lines preserved on the frontals of Pterodon dasyuroides. “Pterodon” africanus is known from a rostral specimen (SMNS 11575) that was sculpted into a complete cranium (Schlosser, 1911). The anterior portion of the skull preserves the broad nasal aperture and gently sloping nasals that are shared with Pterodon dasyuroides, Kerberos, and Akhnatenavus. It also preserves the slight postorbital eminence or subtle peak noted in Brychotherium. There is no indication of distinct, Hyaenodon-like postorbital processes in Akhnatenavus. Instead, the neurocranium is elongate along the anteroposterior axis with very slight waisting in the middle region of the skull as it is in Apterodon and Pterodon dasyuroides. The neurocranium of Hyaenodon and Eurotherium is hourglass-shaped in dorsal view, going from an expanded postorbital frontal to a narrow parietal around the sagittal crest, to a posteriorly expanded squamosal-parietal region.

Compared to the atlas preserved with CGM 83735, the atlas of Megistotherium (NHM M21902) has much broader transverse processes than A. nefertiticyon would have had, based on what is preserved along the fractured lateral margin of the A. nefertiticyon atlas. NHM M9472 is an atlas attributed to “Pterodon” africanus, which shares with Akhnatenavus transverse processes that are less robust than those of Megistotherium. The transverse processes of “Pterodon” africanus and Akhnatenavus sweep laterally, perpendicular to the vertebral foramen, and then caudally, rather than sweeping cranially before trending laterally as they do in Megistotherium. The dorsal arch of each hyainailourine atlas is craniocaudally long and thick compared to the much narrower dorsal arch of Hyaenodon (AMNH 8775; BSPG 1898 IV 32).

The upper dentition of Akhnatenavus shares with other hyainailourines a fused, buccolingually compressed upper molar paracone and metacone, with the paracone taller than the metacone. This arrangement differs from the partially fused paracone and metacone of Teratodontinae, which have taller metacones than paracones, and Hyaenodontinae, which have fused paracones and metacones, but the metacone is the taller of the two cusps and the paracone is fused to the mesial aspect of the metacone. Akhnatenavus differs from “Pterodon” africanus and “Pterodon” phiomensis primarily in size, but there are dental distinctions, particularly in the overall robusticity of the dentition of “Pterodon” africanus. P4 in “Pterodon” africanus (BMNM M21897) has a short, shelf-like protocone compared to the lingually projecting P4 protocone of Akhnatenavus. Like Akhnatenavus, the parastyle of M2 is better developed than the parastyle of M1 and the ectoflexus is slightly deeper, though not as lingually excavated as that of Akhnatenavus. The paracones of each preserved premolar and molar of “Pterodon” africanus sweep distally at stronger angles than the paracones of Akhnatenavus. “Pterodon” syrtos from Quarry M, an early Oligocene locality in the Fayum (Holroyd, 1999), differs from the other Afro-Arabian “Pterodon” species and Akhnatenavus by reducing the parastyles to thin mesial cingula.

Even greater differences in robusticity are evident between Akhnatenavus and the large early Miocene Hyainailouros napakensis (BMNH M19090) and Megistotherium. In the early Miocene taxa, the buccolingual width of the molars and mesiodistal length of the molars are closer to subequal. Like Akhnatenavus, the protocone of M1 in Hyainailouros crosses the transverse plane of the P4 metastyle and the protocone of M2 is narrower and more lingually oriented than the protocone of M1. The upper dentition of Megistotherium is not well-preserved, but the skull does preserve the alveoli, which indicate that small gaps were present between P1 and P2, and between P2 and P3. These diastemata are also present between these teeth in Akhnatenavus. In all hyainailourines discussed so far, including Pterodon dasyuroides, the upper molar paracone and metacone are distinguishable by a shallow buccal and lingual groove that runs between them. This differs from the condition in Metapterodon, a genus known from the early Oligocene (Holroyd, 1999) through the Miocene (Lewis & Morlo, 2010), which has completely fused the paracone and metacone and it is very difficult to distinguish the two cusps from one another. The metastyle, paracone, and metacone of Metapterodon are buccolingually compressed into delicate, blade-like structures. The protocone and parastyle are much more reduced than they are in Akhnatenavus, “Pterodon” africanus, and Pterodon dasyuroides.

Akhnatenavus differs from the European hyainailourines Pterodon dasyuroides and Kerberos in the reduction of M3, which, in the European taxa, retains a distinct protocone that projects as far lingually as the protocone of M2. Kerberos and P. dasyuroides also have prominent paracones on M3, and a sectorial parastyle. The distinction between M1 and M2 in P. dasyuroides is not as clear as it is in Akhnatenavus. In P. dasyuroides, the parastyle of M1 is a distinct cusp and it forms a slight ectoflexus and the protocones of the two molars are shifted mesially at similar angles. The contact between M1 and M2 in P. dasyuroides is small, with the distal-most point of the metastyle of M1 touching the mesial-most point of the parastyle of M2, where the buccolingually more broad parastyle of the M2 in Akhnatenavus buccally embraces the metastyle of M1.

The lower dentition of Akhnatenavus nefertiticyon differs from the lower dentition of Akhnatenavus leptognathus (Holroyd, 1999) in size, with A. nefertiticyon slightly smaller than the younger A. leptognathus. As noted by Holroyd (1999), the protoconid of P4 forms a more acute triangle in buccal view, and M3 is more reduced in A. nefertiticyon. Both taxa share large diastemata between P1 and P2 and P2 and P3, a single-rooted P1, short and low P3 and reduced talonids on all molars. These features set Akhnatenavus apart from other Fayum hyainailourines like “Pterodon” africanus and “P.” phiomensis, which have larger and more robust molars than Akhnatenavus, relatively taller P2 and P3, and no diastemata between any of the premolars. Solé et al. (2014) recovered Furodon in the same clade as Akhnatenavus and Pterodon. As noted by Solé et al. (2014), Furodon shares with Akhnatenavus a single-rooted P1 and with Akhnatenavus and Pterodon tall trigonids on the lower molars with shearing postparacristids and preprotocristids. Unlike Akhnatevnavus, Furodon retains prominent, though short, metaconids on all lower molars, and basined talonids with distinct hypoconids and hypoconulids.

Phylogenetic Results

Summary of the phylogenetic results

In each phylogenetic analysis (Figs. 17–19), Brychotherium is nested within Teratodontinae, either as the sister taxon of Dissopsalis (parsimony analysis and Bayesian analysis) or as the sister taxon of the clade that includes all teratodontines younger than L-41 (tip-dating analysis). Akhnatenavus nefertiticyon is the sister taxon of A. leptognathus using all phylogenetic methods, and Akhnatenavus is resolved within Hyainailourinae, though the specific topology of Hyainailourinae differs between methods. Each analysis also recovered Apterodontinae as part of the clade that includes both Teratodontinae and Hyainailourinae. The name proposed to discuss the clade that includes Apterodontinae, Teratodontinae, and Hyainailourinae is Hyainailouroidea.

Figure 17 Strict and Adams consensus trees.

(A) Strict consensus tree and (B) Adams consensus tree of 90 most parsimonious trees (1,061 steps, consistency index (CI) = 0.182, retention index (RI) = 0.620). P# corresponds to the node to the right of the label and are used in the discussion of clades and the biogeographic analyses. Bremer support values (range 1–10) right of relevant node. Bootstrap support values (range 50–100%) right of relevant node and italicized. Only clades supported by greater than 50% bootstrap support are labeled with bootstrap values. Major clades identified by this study are indicated by the round boxes with the clade name enclosed or overlapping the boundaries of the box.

Figure 18 Standard Bayesian “allcompat” tree.

B# correspond to the node to the left of the label and are used in the discussion and in the biogeographic analyses to reference the clade. Posterior Probabilities (PP, ×100) correspond to the node to the left of value. Strength of PP support summarized by branch and node color with very weak support shown in dark blue, weak support shown in light blue, moderate support shown in purple, and strong support shown in red. Major clades identified by this study are indicated by the round boxes with the clade name enclosed or overlapping the boundaries of the box.

Figure 19 Bayesian tip-dating “allcompat” tree.

T# correspond to Table S4 and biogeography results. Posterior probability (PP) shown in italics to the right or below relevant node. Divergence dates represent mean divergence date for clades and taxa. Branch color indicates absolute median rate (% character change/Ma) along branches. Rapidly evolving lineages are supported by warm colored branches (red to orange) and more slowly evolving branches are shown supported by blue to purple branches. The mean age recovered by the analysis is shown as a vertical line on the estimated range of possible ages for each terminal taxon. See Table S2 for sources of the age ranges used in the tip-dating analysis. Major clades identified by this study are indicated with round boxes with the clade name enclosed or overlapping the boundaries of the box. The continent of origin for each taxon is shown by the shading of the age range: black, Afro-Arabia; grey, Europe; checked, Asia or India; white, North America.

Each phylogenetic method also recovers Propterodon as the sister taxon of Hyaenodon and the clade that contains both genera is referred to as Hyaenodontinae. Hyaenodontinae is recovered by all methods as nested within a clade of European “proviverrines” that includes Matthodon, Eurotherium, Oxyaenoides and Cynohyaenodon. All analyses recover Oxyaenoides as the sister clade of Hyaenodontinae. The clade that includes the common ancestor of Cynohyaenodon and Hyaenodon is referred to in this analysis as Hyaenodontidae.

Several smaller clades are consistently recovered by each phylogenetic method. These include Limnocyoninae, a clade that contains Thinocyon, Limnocyon, and Prolimnocyon; a Galecyon clade that includes Galecyon and Boritia; a monophyletic Sinopa; and a monophyletic Arfia. The position of these clades within the topology is not consistent across all analyses. The fragility of their position is reflected in the very low support values at the nodes that unite these clades with the rest of Hyaenodonta.

This instability provides future areas of research as these unstable clades and unstable taxa, like Preregidens, Boritia, Parvagula and Koholia, are examined for additional anatomical information and continued fieldwork provides opportunities to discover more material referable to fragmentary and primarily dental taxa.

Maximum parsimony

The maximum parsimony analysis recovered 90 MPT each with a length of 1061 steps, a CI of 0.187, and a RI of 0.613. The character-taxon matrix contains 42% missing data with individual OTUs ranging from a minimum of 0% missing data (Hyaenodon horridus) to 84% missing data (Koholia) with a median of 40% of missing data across all OTUs. The parsimony results are shown in Fig. 17, summarized with a strict consensus tree with alphanumeric codes listed to the left of the relevant node. The Adams consensus tree of the 90 MPTs was also constructed to summarize the results of the analysis.

Eomaia was resolved by all MPTs as the outgroup relative to the rest of the OTUs included in the analysis. In all MPTs Maelestes is the sister taxon to the clade that contains all OTUs except Eomaia. Unexpectedly, the next-most deeply nested node is occupied by a clade that unites Tinerhodon and Altacreodus. Tinerhodon has been described as a hyaenodont (Gheerbrant et al., 2006; Solé et al., 2009) but it is most parsimoniously resolved as the sister to Altacreodus (= Cimolestes magnus), a common outgroup taxon for analyses of hyaenodont systematics (Barry, 1988; Polly, 1996; Rana et al., 2015). The clade Tinerhodon + Altacreodus is the sister clade to Hyaenodonta, defined in the parsimony analysis as Eoproviverra and all more deeply nested OTUs included in this analysis.

With regard to the new species described in this study, Brychotherium is recovered as the sister-taxon of Dissopsalis (P41). The Brychotherium + Dissopsalis clade is recovered as the sister-clade of Anasinopa + Furodon (P42). In all MPTs Metasinopa is the sister taxon of a monophyletic Teratodontinae. Both species of Masrasector are resolved within Teratodontinae, but the genus is reconstructed as paraphyletic.

The clade Teratodontinae + Metasinopa (P35) is part of a large polytomy (P34) that includes all Afro-Arabian hyaenodonts. Another clade resolved within this large polytomy is a clade of hyainailourines that includes Akhnatenavus nefertiticyon. The genus Akhnatenavus is monophyletic in all MPTs with strong support (Bremer = 5) and Akhnatenavus is resolved as part of a polytomy with two other Fayum hyaenodonts (“Pterodon” africanus and “Pterodon” phiomensis); two European hyainailourines (Kerberos and Paroxyaena); Hemipsalodon from North America; and a clade that contains the Miocene hyainailourines.

The polytomy that includes the hyainailourine clade that contains Akhnatenavus and the teratodontine clade that includes Brychotherium also includes Apterodontinae (P44), a clade with strong support (Bremer = 5); the “koholiines” (Lahimia, Boualitomus, Koholia and Metapterodon); and the “indohyaenodontines” (Indohyaenodon, Paratritemnodon, and Kyawdawia). Neither Koholiinae nor Indohyaenodontinae are recovered by all MPTs as part of clades. Also included in the large polytomy (P34) that contains hyainailourines (P47), apterodontines (P44), teratodontines (P35), “koholiines,” and “indohyaenodontines” are North American taxa with relatively sectorial dentitions (Tritemnodon, Pyrocyon and Gazinocyon).

Sharing a common node with P34 are the Galecyon clade (P29) and Limnocyoninae (P26). Galecyon is resolved as a paraphyletic genus by parsimony analysis, with Galecyon morloi recovered as the sister taxon of Boritia and Parvagula. Limnocyoninae (P26) contains the taxa traditionally placed in this subfamily, but Thinocyon and Limnocyon were unexpectedly resolved as consecutive sister taxa to Prolimnocyon, the only genus in this sample of limnocyonines that retains M3.

The sister taxon to the clade that includes Galecyon, Limnocyoninae, Hyainailourinae, Apterodontinae, and Teratodontinae (P25) is Prototomus minimus (P24), a taxon from Europe and just younger than Paleocene/Eocene boundary. Prototomus is not a generic clade in the parsimony analysis, but rather successive sister taxa, with Prototomus phobos resolved as the sister taxon (P23) to the larger clade that includes P. minimus. Sinopa is a generic clade (P22), and is the sister group to the clade that includes Prototomus (P23). Arfia is also monophyletic and all MPTs resolve the Arfia clade (P19) as the earliest branching lineage in the clade (P18) that contains all Afro-Arabian hyaenodonts (P34), the Galecyon clade (P29), Limnocyoninae (P26), Prototomus, Sinopa (P22), and Arfia (P19).

The clade resolved with Arfia as the earliest branching clade (P18) is part of a trichotomy (P3), along with Lesmesodon and a large, well-resolved clade (P4) composed of many European “proviverrines,” Propterodon, and Hyaenodon. All MPTs resolve Matthodon as the sister taxon to Oxyaenoides + Hyaenodontinae with strong support (Bremer = 6). Oxyaenoides also has strong support (Bremer = 6) as the sister taxon to Hyaenodontinae (P15). Hyaenodontinae is composed of Propterodon + Hyaenodon and the clade has strong Bremer (9) support. Hyaenodon is a robust clade (Bremer = 9, Bootstrap = 78), though the precise relationships among H. horridus, H. minor, and H. neimongoliensis are unresolved by parsimony analysis.

Bayesian phylogenetic inference

The “allcompat” (majority-rule plus compatible groups) topology recovered through standard Bayesian analysis is shown in Fig. 18 with PP indicated to the right of the relevant node and the alphanumeric code used in this discussion to the left of the relevant node. Nodes with strong PP support are drawn in red, moderately supported nodes are drawn in purple, weakly supported nodes are drawn in light blue, and extremely weak nodes are drawn in dark blue.

The general structure of the Bayesian “allcompat” tree is similar to the topology recovered by the strict consensus and Adams consensus trees of the 90 MPTs. Like the parsimony analysis, Tinerhodon + Altacreodus (B76; PP = 0.81) has strong support (PP = 0.85) as the sister clade of Hyaenodonta (B1; PP = 0.97) and Eoproviverra is the sister taxon of all other hyaenodonts. There is very weak PP support along the spine of the topology, though the “allcompat” tree reconstructs a highly probable arrangement of the taxa, given the character-taxon matrix and models of evolution input for the analysis. Hyainailourinae (B61), Apterodontinae (B57), and Teratodontinae (B49), are part of the clade that also contains the “indohyaenodontines,” another result consistent with parsimony analysis. The Galecyon clade (B34), Limnocyoninae (B30), and the clade that includes Sinopa (B29) are part of the stem lineage supporting the clade that includes “indohyaenodontines” and Hyainailouroidea. The large clade that includes Hyaenodontinae (B9) and several “proviverrines” contains the same taxa as P4 except Proviverra, a taxon that was resolved by Bayesian inference at a more basal branching position. In the Bayesian analysis Proviverra is the sister taxon to the clade that includes Arfia and all more deeply nested hyaenodonts. In the Bayesian analysis Arfia diverges from other Hyaenodonta more basally than it does in the parsimony analysis. Here, Arfia is the sister clade to the large clade that includes Hyaenodontidae and Hyainailourinae. Arfia is weakly supported in this position (B5; PP = 0.23).

The taxa described in this study, Akhnatenavus and Brychotherium, are recovered in the same larger clades they were placed in using parsimony analysis. Akhnatenavus is resolved among the hyainailourines and Brychotherium is placed among the teratodontines, though the structure of Hyainailourinae and Teratodontinae differs between the parsimony and standard Bayesian analysis. In the Bayesian “allcompat” tree, Akhnatenavus (B68; PP = 0.97) is very weakly placed (B66; PP = 0.13) as the sister clade of Hemipsalodon + “Pterodon” africanus (B67; PP = 0.24). Like in the parsimony analysis, the Miocene hyainailourines are supported as a clade (B71; PP = 0.96). In the Bayesian analysis, the weakly supported clade comprised of Paroxyaena and “Pterodon” phiomensis (B70; PP = 0.19) is the sister clade to the Miocene hyainailourines. Brychotherium is again recovered as the sister taxon of Dissopsalis, though this relationship is relatively weak (B54; PP = 0.48). Brychotherium + Dissopsalis is the sister clade of Furodon + Anasinopa (B55; PP = 0.70). In the Bayesian analysis, Masrasector is resolved as monophyletic, though with very weak support for the clade (B51; PP = 0.31), and Metasinopa is again the sister taxon of Teratodontinae.

Like in the parsimony analysis, the standard Bayesian analysis does not recover a monophyletic Indohyaenodontinae nor a monophyletic Koholiinae. Instead the “indohyaenodontines” (Paratritemnodon, Kyawdawia, and Indohyaenodon) are successive sister taxa to Hyainailouroidea (B47: Teratodontinae + Apterodontinae + Hyainailourinae). As in the Adams consensus tree, Lahimia + Boualitomus is part of the larger clade that includes Hyainailouroidea and the “indohyaenodontines.” However, the sister taxon of Lahimia + Boualitomus in the Adams consensus tree is Tritemnodon. Tritemnodon remains in phylogenetic proximity to Lahimia + Boualitomus in the standard Bayesian analysis, but the very weakly supported sister taxon of Lahimia + Boualitomus is Preregidens (B41; PP = 0.25) instead. The fragmentary Koholia and Metapterodon are resolved as sister taxa (B63; PP = 0.49) nested within Hyainailourinae, rather than as part of a clade that also contains Lahimia and Boualitomus.

Tip-dating Bayesian inference

The “allcompat” topology recovered using the Bayesian tip-dating method is shown in Fig. 19 with branches color-coded to indicate percent of change per million years (% change/Ma) that occurred along each branch. Branches with faster rates of change are orange to red and branches with slower rates of change are colored dark purple to blue. The color scale appears exponential, largely because the most rapidly changing branches (i.e., T18) are very different than the overall clock rate. The alphanumeric code to the left of the relevant node corresponds with this discussion and PP are italicized and placed to the right or below the relevant node. Table S3 contains the statistics generated by the tip-dating analysis most relevant to this study: median and mean age, median and mean relative rates, PP, and the 95% confidence interval for the youngest and oldest age estimates for each node. Each taxon is shown with the full estimated stratigraphic age range drawn from the literature (Table S2) and the mean age for each taxon, as recovered with tip-dating Bayesian analysis, is indicated in Fig. 19 as a black vertical line within the estimated age bar for the taxon. Divergence dates discussed below are mean age estimates.

In order to perform the tip-dating analysis and place a prior on the age of Hyaenodonta, that clade had to be explicitly defined. Because Tinerhodon was recovered as the sister taxon of Altacreodus using both parsimony and standard Bayesian analysis, Tinerhodon was excluded from Hyaenodonta in the tip-dating analysis. Tinerhodon, Altacreodus, and Maelestes were recovered as the sister group of Hyaenodonta (T75; PP = 0.80). Tinerhodon and Altacreodus are estimated to have split from each other during the Late Cretaceous (∼73 Ma). Hyaenodonta is estimated to have originated in the Late Cretaceous at ∼73 Ma.

As in the parsimony and standard Bayesian analyses, the monophyly of Akhnatenavus is strongly supported (T67; PP = 0.94), and the divergence between the two species is reconstructed as having occurred ∼38 Ma (late Eocene). Akhnatenavus is nested within Hyainailourinae, a clade that has a slightly different topology from Hyainailourinae in the standard Bayesian analysis. For example, Paroxyaena is resolved as the sister taxon to a clade (T69; PP = 0.32) that originated ∼43 Ma that includes all younger Afro-Arabian hyainailourines and Hyainailouros. In the tip-dating analysis Koholia, an early Eocene taxon, is resolved as the sister taxon of Tritemnodon (T53; PP = 0.53), a clade placed along the relatively rapidly evolving (2.38% change/Ma) stem of Hyainailouridae. This differs from the standard Bayesian topology, which resolved Koholia as nested within Hyainailourinae.

Brychotherium is nested within a weakly supported Teratodontinae (T44; PP = 0.39), a clade that likely originated in the Paleocene (∼60 Ma). In the parsimony analysis and standard Bayesian analysis, Brychotherium is the sister taxon of early Miocene Dissopsalis. In the tip-dating analysis Dissopsalis is the sister taxon of Anasinopa, another early Miocene teratodontine, and Brychotherium is resolved as the sister taxon (T46; PP = 0.67) to a clade that includes all younger teratodontines (T47; PP = 0.57). Masrasector is monophyletic and the sister clade of Teratodon.

The branch that supports the clade that includes Teratodontinae, Hyainailourinae, Apterodontinae, and the “indohyaenodontines” (T43; PP = 0.33) as Hyainailouroidea is a rapidly evolving branch (3.32% change/Ma) that has a Late Cretaceous origin (∼68.6 Ma). In the tip-dating analysis Lahimia + Boualitomus (T42; PP = 0.98) is robustly supported as a rapidly evolving clade (2.74% change/Ma), and Lahimia + Boualitomus is very weakly supported as the sister clade of Hyainailouroidea.

Many of the clades recovered by standard Bayesian analysis are also recovered by tip-dating analysis, including a Galecyon clade that includes Boritia (T37; PP = 0.36). In the tip-dating analysis Galecyon + Boritia is the sister clade of Gazinocyon + Pyrocyon (T36; PP = 0.22). This weakly supported clade (T35; PP = 0.06) is along the stem of Hyainailouroidea as the sister clade to (Lahimia + Boualitomus) + Hyainailouroidea (T41). In the tip-dating analysis, the Sinopa clade (T10; PP = 0.87) and Limnocyoninae (T4; PP = 42) are not along the Hyainailouroidea stem as they were in the standard Bayesian analysis. Instead, they form an extremely weakly supported clade (T2; PP = 0.04) with some European proviverrines (T13; PP = 0.26) that diverged from the all other hyaenodont lineages during the Late Cretaceous (∼72 Ma).

Both the parsimony analysis and standard Bayesian analysis resolved Eoproviverra as the sister taxon to all other hyaenodonts, but in the tip-dating analysis, Eoproviverra is deeply nested as the sister taxon of Proviverra (T17; PP = 0.44) within the early-diverging clade (T2) that includes Sinopa, Arfia, and Limnocyoninae. The clade T2 is sister to the very weakly supported clade that includes both Hyaenodontinae and Hyainailourinae (T18; PP = 0.09). Hyaenodontinae + Hyainailourinae (T18) arose during the Late Cretaceous (∼72 Ma) and arise from the most rapidly evolving lineage in the analysis (7.2% change/Ma).

The clade that includes Hyaenodontidae (T19; PP = 0.15) is sister to the clade that includes Galecyon, Lahimia + Boualitomus, and Hyainailourinae. Preregidens is the earliest diverging lineage in the clade that includes Hyaenodontidae, a position comparable to its place in the parsimony analysis. Within Hyaenodontidae, the branches that support Matthodon and Oxyaenoides as successive sister taxa to the clade that includes Hyaenodontinae imply accelerated rates of evolutionary change (T26; PP = 54; 3.55% change/Ma: T27; PP = 0.97; 3.80% change/Ma) occurred before the origin of Hyaenodontinae (T29; PP = 1.0; 2.66% change/Ma).

Biogeographic Results

Summary of the biogeographic analyses

The three methods of biogeographic reconstruction applied to each consensus topology recover Europe as the place of origin of Hyaenodonta. All topologies and biogeographic methods also recover an Asian origin for Hyaenodontinae, revealing a dispersal of the hyaenodontine ancestor from Europe to Asia. All methods resolve Afro-Arabia as the place of origin of Hyainailourinae (the clade that includes Akhnatenavus), Teratodontinae (the clade that includes Brychotherium), and Apterodontinae. Within Hyainailourinae, all non-Afro-Arabian taxa (i.e., Orienspterodon, Hemipsalodon) are resolved as evidence of independent dispersal events from Afro-Arabia to the northern continents. The biogeographic origin of Hyainailouroidea differs across the topologies. The origins of Limnocyoninae, the Galecyon clade, Arfia, and the Sinopa clade also differ, with each clade having either a European or North American origin depending on the topology under examination.

Biogeographic reconstructions on the maximum parsimony topology

Three methods of biogeographic analysis were applied to the maximum parsimony strict consensus tree: PO of geographic areas, LO of geographic areas, and BBM. The biogeographic reconstruction for each node in the strict consensus parsimony topology is presented in Table S4 and the continental area designated for each OTU and the results of the BBM analysis are shown in Fig. 20. The pie chart over each node represents the probability that the clade originated from each continental area.

Figure 20 BBM biogeographic analysis of strict consensus tree.

Results from Bayesian Binary MCMC (BBM) biogeographic analysis performed on the strict consensus tree based on maximum parsimony analysis. Colored portion of the circle corresponds to the likelihood the node originated from the corresponding continental area. P# corresponds to clade rows in Table S5 where the reconstructed biogeographic origin for each clade is listed using parsimony optimization (PO), and percent likelihood for each area using likelihood optimization (LO) and BBM. Green, Afro-Arabia; purple, Asia; red, Europe; blue, North America; grey (on map), continents without hyaenodonts.

The root node of Hyaenodonta (P1) is unambiguously reconstructed (PO) with Europe as the area of origin for the entire clade. A European origin for Hyaenodonta is strongly supported by LO (LO = 93%) and BBM analysis (BBM = 100%). Hyaenodontidae is (P8) also unambiguously reconstructed with an origin in Europe (LO = 100%; BBM = 100%). Hyaenodontinae (P15) is unambiguously Asian in origin (LO = 98%; BBM = 97%) implying a dispersal of hyaenodontids from Europe to Asia. Hyaenodon (P16) unambiguously originated in Asia (LO = 97%; BBM = 97%) before different Hyaenodon lineages dispersed to Europe and North America.

The clade P18 is the sister clade to the large clade that includes Hyaenodon (P4) and it is unambiguously reconstructed with a North American origin (LO = 75%; BBM = 87.64%). Sinopa (P22), Arfia (P19), Limnocyoninae (P26), and the Galecyon clade (P29) are each unambiguously North American in origin, despite each clade containing at least one non-North American OTU.

The biggest conflict in biogeographic reconstructions occurs at P34, which unites “indohyaenodontines,” “koholiines,” Teratodontinae, Apterodontinae, and Hyainailourinae. PO unambiguously resolves this node as Afro-Arabian in origin, a reconstruction robustly supported by LO (LO = 100%). But, BBM resolves this node as robustly North American in origin (BBM = 99%). The parsimony and likelihood reconstruction of P34 supports a single dispersal event from North America to Afro-Arabia. The BBM reconstruction is more complicated, implying multiple possible dispersals are likely to have occurred between North America, Afro-Arabia, Asia, and Europe depending on the MPT under scrutiny.

Less complicated are the origins of the constituent clades of P34. Regardless of where Teratodontinae (P35) dispersed from, once the clade is established, the radiation is entirely endemic to Afro-Arabia. Apterodontinae (P44) is also unambiguously Afro-Arabian in origin (LO = 99%; BBM = 100%), including the common node of A. gaudryi and A. langebadreae (P46), implying the ancestor of A. gaudryi dispersed from Afro-Arabia to Europe. The clade of hyainailourines that contains Akhnatenavus (P47) is also unambiguously Afro-Arabian in origin though it contains North American (Hemipsalodon) and European (Kerberos, Paroxyaena, and Hyainailouros) OTUs. Like the dispersal of A. gaudryi, this result suggests multiple dispersals occurred from Afro-Arabia to Europe during the evolution of Hyainailourinae.

Biogeographic reconstructions on the “allcompat” Bayesian topology

The results of each method for reconstructing biogeography on the Bayesian “allcompat” topology (PO, LO, and BBM) are listed in Table S5 and the results of BBM are shown in Fig. 21 imposed on the standard Bayesian topology.

Figure 21 BBM biogeographic analysis of standard Bayesian tree.

Results from Bayesian Binary MCMC (BBM) biogeographic analysis performed on the standard Bayesian “allcompat” consensus tree. Colored portion of the circle corresponds to the likelihood the node originated from the corresponding continental area. B# corresponds to clade rows in Table S6 where the reconstructed biogeographic origin for each clade is listed using parsimony optimization (PO), and percent likelihood for each area using likelihood optimization (LO) and BBM. Green, Afro-Arabia; purple, Asia; red, Europe; blue, North America; grey (on map), continents without hyaenodonts.

The root node of Hyaenodonta (B1) is unambiguously resolved as European using PO and is strongly supported as European by the other methods (LO = 98%; BBM = 100%). Unlike the parsimony analysis, which resolved the origins of Arfia as unambiguously North American, in the Bayesian analysis Arfia (B6) is placed at a node closer to the origin of Hyaenodonta and is ambiguously North American (LO = 98%; BBM = 36%) or European (LO = 2%; BBM = 63%) in origin. The common node shared by Arfia and all other more deeply nested hyaenodonts is unambiguously European (B5; LO = 98%; BBM = 100%). As in the biogeographic analysis of the parsimony topology, the node (B8) shared by the clades that contain Hyainailouroidea and Hyaenodontidae originated in Europe (LO = 98%; BBM = 100%). The Bayesian topology also resolves Hyaenodontidae as unambiguously European (B10; LO = 100%; BBM = 100%) and Hyaenodontinae as unambiguously Asian in origin (B19; LO = 96%; BBM = 98%), implying dispersal from Europe to Asia before the origin of Hyaenodon.

Differences between the parsimony topology and standard Bayesian topology, including the differing positions of Parvagula (B24 versus P33), Prototomus minimus (B27 versus P24), and Boritia (B34 versus P33), affected the biogeographic scenario surrounding the origins of Hyainailouroidea. In the parsimony topology, the nodes along the stem supporting Hyainailouroidea (P18, P21, P23, P24, P25) were unambiguously North American. In the Bayesian topology, the nodes through the comparable portion of the tree along the stem supporting Hyainailouroidea are European (B24; LO = 95%; BBM = 100%) or ambiguously North American or European (B25; B33; B38) in origin. Preregidens is the sister taxon (B41) of Lahimia + Boualitomus and the origin of this clade is ambiguous as Afro-Arabian (LO = 23%; BBM = 18%), European (LO = 40%; BBM = 73%), or North American (LO = 28%; BBM = 8%), though a European dispersal of the ancestors of Lahimia + Boualitomus is most likely according to both LO and BBM. The clade B41 is the sister clade to the Tritemnodon and Hyainailouroidea (B43) clade and B40 is their common node, which has an ambiguous origin in Europe or North America, though the likelihood methods support a North American origin for B40 (LO = 51%; BBM = 61%).

In the Bayesian topology, the “indohyaenodontines” are successive sister taxa to Hyainailourinae, with Indohyaenodon occupying the earliest branching position as the sister taxon to the clade that includes Kyawdawia, Paratritemnodon and Hyainailouroidea. Paratritemnodon is the sister taxon to Hyainailouroidea (B47). The common node of each branch (B44, B45, B46) is unambiguously Asian in origin, suggesting the ancestors of Hyainailouroidea dispersed from North America to Asia before dispersing to Afro-Arabia, a dispersal represented at B47, the common node of Teratodontinae and Hyainailouridae that is unambiguously Afro-Arabian (LO = 88%; BBM = 84%).

Every node within the clade Hyainailouroidea is unambiguously Afro-Arabian in origin, including the node that supports Akhnatenavus (B68; LO = 100%; BBM = 100%) within Hyainailourinae and the node that supports Brychotherium + Dissopsalis (B54; LO = 99%; BBM = 100%) within Teratodontinae. This result suggests the ancestor of Orienspterodon dispersed from Afro-Arabia to Asia, the ancestor of Hemipsalodon dispersed from Afro-Arabia to North America, and the ancestors of A. gaudryi, Paroxyaena, Kerberos, and Pterodon dasyuroides dispersed from Afro-Arabia to Europe.

Biogeographic reconstructions on the tip-dating topology

One of the advantages of tip-dating Bayesian analysis is the simultaneous estimation of topology, evolutionary rates, and divergence times given the character-taxon matrix and specific evolutionary models. When PO, LO, and BBM biogeographic methods are applied to the tree, it is possible to not only generate hypotheses surrounding the order of dispersal events, but it is also possible to hypothesize when those dispersals may have taken place. The results of each method are listed in Table S6 and the results of BBM analysis are shown over the corresponding nodes in Fig. 22.

Figure 22 BBM biogeographic applied to Bayesian tip-dating tree.

Results from Bayesian Binary MCMC (BBM) biogeographic analysis performed on the Bayesian tip-dating “allcompat” consensus tree. Colored portion of the circle corresponds to the likelihood the node originated from the corresponding continental area. T# corresponds to clade rows in Table S7 where the reconstructed biogeographic origin for each clade is listed using parsimony optimization (PO), and percent likelihood for each area using likelihood optimization (LO) and BBM. Green, Afro-Arabia; purple, Asia; red, Europe; blue, North America; grey (on map), continents without hyaenodonts. Red vertical dashed lines indicate dispersal intervals discussed by Gheerbrant & Rage (2006).

In the tip-dating analysis topology the origin of Hyaenodonta is unambiguously European (LO = 79%; BBM = 96%), the same result recovered in parsimony analysis and standard Bayesian analysis. The novel clade recovered by tip-dating analysis (T2) that includes a clade of proviverrines, the Sinopa clade, Limnocyoninae, and Arfia unambiguously originated in Europe (LO = 80%; BBM = 99%). In the tip-dating analysis, European Arfia gingerichi is the sister taxon to the North American species included in the analysis. The origin of Arfia is ambiguously resolved as European or North American, though the likelihood methods resolve Europe as the most likely continent of origin for the clade (LO = 53%; BBM = 77%) with dispersal from Europe to North America occurring in the latest Paleocene. The origins of Limnocyoninae (T7) are also ambiguous as Asian (LO = 33%; BBM = 17%), European (LO = 11%; BBM = 27%), or, the most likely continent of origin using the likelihood methods, North America (LO = 54%; BBM = 55%). The common ancestor of Limnocyoninae and Arfia, represented at node T4 is most likely European in origin (LO = 56%; BBM = 76%) though this node is ambiguously resolved with North America as an alternative center of origin (LO = 39%; BBM = 23%). In the tip-dating analysis, Prototomus minimus, a European taxon that is the same age as Arfia gingerichi and Galecyon morloi (Smith & Smith, 2001), is recovered as the sister taxon of the clade (T11) that includes Prototomus phobos and Sinopa. The presence of a European taxon near the origin of the Sinopa clade leaves their common node (T10) ambiguous as European or North American, though Europe is recovered as the most likely continent of origin for the clade (LO = 59%; BBM = 96%). This result indicates the common ancestor of P. phobos and Sinopa dispersed from Europe to North America in the middle Paleocene, then the ancestor of Sinopa jilinia dispersed from North America to Asia during the early Eocene.

The rapidly evolving branch T18, which unites the clade that includes Hyaenodontidae and the clade that includes Hyainailouridae, is European in origin (LO = 79%; BBM = 99%). The precise topology of Hyaenodontidae varies for each phylogenetic analysis, the biogeographic result is consistent across topologies with Hyaenodontidae (T20) unambiguously resolved with a European origin (LO = 100%; BBM = 100%) and Hyaenodontinae unambiguously reconstructed as Asian in origin (LO = 95%; BBM = 98%). The dispersal of the ancestor of Hyaenodontinae from Europe to Asia most likely occurred during the early Eocene, and the ancestors of North American Hyaenodon horridus and European Hyaenodon minor most likely dispersed from Asia to North America and from Asia to Europe during the late middle to early late Eocene (late Bartonian to early Priabonian).

The tip-dating analysis recovered a monophyletic Galecyon (T38) with European Galecyon morloi placed as the sister taxon to a clade that contains three North American Galecyon species. Galecyon has an unambiguous European origin (LO = 56%; BBM = 98%). Boritia, the sister taxon (T37) of Galecyon, is European as well, a result that supports the common node of Galecyon, Boritia, Gazinocyon, and Pyrocyon (T35) as unambiguously European in origin (LO = 55%; BBM = 89%). Within this clade, multiple dispersals from Europe to North America likely occurred during the Paleocene, a dispersal trend also found in the dispersal patters of Limnocyoninae, Prototomus, and Arfia.

The Galecyon clade (T35) is sister to the clade that includes all Afro-Arabian hyaenodonts (T41). The common node of the Galecyon clade and the Afro-Arabian hyaenodont clade (T41) is T34 and it is unambiguously European in origin (LO = 64%; BBM = 92%). Lahimia + Boualitomus (T42) is the sister clade of Hyainailouroidea (T43) and their shared node (T41) is unambiguously Afro-Arabian in origin (LO = 71%; BBM = 98%), suggesting the common ancestor of all Afro-Arabian hyaenodonts dispersed from Europe to Afro-Arabia during the Late Cretaceous. Teratodontinae (T44) is the sister clade of the clade that includes “indohyaenodontines,” Apterodontinae, and Hyainailourinae. In this analysis Teratodontinae is an entirely Afro-Arabian radiation of hyaenodonts that originated during the Paleocene.

The biogeographic history of Hyainailouridae is more complicated, with Indohyaenodon as the sister taxon of the clade that includes Koholia, Tritemnodon, Kyawdawia, Paratritemnodon, Metasinopa, and Hyainailouridae. Their common node (T51) is resolved near the K/Pg boundary and is unambiguously Afro-Arabian in origin (LO = 77%; BBM = 84%), indicating a Paleocene or early Eocene dispersal of the ancestor of Indohyaenodon to Asia from Afro-Arabia. Koholia + Tritemnodon (T55) is a clade that is also Afro-Arabian in origin, indicating the ancestor of Tritemnodon dispersed during the late Paleocene or early Eocene from Afro-Arabia to North America, possibly following the dispersal route of Indohyaenodon through Asia. Clade T54 is unambiguously Afro-Arabian in origin (LO = 71%; BBM = 84%) and it includes Hyainailouridae (T57: Apterodontinae + Hyainailourinae), an unambiguously Afro-Arabian clade (LO = 76%; BBM = 90%); and a clade (T55) that unites Kyawdawia, Paratritemnodon and Metasinopa. The Kyawdawia clade (T55) is ambiguously resolved by PO as either Afro-Arabian or Asian, though the likelihood methods both support an Asian origin for the clade (LO = 83%; BBM = 67%). This result suggests the ancestor of the Kyawdawia clade most likely dispersed from Afro-Arabia during the late Paleocene or early Eocene. Later, the ancestor of Metasinopa dispersed from Asia back to Afro-Arabia during the middle or late Eocene.

Like the parsimony topology and standard Bayesian topology, Apterodontinae is Afro-Arabian in origin, and the ancestor of A. gaudryi dispersed from Afro-Arabia to Europe close to the Eocene/Oligocene boundary. The common ancestor of Hyainailourinae (T61) is unambiguously Afro-Arabian in origin. This reconstruction suggests Orienspterodon then dispersed during the early middle Eocene from Afro-Arabia to Asia. In the tip-dating analysis, European Kerberos is the sister taxon (T63) of European Pterodon dasyuroides and Afro-Arabian Metapterodon (T64). The Kerberos clade (T63) is ambiguously Afro-Arabian or European in origin, though both LO (84%) and BBM (90%) strongly support a European origin for this clade. These results support the hypothesis that the common ancestor of the Kerberos clade (T63) dispersed from Afro-Arabia to Europe during the middle Eocene and the common ancestor of Metapterodon dispersed from Europe to Afro-Arabia during the Oligocene.

All other hyainailourine clades are resolved with unambiguous Afro-Arabian origins, meaning North American Hemipsalodon likely dispersed from Afro-Arabia to North America during the middle Eocene, Paroxyaena dispersed from Afro-Arabia to Europe during the middle Eocene, and Hyainailouros sulzeri dispersed from Afro-Arabia to Europe during the early Miocene. Ultimately, according to the tip-dating analysis, Afro-Arabia is the center of hyainailouroidean radiation at the beginning of the Paleogene and is most likely the continent of origin for the ancestors of Indohyaenodontinae, Teratodontinae, Apterodontinae, and Hyainailourinae.

Discussion

Phylogenetic position of Brychotherium, Teratodontinae, and “Indohyaenodontinae”

In every phylogenetic analysis presented here, Brychotherium is placed within the clade Teratodontinae, a result consistent with the phylogenetic conclusions of Solé et al. (2014) and Rana et al. (2015). Also in every phylogenetic analysis Teratodontinae is recovered as a clade that is closely related to Hyainailouridae (the clade that includes Apterodontinae and Hyainailourinae). The only other analyses that examined the relationships among these three clades are Rana et al. (2015), which found Teratodontinae was the sister clade of Hyainailourinae, and Solé et al. (2014), which found Teratodontinae, Hyainailourinae, and Apterodontinae were distantly related clades. While it is encouraging that the different methods of phylogenetic analysis used in the present study recovered similar results, the results presented by Solé et al. (2014) and Rana et al. (2015), and the low support values for these clades in the present analysis suggest close examination of the relationships within Hyainailouroidea should continue as the fossil records of these clades improve and greater character sampling is conducted.

In both the standard Bayesian analysis and parsimony analysis, Brychotherium is placed in the same clade as Dissopsalis, Anasinopa, and Furodon (P40, B53) to the exclusion of other teratodontines (Figs. 17 and 18). The close relationship of Brychotherium with Dissopsalis and Anasinopa is not particularly surprising, because Solé et al. (2014) also included Brychotherium in their analysis based on the descriptions and images of more fragmentary specimens provided by Holroyd (1994) in her doctoral dissertation, and found a well-supported (Bremer = 4) Anasinopa-Brychotherium-Dissopsalis clade as part of a robust Teratodontinae (Bremer = 4) that also included Masrasector, Teratodon, and Glibzegdouia. The placement of early or middle Eocene Furodon as a derived teratodontine in the analyses presented here is more surprising, as this result differs from the parsimony-based results of Solé et al. (2014) and Rana et al. (2015), both of which found Furodon to be a basal hyainailourine. It is particularly noteworthy that these analyses place Furodon deep within Teratodontinae as the sister taxon of Miocene Anasinopa, requiring an extensive ghost lineage for the Anasinopa branch through most of the Eocene and Oligocene. Similarly extensive ghost lineages are also required by the topology recovered by Rana et al. (2015), who found Miocene Anasinopa and Dissopsalis to be paraphyletic with respect to Paleogene teratodontines, and with the oldest teratodontine in their analysis (Glibzegdouia) being the most deeply nested. Such long ghost lineages, and near-inversions of the expected relationship between node age and stratigraphic succession, hints at the possibility of a misplaced root for teratodontines in those analyses.

The tip-dating analysis presented here (Fig. 19) instead places Furodon as the sister taxon of all other teratodontines, diverging from the other species in the late Paleocene or early Eocene, a result that is more consistent with the stratigraphic succession of species. In the tip-dating topology, Miocene Anasinopa and Dissopsalis are supported (PP = 0.67) as sister taxa, significantly reducing the lengths of the ghost lineages implied by the other methods and previous studies. Tip-dating analysis also resolves Glibzegdouia and Brychotherium as tips of lineages that diverged from other teratodontines during the early and middle Eocene, respectively. This more basal position for Brychotherium is comparable to the position occupied by Brychotherium among “Afroasian proviverrines” in the study of Egi et al. (2005, in which Brychotherium is referred to as “African Sinopa”). The weakly supported, very basal teratodontine placement of Furodon is not radically inconsistent with the parsimony-based results of Rana et al. (2015), which placed Teratodontinae as the sister clade to Hyainailourinae, and Furodon as the basal-most sister group of Hyainailourinae, implying that basal-most hyainailourines and teratodontines might be very similar morphologically. Solé et al.’s (2014) analysis placed Furodon far from Teratodontinae, at the base of Hyainailourinae in a hyainailourine-koholiine clade; note, however, that Solé et al. (2014) sampled fewer taxa than did Rana et al. (2015), and the expanded sampling of Rana et al. might help to explain why their results are more consistent with those presented here.

In the parsimony and standard Bayesian analyses, Metasinopa is the sister taxon of Teratodontinae. But, we do not define Teratodontinae based on the position of Metasinopa because tip-dating analysis resolved Metasinopa in a clade with Kyawdawia and Paratritemnodon, suggesting future work on the anatomy of Metasinopa is required to help clarify its position within Hyaenodonta. Kyawdawia and Paratritemnodon, recovered as part of the clade Indohyaenodontinae with Indohyaenodon by Solé et al. (2014), are also placed in differing phylogenetic positions depending on the method of analysis. This instability is consistent with the results of earlier assessments of hyaenodont systematics. Solé et al. (2014) found Indohyaenodontinae to be the sister group of Apterodon + Sinopinae (which, in their analysis, includes Sinopa, Tritemnodon, Pyrocyon, and Prototomus). Rana et al. (2015) found that “indohyaenodontines” were paraphyletic, a result similar to the standard Bayesian topology (Fig. 18), which places the indohyaenodontines as successive sister taxa to Hyainailouroidea (Apterodontinae, Teratodontinae, and Hyainailourinae). The fact that expanded sampling of taxa and characters, first by Rana et al. (2015), and now by this study, congruently recovers “indohyaenodontines” close to Hyainailouroidea suggests that the still poorly documented “indohyaenodontines,” which may include Metasinopa, are likely to be of great importance for understanding the origin and dispersal of multiple Paleogene Afro-Arabian lineages.

Phylogenetic position of Akhnatenavus and Hyainailourinae

In every phylogenetic analysis performed in this study, Akhnatenavus nefertiticyon was placed as the sister taxon of early Oligocene Akhnatenavus leptognathus. This Akhnatenavus clade was recovered by every analysis nested within a clade of Oligo-Miocene Afro-Arabian hyainailourines, though the relationships within this clade differ between the topologies. In Solé et al. (2014), Akhnatenavus was placed as the sister taxon to Megistotherium, with Afro-Arabian Pterodon species being that clade’s sister group. Rana et al. (2015) found no resolution among hyainailourines, with Akhnatenavus falling into a polytomy that also included Hyainailouros, Koholia, Metapterodon, Oxyaenoides, and Pterodon (note that, in this analysis, Oxyaenoides is placed with hyaenodontines). In their description of Kerberos, Solé et al. (2015) performed a phylogenetic analysis that included Hyainailourinae and found “Pterodon” phiomensis to be the sister group of an unresolved clade that included Akhnatenavus, Isohyaenodon, and Hyainailouros. Ultimately, earlier phylogenetic studies and the results presented in this study have supported Holroyd’s (1999) decision to erect the genus Akhnatenavus rather than keeping A. leptognathus as a species within the genus Pterodon, as was originally done by Osborn (1909). All of these analyses demonstrate Hyainailourinae is a clade, and they illustrate the specific relationships within it are unstable. Future studies should focus on the relationships within Hyainailourinae as there is still much to resolve.

Pterodon is not monophyletic in any of the phylogenetic analyses presented here. Pterodon dasyuroides (the type species of Pterodon) was consistently placed in a more basal position than either “Pterodon” africanus or “Pterodon” phiomensis. Furthermore, “Pterodon” africanus and “Pterodon” phiomensis were not recovered as sister taxa using any method in this study. This result is consistent with the results of Solé et al. (2015) in which Pterodon dasyuroides was also placed at a basal node in Hyainailourinae, with “Pterodon” africanus in a more deeply nested position, and “Pterodon” phiomensis even more deeply nested as the sister taxon to the clade that includes Akhnatenavus. Other studies that have included Pterodon have combined multiple species of Pterodon—the Pterodon OTUs in Solé et al. (2014) and Rana et al. (2015) were a combination of African “P.” africanus, “P.” phiomensis, and “P.” syrtos, while Pterodon in Polly (1996) combines P. dasyuroides and “P.” africanus. The results of Solé et al. (2015) and the present study strongly suggest that the separate species included in Pterodon need to be reexamined and analyzed as separate OTUs in all future phylogenetic analyses, and that revision of the genus is in order. In his discussion of “Hyaenodontinae” (which then included Pterodon, Apterodon, Metapterodon, and Hyaenodon), Savage (1965) synonymized North American Hemipsalodon with Pterodon. Mellett (1969) disputed this, arguing that Hemipsalodon was distinct from Pterodon, though they do share dental and cranial similarities. The results of Solé et al. (2015) and the analysis presented here support the distinction between these taxa. Megistotherium and Hyainailouros are closely related OTUs in every analysis, though they are not consistently recovered as sister taxa as expected if Megistotherium is synonymous with Hyainailouros as suggested by Morales & Pickford (2005) and specifically synonymous with Hyainailouros bugtiensis as suggested by Morlo, Miller & El-Barkooky (2007). Solé et al. (2015) found a monophyletic Hyainailouros in their analysis based on dental characters. Based on the results of the present study, we suggest future analyses of hyaenodont systematics retain separate OTUs for the material referred to Megistotherium (Savage, 1973) and material referred to Hyainailouros, particularly Hyainailouros sulzeri (Ginsburg, 1980), to allow further examination of the relationships among these Miocene hyainailourines.

In both Bayesian analyses and in many MPTs, the major sister clade to Hyainailourinae is Apterodontinae. Hyainailourinae is defined here by the common ancestor of Orienspterodon and Hyainailouros. This is the first time that Orienspterodon has been included in a phylogenetic analysis and it supports the conclusions of Egi, Tsubamoto & Takai (2007), who advocated for a close relationship between hyainailourines and Orienspterodon rather than a close relationship between Orienspterodon, Paratritemnodon, and Kyawdawia, as was suggested by Lewis & Morlo (2010).

Grohé et al. (2012) undertook the first phylogenetic analysis of Apterodontinae and found that species of Apterodon formed a polytomy with Quasiapterodon and Metasinopa. They did not include any hyainailourines in their analysis and the sister clade to their Apterodontinae was Paratritemnodon + Kyawdawia; both of those Asian taxa are clearly in the phylogenetic neighborhood of Apterodontinae, but the results of the current analysis indicate that hyainailourines are probably even more critical for any phylogenetic evaluation of apterodontine relationships. Solé et al. (2014) performed the first phylogenetic analysis that included Apterodon alongside hyainailourines. Their study was limited to dental characters, and Apterodon was recovered as the sister clade to Sinopinae, an assemblage whose monophyly was not recovered by any of the analyses presented here. More recently, Solé et al. (2015) illustrated cranial features that are shared by Apterodontinae and Hyainailourinae, and elevated this group to Hyainailouridae, but these features were not converted into characters for their phylogenetic analysis, which recovered Apterodontinae and Hyainailourinae in a polytomy with Lahimia and Boualitomus. Cranial characters sampled from Polly (1996) were integrated into the analysis performed by Rana et al. (2015) and that study placed Apterodontinae as the sister clade to Teratodontinae + Hyainailourinae. The cranial features illustrated by Solé et al. (2015) were converted into characters for the character-taxon matrix presented here. The matrix also includes cranial characters from Polly (1996), and several new characters, and the “hyainailourid” hypothesis (Apterodontinae as the sister clade to Hyainailourinae) is supported, though it is disrupted in the parsimony analysis by the occasional incursion of “wildcard” taxa. For the sake of improved communication, we support and encourage the future use of the family-level nomen Hyainailouridae, which we recommend for the clade that includes Apterodon gaudryi, Hyainailouros sulzeri, and their last common ancestor. Furthermore, given the consistent placement of Teratodontinae as a major sister group of Hyainailouridae using all phylogenetic methods, we propose the use of the superfamily Hyainailouroidea for the clade that includes Apterodon gaudryi, Hyainailouros sulzeri, Teratodon spekei and their last common ancestor.

Phylogeny of Hyaenodonta

In this analysis the possible basal hyaenodont Tinerhodon, a late Paleocene taxon from Morocco, was found to be more closely related to Altacreodus than to Hyaenodonta, though it should be noted that outgroup sampling is limited and this relationship will need to be reevaluated again as the matrix used here is eventually expanded to include other non-hyaenodonts. Tinerhodon was considered by McKenna & Bell (1997) to be part of Cimolestidae, but Gheerbrant et al. (2006) disputed its cimolestid affinities, suggesting that it was a basal hyaenodont based on comparisons to Boualitomus. Subsequent studies (Solé, 2013; Solé, Falconnet & Yves, 2014; Solé et al., 2014; Solé et al., 2015) found Tinerhodon to be the sister group of Hyaenodonta while Rana et al. (2015) found Tinerhodon to be the sister taxon of Lahimia + Boualitomus, a clade that occupied multiple positions in their analysis. None of the analyses performed here recover Lahimia and Boualitomus in such a basal position, and none place Tinerhodon as the sister taxon to Hyaenodonta to the exclusion of Altacreodus. Tinerhodon has large metaconids that project well above the paraconids, well-developed talonid cusps that include an additional cusp along the entocristid, and wide talonids basins, characters not shared with Boualitomus and Lahimia; the placement of these taxa in separate clades is consistent with the gross morphology of the dentition. Currently, Tinerhodon is only known from isolated lower teeth, and more morphological information from upper teeth would help to further test the phylogenetic position of this Afro-Arabian taxon, as it may either have important biogeographic implications for Hyaenodonta, or be of no relevance to the clade.

Crochet (1988) erected the subfamily Koholiinae to contain Koholia, which was, at the time, the oldest-known Afro-Arabian hyaenodont. Solé et al. (2009) added Boualitomus and Lahimia to Koholiinae based on wear patterns inferred from the fragmentary upper dentition of Koholia that were used to reconstruct the lower dentition and make comparisons to Lahimia. Solé et al. (2014) later found a monophyletic Koholiinae that also included Metapterodon, which together were placed as the sister clade to Hyainailourinae. None of the phylogenetic analyses presented here recover a monophyletic Koholiinae. Instead, Koholia is either in an unresolved position relative to other hyainailouroids (using parsimony), closely affiliated with Hyainailourinae (using standard Bayesian inference), or is a sister group of Hyainailouridae to the exclusion of Teratodontinae (using Bayesian tip-dating). The monophyly of Lahimia + Boualitomus, on the other hand, is strongly supported in both Bayesian analyses, in which it is placed outside of Hyainailouroidea. The other possible “koholiine,” Metapterodon, is the weakly supported sister taxon of Koholia using standard Bayesian analysis. Tip-dating suggests a close relationship between Metapterodon and Pterodon dasyuroides; a placement with hyainailourines is also present in the parsimony-based Adams consensus. The Adams consensus tree demonstrates Koholia is a “wild card” taxon in the parsimony analysis, resolving Koholia at a node outside of Hyainailouroidea and the clade that includes Lahimia and Boualitomus. The instability of Koholia is likely related to the fact it is only known from a single, fragmentary maxilla fragment. Many of the taxa included in this analysis are known at least from relatively complete dentary material. Additional material, especially dentary material, referred to Koholia would likely help clarify its relationship to other hyaenodonts.

The tip-dating approach employed here (a first for Hyaenodonta) is of particular interest given the great age of Lahimia, because some have proposed an African origin of hyaenodonts solely on the basis of the antiquity of Lahimia (Solé, 2013; Morlo et al., 2014; Solé et al., 2014); however, in practice, the expected phylogenetic pattern that would support such an African origin (i.e., paraphyly of multiple African taxa with respect to non-African taxa) has not been found in any phylogenetic analysis that included these species. Even using tip-dating, Lahimia was consistently highly nested within Hyaenodonta, a result consistent with Rana et al. (2015), who also found Lahimia deeply nested in some MPTs. This result might be expected based on the dentition of Lahimia—P1 is absent and the lower molar metaconids are subequal in height to the paraconid, unlike taxa such as Altacreodus, Tinerhodon, and Eoproviverra, which have P1 and taller metaconids than paraconids. The talonid basin of Lahimia is also very narrow compared to the trigonid, and reduced compared to taxa positioned more basally in the analyses presented here. A deeply nested Lahimia nevertheless implies multiple, unsampled ghost lineages of hyaenodonts reaching into the earliest Paleocene and Late Cretaceous. The tip-dating analysis, which tends to favor topologies consistent with stratigraphic succession as more likely, resolved Lahimia + Boualitomus in a position comparable to the position of this clade in the maximum parsimony and standard Bayesian analysis; close to the origins of Hyainailouroidea, and diverging later than the Galecyon clade. This position has very weak support, but the result is consistent regardless of the method used to recover the topology. In the tip-dating analysis Lahimia + Boualitomus is supported by a very rapidly evolving branch, as are many basal branches (except the branch supporting clade T2, the clade that includes Proviverra and Arfia). These rapid basal rates are consistent with an explosive radiation of hyaenodonts during the Late Cretaceous or Paleocene, and we interpret this to be an adaptive radiation that may have involved filling vacant carnivore niche space on multiple continents before evolutionary rates slowed in the early Eocene, indicative of more stable, occupied niche space (e.g., Simpson, 1953).

Rana et al. (2015) proposed that Lahimia, Boualitomus, and Tinerhodon may not belong in Hyaenodonta, but are instead part of an Afro-Arabian radiation of mammals that converged on the carnivorous dental morphology characteristic of Hyaenodonta. The analyses presented here repeatedly recover Tinerhodon outside of Hyaenodonta, in agreement with the speculation of Rana et al. (2015). Additionally, we do not dismiss the possibility that Lahimia and Boualitomus, small taxa only known from dentary specimens, may be morphologically convergent with hyaenodonts. The explosive rates of evolution supporting Lahimia + Boualitomus are tangential support for the hypothesis as multiple lineages converge on carnivory and rapid evolutionary rates may obscure the independent origins of Lahimia and Hyaenodonta. However, neither Lahimia nor Boualitomus were recovered outside of Hyaenodonta with Tinerhodon. The only way to further test the hypothesis that Lahimia and Boualitomus are not hyaenodonts is to expand the ingroup to broadly sample from other, non-hyaenodont Paleogene clades.

One of the clades consistently resolved in a separate clade from Lahimia, Boualitomus, and all other Afro-Arabian taxa is Hyaenodontidae, a clade that includes Hyaenodon and some “proviverrines” that is contains several rapidly evolving branches. A close relationship between Hyaenodontinae and some European proviverrines was first demonstrated by Polly (1996) when Eurotherium was resolved as the sister taxon to Propterodon + Hyaenodon. Contra Solé (2013) and Solé, Falconnet & Yves (2014) who found Proviverra and Eurotherium were part of a monophyletic Proviverrinae, Polly (1996) proposed Proviverra as the sister taxon to all more deeply nested hyaenodonts and Eurotherium as closely related to Propterodon and Hyaenodon. Rana et al. (2015) were the first to include Propterodon and Hyaenodon in a phylogenetic analysis since Polly (1996); they also included many more proviverrines than Polly (1996) and found a monophyletic, entirely European Proviverrinae as the sister clade to Hyaenodontinae. Proviverra is the most deeply nested proviverrine in their analysis and Eurotherium is also deeply nested. The entire clade Hyaenodontinae + Proviverrinae in Rana et al. (2015) is deeply nested within Hyaenodonta. In Rana et al. (2015) the clade that contains Hyaenodontinae and Hyainailourinae is supported by a stem that includes North American “sinopanines” and Limnocyoninae. Arfia and possibly Lahimia are placed near the root of Hyaenodonta. The results presented in this analysis also resolve Arfia near the root of Hyaenodonta, though Proviverrinae is not monophyletic and deeply nested. Instead, some “proviverrines” are placed near the root of Hyaenodonta (Parsimony and standard Bayesian analysis) or in a clade that includes Arfia (tip-dating analysis) and other “proviverrines” are part of a paraphyletic group of stem taxa relative to Hyaenodontinae. Eoproviverra is resolved by parsimony and standard Bayesian analysis as the sister group of all other hyaenodonts, but is more deeply nested when evolutionary rates are incorporated into the analysis.

Solé (2013) and Solé, Falconnet & Yves (2014) proposed multiple clades within “proviverrines” including Sinopinae (including Sinopa, Prototomus, Tritemnodon, and Galecyon), Arfiinae (synonymous with the genus Arfia), and Proviverrinae. Rana et al. (2015) did not recover a monophyletic Sinopinae, instead placing Prototomus as a group that is paraphyletic with respect to all non-arfianine, non-limnocyonine hyaenodonts; the remaining “sinopanines” were placed as basal stem members of a Hyaenodontinae + Proviverrinae clade. None of the methods employed here support a monophyletic Sinopinae, and the positions of the “sinopine” taxa are highly variable depending on the method applied. Tritemnodon, a North American taxon with partially fused upper molar paracones and metacones, and paracones that are taller than metacones, is placed along the stem of Hyainailouroidea using standard Bayesian analysis and as a stem hyainailourid (as the sister taxon of Koholia) using tip-dating. The Galecyon clade occupies more basal positions, in close phylogenetic proximity to Limnocyoninae (except in the tip-dating topology). Both Bayesian methods place Sinopa in a clade with Prototomus and Limnocyoninae. In the standard Bayesian analysis, this clade is more closely related to Hyainailouroidea than to Hyaenodontinae, but tip-dating places Sinopa, Prototomus, and Arfia in a sister clade relationship with the large, rapidly evolving clade that contains Hyaenodontinae and Hyainailourinae. The unstable, or weakly supported phylogenetic positions of these clades may be complicated by convergence within multiple clades upon a diet correlated with relatively specialized dentition that is not fully adapted for hypercarnivory (the metaconid is retained; the talonid retains multiple cusps) but is also not as similar to the outgroup morphology as Eoproviverra or Proviverra. Clearly these Eocene North American taxa are vital for understanding the evolution of major radiations of hyaenodonts and their phylogenetic positions depend upon the analysis and the method employed. Further study of these taxa is required as demonstrated by Zack & Rose (2015).

Another intriguing possibility raised by the phylogenetic results presented here is that Hyaenodonta itself may not be monophyletic. The explosive radiation and long, unsampled ghost lineages of Hyaenodonta recovered by all analyses is partially driven by the great antiquity of Lahimia, a taxon consistently recovered along the stem of Hyainailouroidea. Lahimia may be a late Paleocene example of an endemic radiation of Afro-Arabian carnivorous mammals that includes Teratodontinae, Apterodontinae, and Hyainailourinae. In this scenario Hyaenodontidae is a separate, northern continent radiation of carnivorous mammals that converged on the three sets of carnassials that unite Hyaenodonta as a group. Like the Rana et al. (2015) hypothesis that Tinerhodon, Lahimia, and Boualitomus are not hyaenodonts, this hypothesis can only be tested in a study that samples broadly from other placental mammalian clades and includes a broad sample of hyaenodonts from both Hyainailouroidea and Hyaenodontidae.

Biogeographic history of Hyaenodonta

Each topology and each biogeographic method yielded consistent biogeographic origins for several constituent clades within Hyaenodonta. Hyaenodontidae is unambiguously European in origin and Hyaenodontinae unambiguously originates in Asia. Across all analyses, Limnocyoninae and Sinopa originated in North America. Teratodontinae, Apterodontinae, and Hyainailourinae (from the node shared with Akhnatenavus) are unambiguously Afro-Arabian in origin. The biogeographic origins of Hyainailouridae and Hyainailouroidea differ by topology. Most significantly, the origin of Hyaenodonta across all analyses is reconstructed with an ancestral area in Europe and an Afro-Arabian origin is not likely for Hyaenodonta using any of the methods or topologies employed in this analysis. The nodes closest to the root of Hyaenodonta are weakly supported, but the earliest Afro-Arabian taxon (Lahimia) and earliest Asian taxon (Prolimnocyon chowi) are consistently recovered in deeply nested positions. A European or North American origin of Hyaenodonta is problematic because hyaenodonts are unknown in Europe and North America before the Paleocene/Eocene boundary (Gingerich & Deutsch, 1989; Gunnell, 1998; Smith & Smith, 2001; Zack, 2011; Solé, 2013), though, unlike North America, the Paleocene of Europe is still not well sampled. Tip-dating analysis indicates that the Late Cretaceous through the Paleocene were periods of rapid evolution for Hyaenodonta when many of the major hyaenodont clades originated. This explosive radiation is not fully captured in the fossil record. It also remains the case that Asia and Afro-Arabia are particularly poorly sampled, with some geological intervals, like the late Paleocene, only represented by a few, sparse localities (Meng, Zhai & Wyss, 1998; Seiffert, 2010) that may yet yield important fossils for understanding the origins of Hyaenodonta. Asia, in particular, is situated as a kind of keystone between the other continental areas and very early dispersals to North America or Europe from Asia may explain the sudden Eocene emergence of the group in Europe and North America, which shared a connection as indicated by common hyaenodont taxa (Arfia, Galecyon, Prototomus) between the continents. Another confounding factor in understanding the possible European roots of Hyaenodonta is the fragmentary biogeography and geography of Europe during the early Paleocene and early Eocene with different faunal zones spread across the continent (Hooker, 2010; Solé, 2013; Solé, Falconnet & Yves, 2014) with unsampled or isolated regions possibly serving as the center of origin for early Hyaenodonta. Simulation studies of biogeographic hypotheses show biogeographic signal is dramatically affected by missing data (Turner, Smith & Callery, 2009). Future attempts to examine the origins of Hyaenodonta would benefit from new discoveries from the early Paleogene of each continental area, but particularly Asia as these discoveries may help clarify the dispersal pathways of Hyaenodonta across the northern continents.

One goal of this study was to test the Afro-Arabian origin hypothesis for Hyaenodonta advocated by multiple authors (Gingerich & Deutsch, 1989; Gheerbrant et al., 2006; Morlo et al., 2014; Solé et al., 2014), which has never been tested using a phylogenetic analysis and explicit biogeographic method with assumptions defined. The most recent argument for an Afro-Arabian origin hypothesis was based on the discovery of Lahimia, and the assumption that this taxon is basal within Hyaenodonta or represents an early-diverging clade (“Koholiinae”) from Hyaenodonta (Grohé et al., 2012; Morlo et al., 2014; Solé et al., 2014). The study by Rana et al. (2015) and the results presented here do not support Lahimia as a particularly basal hyaenodont, but rather as a near the root of the clade that ultimately gave rise to Hyainailouroidea. Notably, Lahimia and Boualitomus were consistently recovered at more basal nodes than Asian “indohyaenodontines,” whose placements either within, or basal to, Hyainailouroidea indicate that they may be critically important for understanding the biogeographic origins of Hyainailouroidea and its Afro-Arabian sub-clades.

In the Bayesian analyses and in some MPTs, the Afro-Arabian hyaenodonts were recovered as part of the same Hyainailouroidea clade. The origin of this clade is ambiguous, but once it arrived in Afro-Arabia, likely between 75 and 65 Ma (95% HPD interval for T41), it appears to be an endemic radiation with multiple, later dispersals of hyainailouroidean lineages out of Afro-Arabia to the northern continents. Tinerhodon diverged from Altacreodus during this same interval and may have dispersed to Afro-Arabia with the earliest Afro-Arabian hyaenodonts during the Late Cretaceous or earliest Paleocene. Alternatively, Hyainailouroidea may be a clade convergent upon carnivory and only distantly related to Hyaenodontidae. In this scenario, they originated in Afro-Arabia and later dispersals to the northern continents occurred sporadically throughout the Paleogene and early Neogene. All analyses support a dispersal from Afro-Arabia to Europe that led to Apterodon gaudryi, a dispersal hypothesis also supported by Lange-Badré & Böhme (2005). Both Bayesian analyses support a dispersal from Afro-Arabia to Asia to account for the presence of Orienspterodon in Myanmar, and the Bayesian topologies support dispersals during the middle Eocene from Afro-Arabia to account for the presence of Kerberos, Paroxyaena, and Pterodon dasyuroides in Europe, and Hemipsalodon in North America. Perhaps the lineage that led to Hemipsalodon crossed through Europe, as proposed by Solé et al. (2015), following the same dispersal pathway used by Kerberos and Paroxyaena. It is also possible, given the very weak statistical support for the paraphyly of Hemipsalodon, Paroxyaena, Kerberos, and Pterodon dasyuroides with respect to Afro-Arabian taxa, that additional study of these species will reveal that they are in fact a monophyletic radiation derived from a single out-of-Africa Eocene dispersal that occurred during the middle Eocene around the Lutetian-Bartonian boundary or slightly later (near MP 16 as proposed by Solé et al., 2015). All topologies support Hyainailouros sulzeri dispersing from Afro-Arabia to Europe, and tip-dating analysis reconstructs this dispersal as an early Miocene event, consistent with the middle Burdigalian dispersal interval proposed by Ginsburg (1980) and coincident with the formation of the “Gomphotherium landbridge” that was established between Afro-Arabia and Eurasia by the early Miocene (Sen, 2013).

The biogeographic origin of Hyainailouroidea is complicated by the differences between the tip-dating topology and the standard Bayesian and maximum parsimony topologies. Instead of occupying stem positions relative to the Afro-Arabian clades, the “indohyaenodontines” are weakly supported as closely related only to Hyainailouridae in the tip-dating tree, implying that the ancestor of Indohyaenodon dispersed from Afro-Arabia to Asia during the late Paleocene or early Eocene, and the common ancestor of Paratritemnodon, Kyawdawia, and Metasinopa dispersed from Afro-Arabia to Asia during the same interval (Thanetian to Ypresian), with Metasinopa dispersing from Asia back to Afro-Arabia during the middle Eocene. There is very weak support for the nodes supporting a paraphyletic “Indohyaenodontinae” in either the standard Bayesian or tip-dating analysis. Koholia and Tritemnodon, two “wild card” taxa form the clade (T53) that intervenes between Indohyaenodon and the Kyawdawia/Paratritemnodon clade (T55) in the tip-dating analysis and future studies may recover a monophyletic Indohyaenodontinae similar to Solé et al. (2014). Regardless, the close relationship between Asian and Afro-Arabian hyaenodonts, and the dispersal pathways implied by these relationships, finds parallels in multiple mammalian lineages. Dispersal from Asia to Afro-Arabia during the Ypresian is possible in multiple mammalian lineages, potentially but not unambiguously including the zegdoumyid-like ancestor of anomaluroid rodents, the ancestor of caenopithecine adapiform primates, and the ancestor of more crownward strepsirrhine primates (Seiffert, 2012), though dispersal from Europe is also possible (see below). If the ancestor of Metasinopa dispersed from Asia during the middle Eocene this would coincide with the interval when hystricognathous rodents and anthropoid primates also likely dispersed from Asia to Afro-Arabia. If the ancestor of Orienspterodon dispersed from Afro-Arabia to Asia during the middle Eocene, it may have utilized similar routes as the anomaluroid rodent Pondaungimys, which also dispersed from Afro-Arabia to Asia during this interval (Sallam et al., 2010; Seiffert, 2012; Marivaux et al., 2015).

Using BBM, the common node of “Indohyaenodontinae” + Hyainailouroidea is reconstructed as North American (parsimony; P34), Asian (standard Bayesian, B44), or Afro-Arabian (tip-dating, T43). These conflicting reconstructions make it difficult to confidently assert the ultimate origins of these clades, though there is evidence of dispersal between Afro-Arabia and Europe as initially proposed in early studies of Fayum Hyaenodonta (Andrews, 1904; Andrews, 1906; Osborn, 1909; Schlosser, 1911). Gheerbrant & Rage (2006) note a minor exchange event from Europe to Afro-Arabia near the Lutetian/Bartonian boundary that includes caenopithecine adapiform and “anchomomyin” primates, and the European origin of African caenopithecines was supported by the phylogenetic analyses of Seiffert, Costeur & Boyer (2015). The dispersal of the ancestor of Paroxyaena from Afro-Arabia to Europe (as supported by both Bayesian analyses) may have occurred during this interval (MP 16), following the same dispersal route as Kerberos (MP 16), Hemipsalodon (Duchesnean), and Pterodon dasyuroides. The successful dispersal of hyainailourines to Europe and North America from Afro-Arabia during the middle Eocene has interesting implications for the structure of the hypercarnivorous niche at the time of the dispersal because the hypercarnivorous niche the was already occupied in Europe by hypercarnivorous “proviverrines” like Eurotherium (Solé, Falconnet & Vidalenc, 2015), oxyaenids and possibly early species of Hyaenodon (Mellett, 1977; Lange-Badré, 1979; Rose, 2006).

Multiple relationships recovered in the tip-dating analysis support exchange between North America and Europe near the Paleocene-Eocene Thermal Maximum (PETM), as suggested by Smith & Smith (2001) and Solé, Gheerbrant & Godinot (2013). The divergence (T35) between North American Pyrocyon + Gazinocyon (T36) and European Boritia + Galecyon (T37) reaches across the PETM, as does the divergence between European Galecyon morloi and the three North American Galecyon species; the divergence (T10) between Prototomus minimus and the clade (T11) that contains Prototomus phobos; and the divergence (T5) of European Arfia gingerichi and North American Arfia.

The nodes supporting Prolimnocyon chowi and Sinopa jilinia, both Asian taxa, are reconstructed with North American origins, evidence of exchange from North America to Asia during the late Paleocene and early Eocene. Exchange between Europe and Asia is implied by the close relationship between Hyaenodontinae, reconstructed with an Asian origin, and Oxyaenoides, reconstructed with a European origin. The West Siberian Sea was a major epicontinental seaway that separated Europe from Asia, limiting direct fauna exchange between these continents, but early Eocene exchange likely occurred, as documented in Perissodactyla (Hooker & Dashzeveg, 2003), Primates (Smith, Rose & Gingerich, 2006), and Rodentia (Badiola et al., 2009). The ancestor of Hyaenodontinae may have dispersed directly from Europe to Asia during this interval. Alternatively, Rana et al. (2015) recovered Hyaenodontinae as the sister clade to a monophyletic Proviverrinae. The stem of this clade includes North American taxa and Hyaenodontinae is ambiguously resolved with a North American or European origin. Further analyses will test the sister-taxon relationships to Hyaenodontinae. Hyaenodon is resolved with an Asian origin across all analyses with dispersal to Europe and North America during the late Eocene, and endemic radiations of the genus occurred after dispersal in Europe (Bastl, Nagel & Peigné, 2014) and North America (Mellett, 1977). These conclusions about the biogeographic history of Hyaenodon and Hyaenodontinae should be treated cautiously, as there are many species assigned to the genus, and they are found in Europe, Asia, and North America. The precise relationships among the different Hyaenodon species are not firmly established and further studies focused on the relationships within this genus will likely alter and refine the biogeographic conclusions found here with only four Hyaenodon OTUs.

The evolution of hypercarnivory within Hyaenodonta

The dental specializations of hyaenodonts—such as extended postparacristids and preprotocristids, elongate metastyles, and buccolingually compressed metacones—indicate that the group was adapted, like modern carnivorans, to a primarily faunivorous diet (Van Valkenburgh, 1999). But just as some lineages of carnivorans are more dentally specialized for carnivory than others, so too were some lineages of hyaenodonts. Hypercarnivory in modern carnivores is used to refer to animals that acquire 70% or more of their calories from meat, in contrast to generalist carnivores, which eat 50–60% meat and complete the diet with plant matter and invertebrates (Van Valkenburgh, 1988; Van Valkenburgh, 1989). Dental adaptations correlate with the dietary shift from generalist to hypercarnivore, including the mesiodistal lengthening of the carnassial complex, reduction and simplification of the talonid, reduction of the protocone, and loss of the metaconid (Holliday, 2010).

The first attempts to classify subgroups within Hyaenodonta were based on the degree of specialization in the dentition, particularly in the morphological specialization of the carnassial complex. “Proviverrinae” contained the less dentally specialized, or generalist taxa, and “Hyaenodontinae” contained the more specialized, hypercarnivorous taxa (Matthew, 1909; Matthew, 1915). The distinction between “Proviverrinae,” the hyaenodonts with prominent metaconids and unfused paracones and metacones, and “Hyaenodontinae,” the hyaenodonts with no metaconids and fused paracones and metacones, was utilized through most of the 20th century (Matthes, 1952; Savage, 1965; Van Valen, 1965; Barry, 1988). In this scheme, Pterodon and Hyaenodon were closely related based on their hypercarnivorous dentition (Barry, 1988). The genus Pterodon was defined by a groove between the paracone and metacone while Hyaenodon was recognized by its apparent fusion of these cusps (Savage, 1965). The arrangement caused some debate over how to classify taxa that did not neatly fit the dichotomy, like Apterodon, which has a reduced metaconid but separated paracones and metacones (Van Valen, 1965; Szalay, 1967) and Dissopsalis, which has long, sectorial metastyles and metacones but retains the metaconid on M2 and has a paracone that is reduced and not entirely fused to the metacone (Barry, 1988). Using cranial and postcranial characters, Polly (1996) demonstrated that Hyaenodon and Pterodon evolved specialized, hypercarnivorous shearing dentition (lost the metaconid and fused the paracone and metacone) independent of one another, and each lineage arose from separate “proviverrine” ancestors. In Polly (1996), Dissopsalis is the sister taxon to Hyainailourinae, with its elongate metastyle and prominent metacone reflecting the possible ancestral condition that led to the fused paracone and metacone of Pterodon, while Eurotherium is the sister taxon to Propterodon + Hyaenodontinae, with its divergent paracone and metacone possibly reflecting the ancestral morphology of its sister clade. Solé et al. (2015) detailed additional cranial features that distinguish Hyainailourinae from Hyaenodontinae, further emphasizing the ancient divergence of the two lineages with specialized, or hypercarnivorous dentition. With the separate origins of Hyaenodon and Pterodon supported by the present study and by Rana et al. (2015), the morphology of the carnassial complex in each lineage is worthy of reexamination.

Originally, the dental adaptations of Pterodon and Hyaenodon were assumed to be part of an evolutionary sequence. Pterodon, with incompletely fused paracones and metacones, was viewed as the ancestral condition for Hyaenodon (Matthew, 1915; Van Valen, 1967), which completely fused the paracone and metacone. However, carnivorous mammals have not adapted to hypercarnivory in the exact same way in every lineage that has evolved specialized shearing dentition, particularly in the arrangement of the metacone and paracone. Furthermore, with the establishment of Hyainailourinae and Hyaenodontinae as functionally convergent clades, it is possible to recognize that these two lineages converged on hypercarnivory through fundamentally different arrangements of the paracone, metacone, and metastyle, which in turn affects the occlusal morphology of the lower molars.

In Carnivora the carnassial complex is formed between P4 and M1. The upper carnassial blade stretches between the elongate metastyle of P4 and the buccolingually compressed postparacrista. This differs from the arrangement of the carnassials in carnivorous metatherians, which adapted the tricuspate molars into the upper carnassial rather than the bicuspate premolars. Borhyaenoid metatherians were the dominant carnivores in South America from the Paleocene through the Pliocene (Rose, 2006), and, like hyaenodonts, borhyaenoids formed a shearing carnassial complex between multiple upper and lower molars rather than one carnassial complex between a premolar and molar as in Carnivora. In borhyaenoids (i.e., Miocene Pseudolycopsis and Lycopsis) the upper carnassial is formed through mesiodistal elongation of the metacone and metastyle rather than the paracone and metastyle as in carnivorans (Van Valen, 1967; de Muizon & Lange-Badré, 1997). The paracone apex in borhyaenoids is distinct from the metacone, and much shorter than the taller shearing metacone. Borhyaenoids are not closely related to Dasyuromorphia, the Australian radiation of carnivorous marsupials that includes Thylacinus, but dasyuromorphians converged on the same shearing morphology as borhyaenoids. In Dasyuromorphia the paracone is retained as a distinct, reduced cusp and the metacone is mesiodistally elongate and buccolingually compressed, forming the tallest cusp of the trigon. The postmetacrista is sharp and slopes to meet the sectorial metastyle. This borhyaenoid and dasyuromorphian-style carnassial, with an augmented metacone and reduced paracone, is the same carnassial arrangement exhibited by many hyaenodonts, including Hyaenodon, Eurotherium, Dissopsalis, and Brychotherium (Fig. 23). Hyaenodon differs from Eurotherium, Dissopsalis, and Brychotherium in the degree of fusion between the paracone and metacone. In Hyaenodon, the paracone is a small, vestigial structure that fuses to the mesial metacone. Hyaenodon upper molars are often heavily worn but, in recently erupted M1, the distinct, small paracone is easily distinguished and it is even more evident in dP4. The paracone typically forms a small ridge on the mesial surface of M2 though it is easily worn away. Eurotherium, a middle Eocene taxon in the Proviverrinae/Hyaenodontinae clade, represents the likely ancestral morphology of the upper dentition of Hyaenodon, with the metacone more mesiodistally elongate and taller than the paracone. Oxyaenoides is the sister taxon to Hyaenodontinae and it also has a derived hypercarnivorous dentition with a tall metacone and extended metastyle. In Eurotherium, Oxyaenoides, and Hyaenodon, the paracone apex parallels the metacone apex, pointing ventrally. The arrangement of the upper molar carnassial cusps differs from the likely ancestral condition of the upper dentition, exemplified by Proviverra in Fig. 23. In Proviverra, the paracone and metacone are subequal in height and not buccolingually compressed. Rana et al. (2015) resolved Proviverrinae as the sister clade to Hyaenodontinae. In this scenario, metacone-dominated hypercarnivorous carnassials are convergent in Hyaenodontinae and in Proviverrinae, a hypothesis consistent with Solé, Falconnet & Yves (2014), which proposed multiple, independent hypercarnivorous lineages within Proviverrinae.

Figure 23 Carnassial specialization in Hyaenodontida.

Comparison of carnassial specialization in Hyaenodonta. Sketches show M2 of each taxon (name to the right of the tooth) in buccal view. Mesial direction is to the right of the image; distal direction is to the left. M1 and M2 for Hyaenodon shown. Schematized tree, including divergence estimates, are based on the tip-dating topology. Proviverra represents the unspecialized, condition of the upper molars in Hyaenodonta. Sinopa is slightly more specialized with a more buccolingually compressed paracone and metacone. Eurotherium represents the more specialized carnivorous dentition with the upper carnassial blade formed between the metacone and metastyle and the paracone is smaller, but unfused to the metacone. Hyaenodon represents a very specialized shearing dentition. The metacone is taller than the paracone and the paracone is fused to the mesial aspect of the metacone. Teratodontinae independently evolved specialized carnassial morphology from Hyaenodontinae, but their dental morphology is convergent. Hyainailourinae also converged on specialized, hypercarnivore-like dentition, but in this lineage the paracone is taller than the metacone and the metacone is fused to the distal aspect of the paracone. While not specialized for hypercarnivorous shearing, Apterodon also has taller paracones than metacones. M, metacone; P, paracone. Timeline abbreviations correspond to the first letter for each stage shown in Figs. 19 and 22.

The arrangement of the paracone and metacone are fundamentally different in Hyainailourinae. Instead of the metacone forming the tallest piercing cusp, the paracone is the tallest cusp, and this arrangement is exemplified by Pterodon dasyuroides and Akhnatenavus in Fig. 23. de Muizon & Lange-Badré (1997) noted the difference in paracone height in Hyaenodon and Pterodon but they did not place the distinction into a larger phylogenetic context. In hyainailourines, the metacone is fused to the distal face of the paracone. The postmetacrista becomes homologous to the P4 postparacrista in Carnivora. The paracone is also the tallest cusp of the trigon in Tritemnodon, an early Eocene taxon from North America and, based on the results of this analysis, probably part of the clade that includes Apterodontinae and Hyainailourinae. The trigons of Tritemnodon differ from those of Pterodon and Akhnatenavus by retaining a distinct apex on the metacone and a wider notch between the metacone and paracone. This arrangement represents the likely ancestral condition to the hyainailourine carnassial complex. Apterodontinae shares a taller paracone than metacone with Hyainailourinae.

Teratodontinae was consistently recovered in this study as part of the sister clade to the Apterodontinae and Hyainailourine. However, the dentition of Teratodontinae is arranged more like the dentition of borhyaenoids and derived proviverrines/hyaenodontines than it is like hyainailourines. In Brychotherium the metacone is slightly taller than the paracone and the metacone is more mesiodistally elongate than the paracone. In Dissopsalis the metacone is much taller and more elongate than the tiny paracone, which points slightly mesially rather than directly perpendicular to the alveolar plane. The recovery of Dissopsalis as part of a separate clade from Hyainailourinae differs from Polly (1996) whose topology recovered a sister-taxon relationship between Dissopsalis and Hyainailourinae. The topologies presented in this study imply that Dissopsalis provides evidence for yet another (third) convergence upon specialized hypercarnivory in Hyaenodonta, a possibility also raised by Solé et al. (2014).

The different arrangements of the upper carnassial influence the morphology of the trigonid on the lower molars. In Hyaenodontinae, the paraconid is almost mesial to the protoconid and the postparacristid and preprotocristid are nearly parallel to the mandibular corpus, reflecting the morphology of the postmetacrista and metastyle, which nearly parallel the alveolar margin. In Hyainailourinae, the paraconid is set lingual relative to the protoconid, giving the postparacristid and preprotocristid carnassial an oblique shearing angle relative to the mandibular corpus. This trigonid arrangement shears past a postmetacrista that is slightly lingually inflected at the carnassial notch, accommodating the wide base of the paracone.

Through biogeographic analysis and tip-dating analysis, the evolution of hypercarnivory in Hyaenodontinae, Proviverrinae, Hyainailourinae, and Teratodontinae can be reconstructed in place and time. These conclusions are a preliminary discussion of evolutionary trends inferred from this novel topology, and more detailed ancestral state reconstructions based on dental morphology would be an appropriate direction for future studies. Solé, Falconnet & Yves (2014) observed a general increase in body size and dental specialization in Proviverrinae through the Eocene, a trend supported by this analysis, which recovers the clade that includes Eurotherium as European in origin and dentally specialized like Hyaenodontinae. Hyaenodontinae is even more specialized than Eurotherium, and likely originated in Asia during the early Eocene. Then the hypercarnivorous Hyaenodon dispersed from Asia to Europe and North America where endemic radiations occurred (Mellett, 1977; Bastl, Nagel & Peigné, 2014). Hyainailourinae likely originated in Afro-Arabia and the carnassial morphology dominated by the paracone rather than the metacone also likely originated in Afro-Arabia. Hypercarnivory evolved a second time in Afro-Arabia in the lineage that led to Dissopsalis and Anasinopa, which both possess carnassials dominated by metacones that were taller than paracones. Given the time-calibrated tip-dating topology, this lineage diverged from Masrasector and Teratodon, taxa with less specialized shearing molars, during the late Eocene, and Dissopsalis diverged from Anasinopa, a less-specialized carnivore, during the late Oligocene. Dissopsalis is at the end of the most rapidly evolving branch in the Miocene at 1.45% change/Ma. The increased rate of morphological change in Dissopsalis is expected to have slightly predated, or coincided with, the arrival of Carnivora in Afro-Arabia. This may reflect a general trend to hypercarnivory that left generalist niche space open for the earliest carnivoran immigrants like amphicyonids and the ancestors of Mioprionodon to exploit (Rasmussen & Gutiérrez, 2009), or rapid morphological change through the evolution of Dissopsalis may reflect niche specialization in carnivores as immigrant taxa crowded the carnivorous niche (Van Valkenburgh, Wang & Damuth, 2004). The last-surviving Afro-Arabian hyaenodonts of the Miocene—Isohyaenodon, Megistotherium, and Dissopsalis—were each highly specialized carnivores, and each lineage may have been ecologically vulnerable to extinction as an apex carnivore (Van Valkenburgh, 2007) and morphologically unable to explore novel morphospace with such specialized dentitions, as demonstrated in studies that examine the ecological and morphological flexibility of hypercarnivores and their generalist sister taxa (Holliday & Steppan, 2004; Holliday, 2010). These data can now be used to explore the timing and the ecological context of hypercarnivorous specialization in Hyaenodonta across four continents and can be compared to the timing and location of dental specialization in the other Paleogene carnivore lineages, such as Oxyaenidae, Carnivoramorpha, and Mesonychia.

Conclusions

The character-taxon matrix utilized for this analysis sampled from each hyaenodont lineage that has been proposed, both to place the newly described latest Eocene species Brychotherium ephalmos and Akhnatenavus nefertiticyon in a phylogenetic context, and to rigorously test the hypothesis that Hyaenodonta first arose in Afro-Arabia. All three phylogenetic methods used here supported the monophyly of the clades Apterodontinae, Hyainailourinae, Teratodontinae, and the clade that unites these three clades: Hyainailouroidea. Limnocyoninae, Arfia, Sinopa, and Hyaenodontidae (Hyaenodontinae and some European “proviverrines”) were also recovered consistently. B. ephalmos is one of the most completely known teratodontines, and is either deeply nested with Miocene taxa (parsimony and standard Bayesian results) or is a basal form that branched off from other teratodontines in the middle Eocene (tip-dating results). Akhnatenavus is resolved by all methods as a member of a hyainailourine clade that also includes Oligocene African “Pterodon,” Paroxyaena, Hemipsalodon, and younger Miocene hyainailourines.

All analyses also recovered Europe as the origin of Hyaenodonta. Hyaenodontidae is consistently recovered as an early-diverging branch with European origins. The hyaenodont origin in Europe is problematic as the group is not known from Europe before the PETM, but Asia and Afro-Arabia, both of which have poor Paleocene records and as such have repeatedly been envisioned as the likely home of as-yet unsampled Paleocene hyaenodont ghost lineages, have no support as continents of origin based on the branching patterns recovered here. The presence of Lahimia in the middle Paleocene of Africa, and Tinerhodon in the late Paleocene, are intriguing evidence of early Afro-Arabian hyaenodont diversity in Afro-Arabia, but neither is resolved as a sister group of all other hyaenodonts, or in a basal enough position to influence the geographic reconstruction for the hyaenodont root node.

Tip-dating analysis implies that the early evolutionary history of Hyaenodonta was consistent with an explosive adaptive radiation. The early evolution of Hyaenodonta apparently echoed the larger-scale trend of K/Pg radiations in mammals (Beck & Lee, 2014) and birds (Lee et al., 2014) in which rapid morphological change occurred over a short period of geological time, potentially leaving little or no time for the accumulation of morphological synapomorphies that might otherwise support basal branches. This rapid period of early radiation led to the establishment of endemic clades on different continents. The early biogeographic history of the group is difficult to unravel with the current sample, but once the major clades were established, there appears to have been little large-scale exchange of taxa between continents. Immediately after the PETM there were genera common to North America and Europe, but these shared genera did not persist in Europe, and Solé, Falconnet & Yves (2014) demonstrated that endemic European proviverrines occupied vacant niche space left by genera common to both continents. Dispersal was apparently most likely during the middle and late Paleocene, then only occurred sporadically through the Eocene, mostly as pulses between Afro-Arabia and the northern continents in Hyainailourinae and Apterodontinae. No proviverrine is known to have dispersed from Europe to North America, Afro-Arabia, or Asia during the Paleocene or early Eocene. These results suggest Teratodontinae was entirely limited to Afro-Arabia, though future examinations of the “indohyaenodontines” may affect interpretations of this clade in future analyses. Limnocyoninae is a North American clade that dispersed into Asia during the late Paleocene. On each of these continents, different lineages adapted to different carnivorous niches, and hypercarnivory emerged independently at least once in Eurasia and twice in Afro-Arabia.

Description of the new taxa Brychotherium ephalmos and Akhnatenavus nefertiticyon increases the diversity of the Fayum carnivore fauna and further expands the total faunal diversity of Afro-Arabia before the continent established a filtered contact with Eurasia through the late Oligocene and early Miocene. Better documentation of endemic clades like Hyainailourinae and Teratodontinae is necessary to understand the ecological context that the earliest Eurasian immigrant carnivorans encountered. Only with a detailed understanding of the early evolution of Hyaenodonta in Afro-Arabia is it possible to assess the ecological factors that may have led to the ultimate extinction of this widespread and morphologically diverse group of carnivores.

Supplemental Information

Supplemental Information 1 Descriptions fo morphological characters used in phylogenetic analysis.

Click here for additional data file.

Supplemental Information 2 Hyaenodontida ages and biogeographic ranges.

Each taxon included in the analysis with citations justifying the age ranges and continental provenance used as input for the tip-dating analysis and biogeographic analyses.

Click here for additional data file.

Supplemental Information 3 Bayesian tip-dating statistical results.

Node code corresponds to Figs. 19 and 22. Left portion of chart refers to nodes and clades, right portion of chart (three columns) refers to terminal taxon branch rates and are listed in alphabetical order by taxon. Median age expressed in Ma; Median relative rate expressed in relative % change/Ma compared to all branch rates; Posterior probability expressed as percentage; 95% conf., 95% confidence interval; Tip-dating age is age input for taxon for Bayesian tip-dating analysis.

Click here for additional data file.

Supplemental Information 4 Maximum parsimony biogeographic reconstructions.

Continental origins reconstructed for each node on maximum parsimony strict consensus topology. Left column P# values correspond to node code in Fig. 20. Clade name is a shorthand for identifying the node. Af, Afro-Arabia; As, Asia; E, Europe; NA, North America. Parsimony, parsimony optimization analysis; Likelihood, likelihood optimization analysis (values represent % likelihood); BBM, Bayesian Binary MCMC analysis (values represent % likelihood).

Click here for additional data file.

Supplemental Information 5 Standard Bayesian biogeographic reconstructions.

Continental origins reconstructed for each node on standard Bayesian topology. Left column B# values correspond to node code in Fig. 21. Clade name is a shorthand for identifying the node. Af, Afro-Arabia; As, Asia; E, Europe; NA, North America. Parsimony, parsimony optimization analysis; Likelihood, likelihood optimization analysis (values represent % likelihood); BBM, Bayesian Binary MCMC analysis (values represent % likelihood).

Click here for additional data file.

Supplemental Information 6 Bayesian tip-dating biogeographic reconstructions.

Continental origins reconstructed for each node on tip-dating Bayesian topology. Left column T# values correspond to node code in Fig. 22. Clade name is a shorthand for identifying the node. Af, Afro-Arabia; As, Asia; E, Europe; NA, North America. Parsimony, parsimony optimization analysis; Likelihood, likelihood optimization analysis (values represent % likelihood); BBM, Bayesian Binary MCMC analysis (values represent % likelihood).

Click here for additional data file.

Supplemental Information 7 Character-taxon matrix used in phylogenetic analysis.

Click here for additional data file.

Supplemental Information 8 Parsimony analysis input for TNT.

Click here for additional data file.

Supplemental Information 9 Input NEXUS file for standard Bayesian analysis in MrBayes.

Click here for additional data file.

Supplemental Information 10 Input NEXUS file for Bayesian tip-dating analysis in MrBayes.

Click here for additional data file.

Supplemental Information 11 All most parsimonious trees (MPTs) recovered in parsimony analysis.

Click here for additional data file.

Supplemental Information 12 Standard Bayesian “allcompat” consensus tree file.

Click here for additional data file.

Supplemental Information 13 Tip-dating Bayesian analysis “allcompat” tree with node statistics.

Note that there is no extant OTU in the analysis. Therefore one date had to be fixed. 54.5 Ma was fixed as the age of Indohyaenodon and all other terminal taxon ages were allowed to fluctuate within their published age range. To calculate the mean, median, and 95% confidence brackets for age estimates, the age results need to be converted by the difference between 54.5 and the mean tip-dating age of Indohyaenodon (54.5 − 42.9069 = 11.59). 11.59 Ma is added to the tip-dating age of ever taxon and node to get the actual mean age estimate. For instance, for the mean age of Akhnatenavus nefertiticyon the raw output from the tip-dating analysis is 22.73 Ma. To convert to the correct mean age: 22.85 + 11.59 = 34.44 Ma.

Click here for additional data file.

We thank our collaborators at the Egyptian Mineral Resources Authority and the Egyptian Geological Museum for facilitating fieldwork in the Fayum area, P. Chatrath for managing fieldwork, and multiple Fayum field crews for their efforts in excavating the L-41 locality. Scanning and imaging assistance at Duke University was provided by H. Sallam, and at Stony Brook University by S. Heritage. M.R.B thanks the many curators who have supported access to their collections including C. Argot and G. Billet (Muséum National d’Histoire Naturelle, Paris), M. Brett-Surman (National Museum of Natural History, Washington, D.C.), P. Brewer (Natural History Museum, London), L. Costeur (Naturhistorisches Museum, Basel), J. Chupasko and J. Cundiff (Museum of Comparative Zoology, Cambridge), J. Galkin (AMNH), G. Gunnell (DPC), M. Hellmund (Geiseltal Museum, Halle), J. Hooker (Natural History Museum, London), A. Lavrov (Paleontological Institute, Moscow), E. Mbua and F. Ndiritu (KNM), C. Norris (Yale Peabody Museum, New Haven), S. Pierce (Museum of Comparative Zoology, Cambridge), T. Smith (Institut royal des Sciences naturelles de Belgique, Brussels), S. Schaal (SMF), F. Solé (Institut royal des Sciences naturelles de Belgique, Brussels), B. Sanders (University of Michigan, Ann Arbor), G. Rössner (BSPG), A. Sileem (CGM), A. Vogal (Naturmuseum Senckenberg, Frankfurt am Main), E. Westwig (AMNH), N. Xijun (Institute for Vertebrate Paleontology and Paleoanthropology, Beijing), and R. Ziegler (SMNS). We also thank P. Chatrath and F. Ankel-Simons for preparing the fossils described here, E. Gheerbrant, F. Solé and A. Lavrov for sharing fossil casts, G. Gunnell, M. O’Leary and D. Krause for constructive discussion and feedback, N. Stevens for support and discussion, and S. Zack and F. Solé for reviewing the manuscript and offering insight and comments that greatly improved the quality of this manuscript. This is Duke Lemur Center publication #1332.

Institutional Abbreviations

AMNH American Museum of Natural History, New York

BMNM Natural History Museum, London, United Kingdom

BSPG Bayerische Staatssammlung für Paläontologie und Historische Geologie, Munich

CGM Egyptian Geological Museum, Cairo

DPC Duke Lemur Center, Division of Fossil Primates, Durham

KNM National Museums of Kenya, Nairobi

SMNS Staatliches Museum für Naturkunde, Stuttgart.

Additional Information and Declarations

Competing Interests

Author Contributions

Field Study Permissions

Data Deposition

New Species Registration

The authors declare that they have no competing interests.

Matthew R. Borths conceived and designed the experiments, performed the experiments, analyzed the data, contributed reagents/materials/analysis tools, wrote the paper, prepared figures and/or tables, reviewed drafts of the paper.

Patricia A. Holroyd conceived and designed the experiments, analyzed the data, reviewed drafts of the paper.

Erik R. Seiffert conceived and designed the experiments, performed the experiments, analyzed the data, contributed reagents/materials/analysis tools, reviewed drafts of the paper.

The following information was supplied relating to field study approvals (i.e., approving body and any reference numbers):

Permission to collect and export fossils was granted by the Egyptian Mineral Resources Authority (formerly the Egyptian Geological Survey and Mining Authority) and the Egyptian Geological Museum.

The following information was supplied regarding data availability:

All fossil material described that bears a DPC specimen designation is deposited at the Duke Lemur Center, Division of Fossil Primates, Durham, NC. The specimens associated with this description that are deposited at the Duke Lemur Center are: DPC 17627, DPC 11990, DPC 11569A, DPC 11569B, DPC 13518, DPC 7765, Available at http://lemur.duke.edu/discover/division-of-fossil-primates/.

All fossil material described that bears a CGM specimen designation is deposited at the Cairo Geological Museum, Cairo, Egypt (though as of June 3, 2016 these specimens are on loan to the Duke Lemur Center in Durham, NC). The specimens associated with this description that are deposited at the Cairo Geological Museum: CGM 83735 and CGM 83750.

Morphobank Project 2336, http://dx.doi.org/10.7934/P2336; MorphoSource project Available at http://morphosource.org/Detail/ProjectDetail/Show/project_id/200.

The following information was supplied regarding the registration of a newly described species:

Publication LSID: urn:lsid:zoobank.org:pub:4EB91175-33FF-4A6C-B5B2-2F9933C0DED9; Akhnatenavus nefertiticyon nov. sp. LSID: urn:lsid:zoobank.org:act:19CBE178-447C-4182-9AED-C70280CD0673.

Brychotherium nov. gen. LSID: urn:lsid:zoobank.org:act:A39C1414-CF72-4FDC-A087-9912FCEDB0C8; Brychotherium ephalmos gen. et. sp. nov. LSID: urn:lsid:zoobank.org:act:BCAACF37-E200-4172-A875-C4D5F6FFCEFB.

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
