# Peer review of "Hyainailourine and teratodontine cranial material from the late Eocene of Egypt and the application of parsimony and Bayesian methods to the phylogeny and biogeography of Hyaenodonta (Placentalia, Mammalia)"

_PeerJ, doi:10.7717/peerj.2639_

## Round 0.1 · original submission · Major Revisions

Please address the referees' comments in detail, especially where there are points of disagreement. The journal does not provide copy-editing, and thus you should carefully review the text, figure legends, and bibliography.

·

Basic reporting

No Comments

Experimental design

No Comments

Validity of the findings

Overall, the authors have taken great strides to make the data used in this study available. However, there is one significant exception that needs to be addressed.

A table indicating what specimens or references were used to score each taxon needs to be added to the supplemental information to ensure replicability. In the absence of a clear indication of what was used to score each OTU, the phylogenetic analysis is unreproducible. Its not clear which (if any) OTUs were scored from the literature and which were scored from direct observation of specimens. Its also difficult to get any sense of sample sizes. Looking at the matrix, I’m surprised that some characters that, in my experience, show substantial intraspecific variability are not scored as polymorphic for any OTU (e.g., character 84). I suspect that’s a reflection of only using one or two specimens to score most OTU’s, but in the absence of a list of material examined, its difficult to be sure. Finally, given the checkered histories of many hyaenodontid taxa, a list of material examined is critical both to verify that sampled specimens are correctly identified and to maintain the utility of the matrix in the face of future reidentifications and other taxonomic changes.

Additional comments

Overall, this is a very impressive contribution to our understanding of hyaenodontidan morphology, systematics, and evolution. It provides an extremely detailed and well-illustrated description of two new taxa and uses this material to clarify aspects of hyaenodontidan phylogeny. The manuscript is well-written, particularly the discussion of the evolution of carnassial specialization. There are, however, places where the contribution could stand improvement.

Minor comments are made to the attached pdf.

In terms of more significant issues, I think the manuscript needs to do a better job of explaining how (or if) the three analyses of the character/taxon matrix combine to improve our understanding of hyaenodontid evolution than a single analysis would. The manuscript devotes considerable space to describing the results of the three analyses and stating how the topologies and biogeographic reconstructions differ across the three analyses, but there’s not enough effort devoted to filtering through that information to explain what the important take home messages are, particularly in terms of synthesizing that across analyses. In particular, is there anything you can confidently say about hyaenodontid phylogeny based on one analysis that couldn’t be said based only on the other two? Is there anything that seems strongly supported based one analysis that is contradicted by one or more of the other two? My own sense of things is that most of the conflict across analyses concerns areas of the trees with weak statistical support or (in the case of the parsimony analysis) poor resolution.
Along the same lines, the sections of the results devoted to the phylogenetic and biogeographic analyses (lines 1148-1700) could be shortened dramatically. The results of the two Bayesian analyses, in particular, are well-illustrated by Figures 18-19 and 21-22. A granular description of the topologies and biogeographic reconstructions is unnecessary. In the case of the parsimony analysis, there is ambiguity due to the presence of numerous MPTs and considerable non-resolution of the consensus. Even here, however, far too much space is devoted to descriptions of aspects of the topology that are evident from the figures, which obscures the information that is worth presenting. It would be better to focus on analysis and interpretation, including comparisons to the results of prior analyses. Some of this is included, but it gets buried in excessive descriptions of tree topology. For instance, the paragraph from lines 1208-1231 obscures two informative observations (Hyaenodontinae is deeply nested within “Proviverrinae”; the position of Oxyaenoides is very different from Rana et al.) by including sentence after sentence describing the taxa subsumed by individual nodes, which should be obvious from Figure 18.

The matrix itself is in good shape given the number of characters and taxa involved. Nevertheless, there are a number of issues that need to be addressed before publication. Because many of the results of the phylogenetic analyses are unresolved or weakly resolved, I suspect that making the appropriate changes to the matrix will result in topological changes which, in turn, will necessitate appropriate changes to the text and figures. As a result, while most of the changes to the matrix are likely to be relatively minor, their effect on the manuscript is likely to be substantial.
1. There are several errors in the scorings that should be corrected:
a. Galecyon chronius: scored for character 84 (M3 metacone). Unless the authors have access to unpublished material, the morphology of M3 is unknown in Galecyon. I suspect this character was accidentally scored from M2.
b. Koholia: scored for character 41 (m2 talonid length). The lower dentition of Koholia is unknown, and no other lower character is scored, suggesting that this scoring is in error.
c. Limnocyon verus: two characters describing the morphology of the m3 talonid are scored for L. verus. Given that m3 is absent in this taxon, these characters must have been inadvertently scored from m2. They should be rescored as “?”.
d. Megistotherium: assuming the postcranial scorings for Megistotherium are based on the material described by Savage (1973), the astragalar scorings should be removed. Ginsburg (1980) noted that the astragalus attributed to M. osteothlastes differs substantially from that of Hyainailouros sulzeri and concluded that the astragalus actually pertains to the large carnivoran Amphicyon major.
e. Oxyaenoides lindgreni: scored as having a fifth lower premolar (character 22, state 0). Neither account of the morphology of this species supports this scoring, and it should be rescored as “1”.
f. Prototomus minimus and P. phobos: the paracone of M1-2 is taller than the metacone in both species (P. minimus: Smith and Smith, 2001, fig. 2.3; P. phobos: Gingerich and Deutsch, 1989, fig. 20d). I’m not sure if the height difference is enough to score character 77 as “0” (paracone taller than metacone) versus “1” (paracone and metacone subequal), but the current scoring of “2” (metacone taller than paracone) is not correct.
g. There is inconsistency in how m3 structures are scored in taxa with reduced m3’s. Taxa that lack m3 are scored as “?” for character 39, which describes the size of the m3 metaconid, rather than state 2 (shorter than paraconid or absent). On the other hand, three species of Hyaenodon taxa that lack an m3 talonid are scored with state 2 (weak ridge or absent) for character 26, which describes development of the m3 entocristid rather than “?”.

2. The lower dentition of Paroyxaena galliae (“Schizophagus dilatatus”) needs to be scored. It would be unreasonable to expect that every possible character be scored for every taxon. Some material is difficult and/or expensive to access relative to its marginal benefit. As long as there is clear documentation of what material was examined, I’m fine with some potentially scoreable characters being left unscored. However, I don’t think the authors can ignore the entire lower dentition of a taxon, particularly since the lower dentition can otherwise be scored for every OTU except Koholia, making lower dental characters the core of the matrix. I suspect that the instability in the position of Paraoxyaena noted by the authors is attributable to the lack of lower dental scorings for either species. Material of “S. dilatatus” is adequately described and illustrated by Lange-Badré (1975, 1979) and Mathis (1985), so it shouldn’t be too difficult to score most characters. If it’s a question of an ambiguous synonymy of “S. dilatatus” with P. galliae versus P. pavlovi, I would suggest combining them into a single Paroxyaena OTU since there doesn’t seem to be any meaningful question as to whether Paroxyaena is monophyletic.

3. Pyrocyon strenuus: as long as other changes need to be made to the matrix, I would be happy to provide a cast of an upper dentition of P. strenuus since most upper molar characters are scored as missing data for this OTU.

4. States 0 and 1 of character 40, describing the relative lengths of m2 and m3, need to be swapped if it is going to be treated as ordered. The states of this character, as currently arranged, do not form a morphocline. The transition from state 0 to 1 requires an increase in the relative size of m3. The other two transitions (1 to 2, 2 to 3) require decreases in the relative size of m3.

5. There are a few pairs of characters that seem to violate the assumption of character independence that underlies parsimony analysis. This includes a two or three of pairs of characters that describe the same variation in serially homologous features on multiple dental loci. There are also a few character pairs in which the same variation is described multiple ways and one character that is dependent on two other characters. For all of these character sets, unless a compelling case can be made that selection is acting independently at each locus, these character pairs should be combined or one should be deleted:
a. Characters 9 and 14 describe the development of a paraconid on p3 and p4, respectively. The relationship between scorings is one way. The paraconid of p3 is never scored as stronger than the paraconid of p4. (Characters 11 and 16 describe entoconid development on the same loci but there’s at least a little evidence for independent selection)
b. Characters 38 and 39 describe metaconid development on m1 and m3, respectively. The metaconid of m3 is never stronger than the metaconid of m1. Both characters might be better off being replaced by one describing variation in the m2 metaconid. Its more likely to be scoreable than m1 because advanced taxa don’t wear it flat quite as badly, and its distribution shouldn’t be complicated by the small number of taxa that strongly reduce m3.
c. Character 37 describes the morphology of the m2-3 metaconids, while character 39 describes the size of the m3 metaconid relative to the paraconid. However, both characters appear to be describing metaconid size, albeit in different ways. Every taxon scored with state 2 (fold or ridge) or 3 (absent) for character 37 is scored as state 2 (metaconid shorter than paraconid or absent) for character 39.
d. Characters 46-48 describe cingulids on the lower molars. The first two characters describe the development of, respectively, the ectocingulid and postcingulid on m1-3. Character 48 describes the development of a link between them. As a result, character 48 is completely dependent on characters 46-47. All taxa scored as 1 (presence of a connection) for character 48 are scored as 1 (presence of an ectocingulid and postcingulid, respectively) for characters 46-47. This makes logical sense because it would be impossible to link two structures if one or both does not exist. These characters should be combined.
e. Character 57 describes development of a protocone/protocone lobe on P3. Character 58 describes the number of P3 roots. However, development of a protocone on P3 is probably driving acquisition of a third root. When the protocone is absent or small, P3 only needs two roots to support the crown. As the protocone becomes larger, a third root is needed. With the exception of one outgroup (which is scored as polymorphic for both characters), every taxon that is scored as having three roots to P3 (character 58, state 1) is either scored as having a protocone on P3 (character 57, states 1 and 2) or is not scored for this character (Megistotherium). Of the two taxa scored as having a large P3 protocone (character 57, state 2), one is scored as having three P3 roots (Teratodon), while the other (Paroxyaena gallinae) is not scored for character 58, presumably because the roots of P3 are not visible.

6. One final (minor) criticism of the character taxon matrix itself is that the characters could stand to be reordered. Characters 57-62 describe upper premolar morphology, as do characters 71 and 72. All of the upper premolar characters should be put together. Characters 85 to 87 are out of order with the remainder of the upper dentition characters and need to be put in logical sequence. The astragalar and calcaneal characters should also be together, not at opposite ends of the postcranial character list. Having related characters out of sequence can obscure patterns of character correlation. On a more practical level, this is frustrating because it can mean having to bounce back and forth between specimens when scoring a taxon.

7. In Supplemental Table S1, there are few places where character descriptions need clarification:
a. Character 35: would it be possible to add “premetacristid” to Figure 2? Its not a commonly used term, and it would be helpful to make sure its clear what you’re referring to.
b. Character 36: character states are confusing because each state refers to different crests. State 0 compares the premetacristid and preprotocristid, but states 1-2 compare the postparacristid to the preprotocristid. I’m guessing that state 0 should say postparacristid instead of premetacristid, but one way or another this needs to be clarified.
c. Character 78: states 0 and 1 refer to the “paracone/metacone separation,” but its not clear what that means in context. Are you referring to the point at which the protocone and metacone cease to be connate? Please clarify. If it is, then isn’t this character measuring paracone/metacone fusion as much as protocone height? A hyaenodontid with a low protocone and no basal fusion of the paracone and metacone would be scored the same as a hyaenodontid with a tall protocone and extensive fusion.

References cited in this review:
Barry, J. C. 1988. Dissopsalis, a middle and late Miocene proviverrine creodont (Mammalia) from Pakistan and Kenya. Journal of Vertebrate Paleontology 8:25-45.
Cope, E. D. 1884. The Creodonta. American Naturalist 18:255-267, 344-353, 478-485.
Gingerich, P. D., and H. A. Deutsch. 1989. Systematics and evolution of early Eocene Hyaenodontidae (Mammalia, Creodonta) in the Clarks Fork Basin, Wyoming. Contributions from the Museum of Paleontology, The University of Michigan 27:327-391.
Ginsburg, L. 1980. Hyainailouros sulzeri, mammifère créodonte du Miocène d'Europe. Annales de Paléontologie 66:19-73.
Gunnell, G. F. 1998. Creodonta; pp. 91-109 in C. M. Janis, K. M. Scott, and L. L. Jacobs (eds.), Evolution of Tertiary Mammals of North America. Volume 1: Terrestrial Carnivores, Ungulates, and Ungulatelike Mammals. Cambridge University Press, Cambridge.
Lange-Badré, B. 1975. Données récentes sur les Créodontes européens. Colloque international du CNRS 218:675-682.
Lange-Badré, B. 1979. Les créodontes (Mammalia) d'Europe occidentale de l'Éocéne supérieur a l'Oligocéne supérieur. Mémoires du Muséum National d'Histoire Naturelle, Série C 42:1-252.
Lavrov, A. V., and A. V. Lopatin. 2004. A new species of Arfia (Hyaenodontidae, Creodonta) from the basal Eocene of Mongolia. Paleontological Journal 38:448-457.
Mathis, C. 1985. Contribution à la connaissance des Mammifères de Robiac (Éocène supérieur): Creodonta et Carnivora. Bulletin du Museum National d'Histoire Naturelle, Section C 7:305-326.
Rana, R. S., K. Kumar, S. P. Zack, F. Solé, K. D. Rose, P. Missiaen, L. Singh, A. Sahni, and T. Smith. 2015. Craniodental and postcranial morphology of Indohyaenodon raoi from the early Eocene of India, and its implications for ecology, phylogeny, and biogeography of hyaenodontid mammals. Journal of Vertebrate Paleontology 35:e965308.
Savage, R. J. G. 1973. Megistotherium, gigantic hyaenodont from Miocene of Gebel Zelten, Libya. Bulletin of the British Museum (Natural History), Geology series 22:483-511.
Smith, T., and R. Smith. 2001. The creodonts (Mammalia, Ferae) from the Paleocene-Eocene transition in Belgium (Tienen Formation, MP7). Belgian Journal of Zoology 131:117-135.
Wortman, J. L. 1901-1902. Studies of Eocene Mammalia in the Marsh Collection, Peabody Museum. Part I. Carnivora. American Journal of Science 9:333-348, 437-450, 12: 143-154, 193-206, 281-296, 377-382, 421-432, 13: 39-46, 115-128, 197-206, 433-448, 14: 17-23.

·

Basic reporting

This manuscript represents an important step for the knowledge of the hyaenodonts. Besides the description of two new well-represented and interesting taxa (Brychotherium ephalmos and Akhnatenavus nefertiticyon), the authors bring numerous and important data concerning the geographic origin and morphologic evolution of these successful carnivorous mammals.
The manuscript is well written: the ideas are clearly and logically presented; no orthographic or grammatical error has been noted. I notably appreciate the introduction: it is a very nice reviewed of the history of the classifications of the hyaenodonts and on the phylogenies performed in order to resolve the question of their relationships.
The new taxa are finely and exhaustively described and compared (the diagnosis of Brychotherium ephalmos could however be shorten).
The methodology is sound and bring new interesting and crucial results. It is worth noting that the authors use a broad range of methods, exploring different possibilities and thus limiting the potential critics.
The creation of the concept of Hyainailouroidea makes sense and is clearly supported by the phylogenetic analysis. The discussion concerning the evolution of hypercarnivory is a fine updating of the results of Polly (1996) and of Muizon & Lange-Badré (1997).
The illustrations are fine. The new fossils are well presented. The availability of the scans is a real plus.
The references are up to date.
The supplementary files are important and provide necessary data.

We however have several critics. The most important reserve that I have is the lack of self-criticism. The authors do not insist enough on the differences in the fossil record between, on one side, Europe and North America, and on the other side, Asia and Africa for the Paleocene, and early and middle Eocene. This is particularly striking when compared to the huge diversity recorded in Europe and North America for these periods. Consequently, the two previously envisaged centres of origin (Asia and Africa) of Hyaenodonta are less represented in the phylogenetic analyses than the two geographic centres reconstructed in the present manuscript. Therefore, the biogeographic analyses are biased. I recommend that the authors write a point-by-point paragraph in order to discuss the limits of their study. I think that mentions of state of the art concerning the fossil record must be clearly explained in the manuscript. These observations could be considered either as arguments pro an European/North American origin (i.e. a poor Paleocene-early Eocene diversity could be due to an geographic origin elsewhere), or against an European/North American origin (i.e. two paleocontinents are too poorly represented). I think that this lack has strong influences on the results.
Moreover, the authors do not insist on the fact that the method used for phylogenetic analyses have strong impact on the results. One can note the numerous differences between trees presented. In my opinion, this highlights the instability of the matrix. This is highly understandable, but this must be discussed more clearly and not to be lost in numerous part of the manuscript. For instance, few nodes that have a PP superior to 75%, notably for the large clades, in the Standard Bayesian "allcompat" tree. The most strongly supported nodes correspond to groups for which the evolution is well-known (e.g. Proviverrinae) or which have peculiar morphology (e.g. Apterodontinae).
The numerous contradictory results (concerning the relationships and the biogeographic history) imply that the matrix must be improved in the future. This has to be highlighted more clearly in the text. The authors have made all that they can with the present data, but several problems have to be explored in the future.
Another important remark. We disagree with the use of Hyaenodontida. We think that Hyaenodonta is better even if the definition is different from that of Van Valen (1967). We actually use Creodonta even if the definition (i.e. its composition) has changed since its first mention. Moreover, Hyaenodontida is a badly constructed.
To conclude, I recommend minor revisions because everything looks good to me.

Experimental design

The methods used by the authors are all sound. The authors explored different ways to analyse the data. The utilisation of such modern methods is an important contribution to the study of hyaenodonts.
The addition of new character states and new characters, and redefinition of previous characters are very interesting novelties. However, the main problem when dealing with carnivorous mammals are the convergences to similar ecomorphotype. As noticed by the authors, the hyaenodonts evolved several times hypercarnivorous dentition. It seems that several characters could be redundant (e.g. 25 and 26, 28 and 39), and may be combined. Moreover, I have the sensation of the new characters often lead to group taxa based on convergent features
The most surprising relationships is the grouping of the hyaenodonts Tritemnodon agilis, Pyrocyon strenuus, Lahimia selloumi, Boualitomus marocanensis, and Preregidens langebadrae. This group is actually surprising because it contains one European, two North American, and two African. These hyaenodonts have indeed similar dentition, but I think that they share only convergent features: their dental morphology appear as a transient stage between omnivorous and hypercarnivorous dentition. Lahimia and Boualitomus are indeed characterized by the absence of p1, while the other taxa display two-rooted p1. The differences in the p1 agrees with geographic distribution; Preregidens is included among Proviverrinae (logical position to me) in the parsimony analysis. It could be interesting to perform phylogenetic analyses without these taxa. The results could be presented in Supplementary material.
Another possible analyse to perform would be to “force” the presence of African Tinerhodon disputatus among Hyaenodonta. It will be moreover interesting to analyse the Paleobiogeographic hypotheses that would result from this inclusion. The results could be also presented in Supplementary material.
Finally, the authors could also perform a phylogenetic analysis that include only Paleocene, early and middle Eocene taxa in order to reduce the impact of the convergences. The results could be presented in Supplementary material.
I think that India could be considered as an independent area from Asia in the biogeographic analyses, because (1) the Indian taxa (“Indohyaenodontinae”) seem to represent a monophyletic group, and (2) Indian subcontinent and its fauna have a particular history during the Paleocene and Eocene (Smith et al., 2016; New early Eocene vertebrate assemblage from western India reveals a mixed fauna of European and Gondwana affinities; Geoscience Frontiers; doi:10.1016/j.gsf.2016.05.001).
As made by Rana et al. (2015), the authors should perform phylogenetic analyses, on one side, with dental characters only, and, on the other side, with cranial and postcranial characters. This will allow to understand the impact of each type of data in the results.
The inclusion of A. gingerichi which is possibly the most primitive Arfia species could have impact on the biogeographic reconstruction because it is European and possibly close to the base of the trees – the sole Arfia species included are actually from North America. The same remark can be done for Galecyon gallus. Such early representatives could have impact on biogeographic hypotheses.

Validity of the findings

The fossils are important because they illustrate two new species, and they provide moreover knowledge on the entire dentition. Their description is an important contribution to the knowledge of the African hyaenodonts.
The phylogenetic results are very interesting. They allow discussing the origin of the hyaenodonts, as well as the evolution of the hypercarnivory in these mammals. However, the variable position (and composition) of several clades, as well as the possibilities that either Europe or North America may represent the center of origin of Hyaenodonta implies that there is still work to do and that the present study is an important step but not the conclusive end.
Concerning Europe as a possible center of origin for the hyaenodonts, the fossil record is indeed scarcer than the North American one. However, it is worth reminding that the late Paleocene fauna from Europe is well-known (Cernay, Berru, Rivecourt) and that several carnivorous mammals have been discovered (notably the mesonychids, which are generally rare).
The authors should mention that the relationships among proviverrines are very different from those found by Solé et al. (2014a). The close relationships between Cynohyaenodon and Quercytherium are not found, as well as those between Proviverra, Allopterodon and Leonhardtina. This result questions the validity of this matrix because the relationships among proviverrines previously proposed are indeed supported by the stratigraphic record.
To conclude, one can note that the theory of a North American origin of the Hyainailouridae recalls the hypothesis of Schlosser (1911) presented l.77-83. It could be interesting to remind this in he discussion.

Additional comments

Main text
l. 56 Concerning the dispersal of Hyainailouros in Eurasia, the authors could cite Ginsburg 1980 (Hyainailouros sulzeri, Mammifère créodonte du miocène d'europe; Annales de Paléontologie)
l.58. Replace Bear-sized by Rhinoceros-sized in order to be in accordance with the abstract.
l.60. Replace “Oxyaenida” by “Oxyaenidae”. Replace other mentions in the text. The name Oxyaenodonta has been proposed by Van Valen (1971; Adaptive Zones and the Orders of Mammals; Evolution) in order to elevate Oxyaenidae.
l. 77 Please cite Polly (1996). He clearly explained and illustrated the systematic of the “Proviverrinae” and “Hyaenodontinae”.
l.84-94 Please cite Gheerbrant (1995; Les mammifères paléocènes du basin d'Ouarzazate (Maroc) III. Adapisoriculidae et autres mammifères (Carnivora, ? Creodonta, Condylarthra, ?Ungulata et incertae sedis); Palaeontographica Abt. A) because he described at that time the first possible Paleocene hyaenodont, and Gingerich & Deutsch (1989) because he proposed Africa as the center of origin of Hyaenodonta.
l.113-l.128 Solé (2013) do not focus only on European proviverrines, but on early North American and European proviverrines in order to question the validity of “Proviverrinae”. Moreover, he used one cranial character and 14 postcranial characters from Polly (1996), thus before Rana et al. (2015).
l.134 The authors mistakenly cite the paper of Solé, Falconnet & Yves (2014a) instead of Solé (2014b) where the Teratodontinae are studied.
l.144-145 The mention of the use of the characters of Polly (1996) by Rana et al. (2015) is redundant with its previous mention (l.124-125).
l.212-223 As mentioned in Solé et al. (2015), Hyaenodontida has been mistakenly created and used by several authors. The name Hyaenodonta created by Van Valen (1967) has a different meaning but I think that this name should be preferred to that of Hyaenodontida because the latter is redundant in its construction.
l.446. Delete one “was”.
l.485 Replace “DP4” by DP4”.
l.550 Replace “Neither” by “No”.
l.659 I think that the authors should first figured the holotype (CGM 83750; Fig. 7), rather than DPC 17627 (Fig. 6) in order to be in accordance with the description.
l.749-l.761. The differences between the mandibles referred to Brychotherium ephalmos could represent sexual dimorphism. I think that the authors should discuss this point. Moreover, I do not find measurements of the height of the mandible. These data are useful for discussing sexual dimorphism question. This question has been discussed for hyaenodonts in Smith & Smith (2001) and Solé et al. (2015; New fossil Hyaenodonta (Mammalia, Placentalia) from the Ypresian and Lutetian of France and the evolution of the Proviverrinae in southern Europe; Palaeontology).
l.749 Please mention the figures that illustrate the additional mandibles (Figs 6, 8 and 9) in the text.
l.795 Please cite Osborn (1909) and Szalay (1967) rather than Lewis & Morlo (2010) concerning Apterodon macrognathus. The former articles actually provide good illustrations of the skull of this hyaenodont species.
l.815-l.860 The authors should compare the upper dentition of Brychotherium with the molar fragment of Glibzegdouia and with the molar of Furodon because they consider the latter species to be close to the new genus.
l.848-851. Please mention that the paracone is taller than the metacone in Furodon (thus different from what is observed for Brychotherium), as in Koholia and Hyainailourinae.
l.912. Please delete HYAENODONTIDA Solé, 2013 (redundancy).
l.949-954 The authors described only the skulls and upper dentition for the new species (A. nefertiticyon). So, how can they compared the lower dentition of A. leptognathus with that of A. nefertiticyon?
l.984 I would add “as” before “in Pterodon dasyuroides and Apterodon macrognathus”?
l.1051-1053. This particular morphology seems to be similarly present in Kerberos. It could be a feature of several (primitive?) hyainailourines.
l.1064-l.1146. I think that the authors must discussed the morphology of the upper molar of Furodon because the morphology of that tooth is close to that of Akhnatenavus (e.g. short protocone, partially fused paracone and metacone) rather than with Brychotherium. Furodon thus may represent a composite species.
l.1153. Please indicate the three uninformative characters.
l.1179-1181. Why mentioning Dissopsalis, Anasinopa, Furodon, but not Paroxyaena?
l.1232 The authors should indicate that Eoproviverra is represented only by few isolated teeth that show primitive features. This could explain its position outside Proviverrinae (contra Solé et al., 2014a).
l.1283 The authors should indicate that Preregidens has been considered as a proviverrine when described.
l. 1497 Add a space between “(P43)” and “–“ .
l.1530-1531 Based on Supplemental Table 6, it seems that the BBM value for Europe equals 76.72% (not 45.35%), while the LO value for North America equals 45.35% (not 76.72%).
l.1704 Please add “Indohyaenodontinae” to the title of this section because you discuss this group.
l.1851-1852. Lavrov (2007) only included Paroxyaena among Paroxyaenini.
l.1910-1911 The authors should also mention the loss of the p1 in Lahimia, which is very surprising for such an antique hyaenodont.
L1974-1975 Please cite Smith & Smith (2001).
l.2009 The authors should mention the work of Lange-Badré and Böhme (2005) because it deals with the dispersal of Apterodon (Apterodon intermedius, sp. nov., a new European Creodont Mammal from MP22 of Espenhain (Germany); Annales de Paléontologie).
l.2013-2014. A dispersal of Hemipsalodon from Africa to North America through Europe has been mentioned by Solé et al. (2015). Moreover, the North American taxa may have dispersed the same time as Kerberos and Paroxyaena (the two taxa appeared the same time (MP16) in Europe).
l.2036-2037. The tip-dating seems to reconstruct the common node (T53) of “Indohyaenodontinae”+Hyainailouroidae as African not North American (in green on Fig. 22).
l.2046-2050. The first occurrence of hyainailourines in Europe (Paroxyaena and Kerberos) date from MP16. The first occurrence of Hyaenodon in Europe is MP17a. Therefore, hyaenodontines appeared in Europe after hyainailourines, but hyainailourines found the hypercarnivorous niche possibly occupied by endemic proviverrines (e.g. Paenoxyaenoides). Moreover, as far as I know both Hyaenodon and Hemipsalodon appeared in North America during Duchesnean.
l.2059. The species Galecyon gallus has been erected by Solé, Gheerbrant & Godinot (2013) (Sinopaninae and Arfianinae (Hyaenodontida, Mammalia) from the Early Eocene of Europe and Asia; evidence for dispersal in Laurasia around the Paleocene/Eocene boundary and for an unnoticed faunal turnover in Europe; Geobios).
l.2186-2188. Solé, Falconnet & Yves (2014a) indicated that hypercarnivory probably evolved two times among Proviverrines (Oxyaenoides and Paenoxyaenoides). Solé et al. (2014) also remarked that the development of hypercarnivory is an important characteristic of several teratodontines that has evolved independently from hyaenodontines and hyainailourines.
l.2270-2271. Solé et al. (2014) also indicated that hyaenodontids radiated and dispersed (but around the P/E boundary in their article) in a first phase. This phase was followed by endemic evolution (p. 18 and fig. 5).
l. 1485 & l. 1943. The family names “Sinopaninae” and “Arfianinae” has been corrected into “Sinopinae” and “Arfiinae” by Solé, Falconnet & Yves (2014a).

Figures
Fig. 11. A view of the occipital area would be fine in order to compare the morphology of Akhnatenavus nefertiticyon with those illustrated in Polly (1997).
Why the authors figure two times the Adams consensus? They could illustrate the majority-rule consensus.
Fig. 16, 17, 18, 20, 22. Please correct Paroxyaena gailiae in Paroxyaena galliae
Fig. 16-22. Use everywhere Lesmesodon spp., as well as Eoproviverra eisenmanni in order to highlight that this species level.
The vertical lines on Fig. 19 are hard to understand and follow. It seems to correspond to Million years. It would be better to transfer the number from the top of the page (the Million years) to the bottom, or to move the stratigraphic scale at the bottom of the figure. The inadequacy of the vertical lines with the stratigraphic scale produces a strange effect.
Fig. 19. On Figs 16 and 17, Orienspterodon is included among Hyainailourinae, but not on Fig. 19.

Supplementary Tables
The data provided on S2 (Hyaenodontida data references) is not consistent with the matrix.

---

## Round 0.2 · accepted · Accept

The authors are to be congratulated on an excellent revision of the original manuscript. They have carefully addressed the numerous technical comments offered by the reviewers. I recommend the revised manuscript for acceptance for publication.